# Adaptive Methods Are Preferable in High Privacy Settings: An SDE Perspective

**Enea Monzio Compagnoni, Alessandro Stanghellini**
**Rustem Islamov, Aurelien Lucchi**
Department of Mathematics and Computer Science
University of Basel
Basel, Switzerland
{enea.monziocompagnoni, a.stanghellini,
rustem.islamov, aurelien.lucchi}@unibas.ch

**Anastasiia Koloskova**
Department of Mathematical Modeling
and Machine Learning
University of Zurich
Zurich, Switzerland
anastasiia.koloskova@uzh.ch

## Abstract

Differential Privacy (DP) is becoming central to large-scale training as privacy regulations tighten. We revisit how DP noise interacts with *adaptivity* in optimization through the lens of *stochastic differential equations*, providing the first SDE-based analysis of private optimizers. Focusing on `DP-SGD` and `DP-SignSGD` under per-example clipping, we show a sharp contrast under fixed hyperparameters: `DP-SGD` converges at a Privacy-Utility Trade-Off of $\mathcal{O}(1/\varepsilon^2)$ with speed independent of $\varepsilon$, while `DP-SignSGD` converges at a speed *linear in $\varepsilon$* with a $\mathcal{O}(1/\varepsilon)$ trade-off, dominating in high-privacy or large batch noise regimes. By contrast, under optimal learning rates, both methods achieve comparable theoretical asymptotic performance; however, the optimal learning rate of `DP-SGD` scales linearly with $\varepsilon$, while that of `DP-SignSGD` is essentially $\varepsilon$-independent. This makes adaptive methods far more practical, as their hyperparameters transfer across privacy levels with little or no re-tuning. Empirical results confirm our theory across training and test metrics, and empirically extend from `DP-SignSGD` to `DP-Adam`.

## 1 Introduction

The rapid deployment of large-scale machine learning systems has intensified the demand for rigorous privacy guarantees. In sensitive domains such as healthcare or conversational agents, even the disclosure of a single training example can have serious consequences. Legislation and policy initiatives show that AI regulation is tightening rapidly. In the United States, the *Executive Order of October 30, 2023* mandates developers of advanced AI systems to share safety test results and promotes privacy-preserving techniques such as differential privacy (Biden, 2023). Additionally, the NIST, a U.S. federal agency, released draft guidance (SP 800-226) on privacy guarantees in AI (NIST, 2023a) and included "privacy-enhanced" as a key dimension in its AI Risk Management Framework (RMF 1.0) (NIST, 2023b). In Europe, the *EU AI Act* sets binding obligations for high-risk systems (European Parliament, 2023), while ENISA recommends integrating data protection into AI development (European Union Agency for Cybersecurity (ENISA), 2025). In this context, DP (Dwork et al., 2006) is emerging as the de facto standard for ensuring example-level confidentiality in stochastic optimization. By injecting carefully calibrated noise into the training process, DP optimizers protect individual data points while inevitably trading off some population-level utility.

A central open question is how differential privacy noise influences optimization dynamics, and in particular, how it interacts with adaptivity and batch noise. In this work, we revisit this problem through the lens of SDEs, which, over the last decades, have proven to be a powerful tool for analyzing optimization algorithms (Li et al., 2017; Mandt et al., 2017; Monzio Compagnoni et al., 2023). While SDEs have not yet been applied to DP methods, we utilize them here to uncover a key and previously overlooked phenomenon: *DP noise affects adaptive and non-adaptive methods in structurally distinct ways*. We focus on two fundamental DP optimizers: `DP-SGD` (Abadi et al., 2016) and `DP-SignSGD`. The former serves as the baseline for DP optimization; although the latter is not widely used in practice, it is substantially simpler to analyze than the popular DP optimizer

`DP-Adam` (Gylberth et al., 2017; Zhou et al., 2020; Li et al., 2022b; McKenna et al., 2025). Relying on `SignSGD` as a proxy for `Adam` is standard in prior work (Monzio Compagnoni et al., 2025c; Balles & Hennig, 2018; Zou et al., 2023; Peng et al., 2025; Li et al., 2025), and this motivates our focus on `DP-SignSGD` for the theoretical development. We leave the study of more advanced DP optimizers to future work, as each would require a separate technical treatment. Under standard assumptions and with per-example clipping, our analysis isolates how the privacy budget $\varepsilon$, which governs the overall level of privacy, influences the dynamics.

In practice, private training is usually performed across a range of privacy budgets $\varepsilon$, and for each value one searches for the best-performing hyperparameters. A change in $\varepsilon$ can therefore arise either from this exploratory sweep or from stricter regulatory requirements as discussed above. To capture these two possibilities, we study two complementary protocols. **Protocol A (fixed hyperparameters):** To examine the situation when re-tuning is not feasible, e.g., due to low computational budget, we first fix a privacy budget $\varepsilon$ and find the optimal configuration $(\eta, C, B, \dots)$ via grid search. Then, we analyze how performance changes if training is repeated under different $\varepsilon$, without adjusting hyperparameters, thereby isolating the impact of $\varepsilon$ on performance. **Protocol B (best-tuned per $\varepsilon$):** When re-tuning is allowed, we search the optimal hyperparameters for each $\varepsilon$, thereby isolating the *intrinsic scaling of the optimal learning rates with respect to $\varepsilon$*.

**Contributions.** We make the following contributions:

1. We provide the first SDE-based analysis of differentially private optimizers, using this framework to expose how DP noise interacts with adaptivity and batch noise;

2. **Protocol A:** We show that `DP-SGD` converges at a speed *independent* of $\varepsilon$, with a privacy-utility trade-off that scales as $\mathcal{O}\left(1/\varepsilon^2\right)$, consistent with prior work;

3. **Protocol A:** We prove a novel result for `DP-SignSGD`: its convergence speed scales linearly in $\varepsilon$, while its privacy-utility trade-off scales as $\mathcal{O}\left(1/\varepsilon\right)$;

4. **Protocol A:** When batch noise is sufficiently large, `DP-SignSGD` always dominates. When batch noise is sufficiently small, the outcome depends on the privacy budget: for strict privacy ($\varepsilon < \varepsilon^\star$), `DP-SignSGD` is preferable, while for looser privacy ($\varepsilon > \varepsilon^\star$), `DP-SGD` has better performance;

5. **Protocol B:** We theoretically derive that the optimal learning rate of `DP-SGD` scales as $\eta^\star \propto \varepsilon$, while that of `DP-SignSGD` is $\varepsilon$-independent. This tuning allows the two methods to reach theoretically *comparable* asymptotic performance, including at very small $\varepsilon$;

6. We empirically validate all our theoretical insights on real-world tasks, and show that the qualitative insights extend from training to *test* loss and from `DP-SignSGD` to `DP-Adam`.

In summary, our results refine the privacy-utility landscape, which, to our knowledge, has not yet yielded a definitive answer on which of `DP-SGD` or `DP-Adam`/`DP-SignSGD` performs best, and in what regimes. Under Protocol A, adaptivity is preferable in stricter privacy regimes: `DP-SignSGD` converges more slowly but achieves better utility when $\varepsilon$ is small or batch noise is large, whereas `DP-SGD` converges faster but suffers sharper degradation. Under Protocol B, both methods achieve comparable asymptotic performance; however, adaptive methods are far more practical, as their optimal learning rate is essentially $\varepsilon$-independent, allowing it to transfer across privacy levels with little or no re-tuning. This matters not only for computational cost but also for privacy, since each hyperparameter search consumes additional budget (Papernot & Steinke, 2022). In contrast, `DP-SGD` requires an $\varepsilon$-dependent learning rate tuned *ad hoc*, making it brittle if the sweep grid misses the "right" value. Intuitively, adaptive methods inherently adjust to the scale of DP noise, whereas non-adaptive methods require explicit tuning of the learning rate to counter the effect of privacy noise.

## 2 RELATED WORK

**SDE approximations.** SDEs have long been used to analyze discrete-time optimization algorithms (Helmke & Moore, 1994; Kushner & Yin, 2003). The most popular framework to derive SDEs was rigorously formalized by Li et al. (2017). Beyond their foundational role, these approximations have been applied to practical tasks such as learning-rate tuning (Li et al., 2017; 2019) and batch-size selection (Zhao et al., 2022). Other works have focused on deriving convergence bounds (Monzio Compagnoni et al., 2023; 2024; 2025c; 2026), uncovering scaling laws that govern

optimization dynamics (Jastrzebski et al., 2018; Monzio Compagnoni et al., 2025b;c), and revealing implicit effects such as regularization (Smith et al., 2021; Monzio Compagnoni et al., 2023) and preconditioning (Xiao et al., 2025; Marshall et al., 2025). In particular, SDE-based techniques have been used to study a broad class of modern adaptive optimizers, including RMSProp, Adam, AdamW, and SignSGD, as well as minimax and distributed variants (Monzio Compagnoni et al., 2024; 2025b;c; Xiao et al., 2025). Despite this progress, prior work has exclusively focused on non-private optimization. To our knowledge, ours is the first to extend the SDE lens to DP optimizers, including explicit convergence rates and stationary distributions as functions of the privacy budget.

**Differential privacy in optimization.** Differentially private training is most commonly implemented via `DP-SGD` (Abadi et al., 2016), which clips per-example gradients to a fixed norm bound to control sensitivity and injects calibrated Gaussian noise into the averaged update. Advanced accounting methods such as the moments accountant (Abadi et al., 2016) and Rényi differential privacy (Mironov, 2017; Wang et al., 2019), combined with privacy amplification by subsampling (Balle et al., 2018; 2020), allow practitioners to track the cumulative privacy cost tightly over many updates and have made large-scale private training feasible. A central challenge is that clipping, while essential for privacy, also alters the optimization dynamics: overly aggressive thresholds bias gradients and can stall convergence (Chen et al., 2020), prompting extensive work on how to set or adapt the clipping norm. Approaches include rule-based or data-driven thresholds, such as `AdaCliP` (Pichapati et al., 2019) and quantile-based adaptive clipping (Andrew et al., 2021), as well as recent analyses that characterize precisely how the clipping constant influences convergence (Koloskova et al., 2023). Together, these contributions have positioned `DP-SGD` and its variants as the standard backbone for differentially private optimization.

**Adaptive DP optimizers.** Adaptive methods such as `AdaGrad` (Duchi et al., 2011; McMahan & Streeter, 2010), `RMSProp` (Tieleman & Hinton, 2012), and `Adam` (Kingma & Ba, 2015) generally outperform non-adaptive SGD in non-private training. However, under DP constraints, this performance *gap i)* narrows considerably (Zhou et al., 2020; Li et al., 2022a) and *ii)* essentially vanishes when both optimizers are carefully tuned, as observed for large-scale LLM fine-tuning in Li et al. (2022b, App. S). Consistent with non-DP training, non-adaptive methods are sometimes still preferred in vision tasks (De et al., 2022). Therefore, which of `DP-SGD` and `DP-Adam` is preferable remains an open question. Under assumptions that include bounded/convex domain, bounded gradient norm, bounded gradient noise, convexity of the loss, and possibly without performing clipping of the per-sample gradients, several strategies have been theoretically and empirically explored to mitigate the drop in performance of adaptive methods in DP. These include bias-corrected `DP-Adam` variants (Tang & Lécuyer, 2023; Tang et al., 2024), the use of non-sensitive auxiliary data (Asi et al., 2021), and scale-then-privatize techniques that exploit adaptivity before noise injection (Li et al., 2023; Ganesh et al., 2025). A recent related work by (Jin & Dai, 2025) studies `Noisy SignSGD`: Conceptually, they investigate how the sign compressor amplifies privacy, and argue that the sign operator itself provides privacy amplification beyond the Gaussian mechanism. Their analysis establishes convergence guarantees in the distributed learning setting while relying on *bounded gradient norms and bounded variance* assumptions, thereby avoiding the need for clipping and explicitly leaving its study to future work.

We view these contributions as providing valuable theoretical and empirical advances in the design of adaptive private optimizers, clarifying many important aspects of their behavior as well as trying to restore the aforementioned performance *gap*. Yet, the fundamental question of *which privacy regimes are most favorable to adaptivity* remains largely unanswered, and addressing it could explain at least one aspect of the nature of this *gap*. Our work addresses this *open question* by analyzing *why and when adaptivity matters* under DP noise, identifying the regimes where adaptive methods dominate and where they match non-adaptive ones. Crucially, we incorporate *per-example clipping*, a central element of `DP-SGD`, and a heavy-tailed batch noise model that captures unbounded variance.

## 3 PRELIMINARIES

**General Setup and Noise Assumptions.** We model the loss function with a differentiable function $f : \mathbb{R}^d \to \mathbb{R}$ with global minimum $f^* = 0$: This is not restrictive, as one can always consider the suboptimality $f(x) - f^*$ and rename it as $f$. Regarding noise assumptions, recent literature commonly assumes that the stochastic gradient of the loss function on a minibatch $\gamma$ of size $B \geq 1$ can be decomposed as $\nabla f_\gamma(x) = \nabla f(x) + Z_\gamma$ where batch noise $Z_\gamma$ is modeled with a Gaussian (Ahn et al., 2012; Chen et al., 2014; Mandt et al., 2016; 2017; Zhu et al., 2019; Jastrzebski et al., 2018; Wu

et al., 2020; Xie et al., 2021), often with constant covariance matrix (Li et al., 2017; Mertikopoulos & Staudigl, 2018; Raginsky & Bouvrie, 2012; Zhu et al., 2019; Mandt et al., 2016; Ahn et al., 2012; Jastrzebski et al., 2018). In this work, we refine the standard noise assumption to distinguish the two regimes induced by per-example clipping in DP training. In this setting, each mini-batch contains a mix of *clipped* and *unclipped* example-level gradients. For *unclipped* examples, we follow the usual literature and model the batch-averaged noise as Gaussian, e.g., $Z_\gamma \sim (\sigma_\gamma/\sqrt{B})\mathcal{N}(0, I_d)$. To capture the heavy-tailed gradient noise observed at the per-example level ($B = 1$), we model $Z_\gamma(x) \sim \sigma_\gamma\, t_\nu(0, I_d)$, a Student-$t$ with $\nu$ degrees of freedom (recovering the Gaussian case as $\nu \to \infty$). See Assumption B.2 and Remark B.2 for more details. Finally, we use the following approximation, formally derived in Lemma A.2: $\mathbb{E}\left[\frac{\nabla f_\gamma(x)}{\|\nabla f_\gamma(x)\|}\right] \approx \frac{\nabla f(x)}{\sigma_\gamma\sqrt{d}}$. The approximation is valid under two assumptions: $i$) The parameter dimension $d$ is sufficiently large ($d = \Omega(10^4)$), consistent with modern deep learning models that often reach billions of trainable parameters; $ii$) The signal-to-noise ratio satisfies $\frac{\|\nabla f(x)\|_2^2}{2\sigma_\gamma^2} \ll d$: This condition has been thoroughly empirically studied by Malladi et al. (2022) (Appendix G), who observed that across multiple tasks and architectures the ratio $\frac{\|\nabla f(x)\|_2^2}{2\sigma_\gamma^2}$ never exceeds $\mathcal{O}(10^2)$, well below typical values of $d$. See Remark A.1 for more details, including experimental validations. We highlight that our experiments confirm that the insights derived from our theoretical results carry over to real-world tasks. Importantly, while our theory is developed for `DP-SignSGD`, we further validate that the same insights hold empirically for `DP-Adam`, showing that our insights extend directly to this widely used private optimizer, as well as also transfer from training to test loss. This is a testament to both the mildness of the assumptions and the robustness of the analysis.

**SDE approximation.** The following definition formalizes in which sense a continuous-time model, such as a solution to an SDE, can accurately describe the dynamics of a discrete-time process, such as an optimizer. Drawn from the field of numerical analysis of SDEs (see Mil'shtein (1986)), it quantifies the disparity between the discrete and the continuous processes. Simply put, the approximation is meant in a *weak sense*, meaning in distribution rather than path-wise: We require their expectations to be close over a class of test functions with polynomial growth, meaning that all the moments of the processes become closer at a rate of $\eta^\alpha$ and thus their distributions.

**Definition 3.1** *Let $0 < \eta < 1$ be the learning rate, $\tau > 0$ and $T = \lfloor\frac{\tau}{\eta}\rfloor$. We say that a continuous time process $X_t$ over $[0, \tau]$, is an order-$\alpha$ weak approximation of a discrete process $x_k$, if for any polynomial growth function $g$, $\exists M > 0$, independent of the learning rate $\eta$, such that for all $k = 0, 1, \ldots, T$, $|\mathbb{E}g(X_{k\eta}) - \mathbb{E}g(x_k)| \leq M\eta^\alpha$.*

While we refer the reader to Appendix B for technical details, we illustrate with a basic example. The SGD iterates follow $x_{k+1} = x_k - \eta\nabla f_{\gamma_k}(x_k)$, and, as shown in Li et al. (2017), it can be approximated in continuous time by the first-order SDE

$$dX_t = -\nabla f(X_t)dt + \sqrt{\eta}\sqrt{\Sigma(X_t)}dW_t, \tag{1}$$

where $\Sigma(x) = \frac{1}{n}\sum_{i=1}^{n}(\nabla f(x) - \nabla f_i(x))(\nabla f(x) - \nabla f_i(x))^\top$ is the gradient noise covariance. Intuitively, the iterates drift along the gradient while the stochasticity scales with this covariance.

**Differential Privacy.** Here, we outline the relevant background of foundational prior work in DP optimization. We adopt the $(\varepsilon, \delta)$-DP framework (Dwork et al., 2006).

**Definition 3.2** *A random mechanism $\mathcal{M} : \mathcal{D} \to \mathcal{R}$ is said to be $(\varepsilon, \delta)$-differentially private if for any two adjacent datasets $d, d' \in \mathcal{D}$ (i.e., they differ in 1 sample) and for any subset of outputs $S \subseteq \mathcal{R}$ it holds that $\mathbb{P}\left[\mathcal{M}(d) \in S\right] \leq e^\varepsilon\mathbb{P}\left[\mathcal{M}(d') \in S\right] + \delta$.*

In this work, we consider example-level differential privacy applied by a central trusted aggregator. We implement this using the sub-sampled Gaussian mechanism (Dwork & Roth, 2014; Mironov et al., 2019) to perturb the SGD updates: At each iteration, a random mini-batch is drawn, per-example gradients are clipped to a fixed bound to limit sensitivity, and Gaussian noise is added to the averaged clipped gradients. The following definition formalizes these mechanisms and provides the update rules for `DP-SGD` and `DP-SignSGD`.

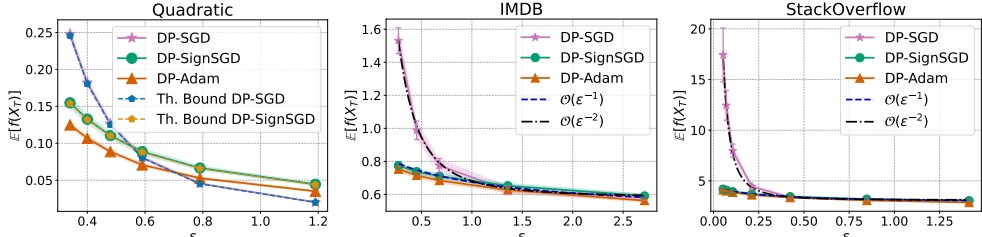

Figure 1: Empirical validation of the privacy-utility trade-off predicted by Thm. 4.1 and Thm. 4.3, comparing `DP-SGD`, `DP-SignSGD`, and `DP-Adam`: Our focus is on verifying the functional dependence of the asymptotic loss levels in terms of $\varepsilon$. **Left:** On a quadratic convex function $f(x) = \frac{1}{2}x^\top H x$, the observed empirical loss values perfectly match the theoretical predictions (Eq. 6, Eq. 9). **Center and Right:** Logistic regressions on the IMDB dataset (center) and the StackOverflow dataset (right), confirm the same pattern: the utility of `DP-SGD` scales as $\frac{1}{\varepsilon^2}$, while the utility of `DP-SignSGD` scales linearly as $\frac{1}{\varepsilon}$. Across all settings, we observe that the insights obtained for `DP-SignSGD` extend to `DP-Adam` as well as to the test loss (see Figure C.7). For experimental details see Appendix C.2.

**Definition 3.3** *For $k \geq 0$, learning rate $\eta$, variance $\sigma_{DP}^2$, and batches $\gamma_k$ of size $B$ modeled as i.i.d. uniform random variables taking values in $\{1, \ldots, n\}$. Let $g_k$ be the private gradient, defined as*

$$g_k := \frac{1}{B} \sum_{i \in \gamma_k} \mathcal{C}[\nabla f_i(x_k)] + \frac{1}{B}\mathcal{N}(0, C^2\sigma_{DP}^2 I_d) \tag{2}$$

*and $\mathcal{C}[\cdot]$ be the clipping function $\mathcal{C}[x] = \min\left\{\frac{C}{\|x\|_2}, 1\right\} x$.*

*The iterates of `DP-SGD` are defined as*

$$x_{k+1} = x_k - \eta g_k, \tag{3}$$

*while those of `DP-SignSGD` are defined as*

$$x_{k+1} = x_k - \eta \operatorname{sign}[g_k], \tag{4}$$

*where $\operatorname{sign}[\cdot]$ is applied component-wise. Finally, those of DP-Adam are defined in Eq. 196 (Appendix C).*

The following theorem from (Abadi et al., 2016) gives the conditions under which `DP-SGD`, and thus also `DP-SignSGD`, is a differentially-private algorithm.

**Theorem 3.1** *For $q = \frac{B}{n}$ where $B$ is the batch size, $n$ is the number of training points, and number of iterations $T$, $\exists c_1, c_2$ s.t. $\forall \varepsilon < c_1 q^2 T$, if the noise multiplier $\sigma_{DP}$ satisfies $\sigma_{DP} \geq c_2 \frac{q\sqrt{T\log(1/\delta)}}{\varepsilon}$, `DP-SGD` is $(\varepsilon, \delta)$-differentially private for any $\delta > 0$.*

In the following, we will often use $\sigma_{DP} = \frac{\sqrt{T}\Phi}{\varepsilon}$, where $\Phi := q\sqrt{\log(1/\delta)}$ to indicate the DP noise multiplier.

## 4 THEORETICAL RESULTS

In this section, we investigate how the privacy budget $\varepsilon$ influences convergence speed and shapes the *privacy-utility trade-offs* in both the loss and the gradient norm. To do so, we leverage SDE models for `DP-SGD` and `DP-SignSGD`, which can be found in Theorem B.5 and Theorem B.10, respectively, and are experimentally validated in Figure C.1. In addition, we provide the first stationary distributions for these optimizers, presented in Theorem B.9 and Theorem B.15 in the Appendix. This section is organized as follows:

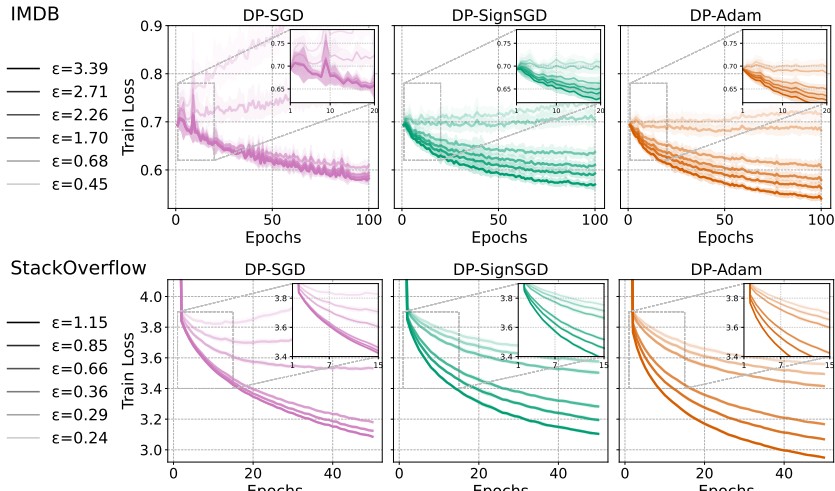

Figure 2: Empirical validation of the convergence speeds predicted by Thm. 4.1 and Thm. 4.3. We compare `DP-SGD`, `DP-SignSGD`, and `DP-Adam` as we train a logistic regression on the IMDB dataset (**Top Row**) and on the StackOverflow dataset (**Bottom Row**). In both tasks, we verify that when `DP-SGD` converges, its speed is unaffected by $\varepsilon$. As expected, it diverges when $\varepsilon$ is too small. Regarding `DP-SignSGD` and `DP-Adam`, they are faster when $\varepsilon$ is large and never diverge even when this is small. Crucially, Figure C.8 shows that these insights are also verified on the test loss. For experimental details see Appendix C.3.

1. **Protocol A (Section 4.1).** Section 4.1.1 analyzes `DP-SGD`, yielding bounds for the loss (Thm. 4.1) and the gradient norm (Thm. 4.2) in the $\mu$-PL and $L$-smooth cases, respectively: We observe that the convergence speed is *independent* of $\varepsilon$, while the privacy-utility trade-off scales as $\mathcal{O}(1/\varepsilon^2)$. Section 4.1.2 analyzes `DP-SignSGD`, and Thm. 4.3 and Thm. 4.4 show a qualitatively different behavior: Convergence speed scales linearly with $\varepsilon$, while the privacy-utility terms scale as $\mathcal{O}(1/\varepsilon)$, making adaptivity preferable if the privacy budget is small enough. Finally, Theorem 4.5 in Section 4.1.3 shows that when batch noise is large enough, `DP-SignSGD` always dominates. When batch noise is small, the outcome depends on the privacy budget: There exists $\varepsilon^\star$ such that for strict privacy ($\varepsilon < \varepsilon^\star$), `DP-SignSGD` is preferable, while for looser privacy ($\varepsilon > \varepsilon^\star$), `DP-SGD` is better.

2. **Protocol B (Section 4.2).** In this section, we derive the optimal learning rates of `DP-SGD` and `DP-SignSGD`: That of `DP-SGD` scales linearly in $\varepsilon$, while that of `DP-SignSGD` is independent of it. Under these parameter choices, they achieve the same asymptotic neighbourhoods.

We empirically validate our theoretical insights on real datasets[1]. Crucially, the same insights derived from `DP-SignSGD` *empirically* extend to `DP-Adam` as well as to test metrics: This underscores the mildness of our assumptions and the depth of our analysis.

**Notation.** In the following, we use the symbol $\lesssim$ to suppress absolute numerical constants (e.g., $2$, $\pi$, etc.), and never problem-dependent quantities such as $d$, $\mu$, $L$, or $\varepsilon$: This convention lightens the presentation. We will often use $\varepsilon$ to highlight the privacy budget in relevant formulas. We use "high privacy" to mean a small privacy budget, i.e., $\varepsilon \downarrow 0$. Whenever we state an $\varepsilon$-scaling such as $\mathcal{O}(1/\varepsilon)$, we refer to the leading $\varepsilon$-dependence as $\varepsilon \downarrow 0$ with all other quantities fixed under the protocol being discussed.

## 4.1 PROTOCOL A: FIXED HYPERPARAMETERS

Following the tuning routine of Li et al. (2023), we conduct an extensive grid search to select optimal hyperparameters $(\eta, C, B, \dots)$ for a chosen $\varepsilon$. To isolate the effect of $\varepsilon$, we then keep the optimal hyperparameters fixed as we vary $\varepsilon$. In particular, $\eta$ does *not* depend on $\varepsilon$ or on other hyperparameters. This absolute comparison exposes structural differences in how DP noise interacts with adaptive vs non-adaptive updates.

---

[1] For all our experiments, we use the official GitHub repository https://github.com/kenziyuliu/DP2 released with the Google paper Li et al. (2023).

### 4.1.1 DP-SGD: THE *Privacy-Utility Trade-Off* IS $\mathcal{O}\left(1/\varepsilon^2\right)$

By construction, a mini-batch can contain both *clipped* and *unclipped* per-example gradients. For exposition, we first study two idealized regimes: **Phase 1**, where all examples are clipped, and **Phase 2**, where none are clipped. This separation is purely pedagogical and allows us to isolate how the privacy budget $\varepsilon$ enters the dynamics. The fully realistic mixed setting, where clipped and unclipped examples coexist within the same batch, is handled by Theorem B.8.

**Theorem 4.1** *Let $f$ be $\mu$-PL and $L$-smooth, then during*

• *Phase 1: when each gradient is clipped, the loss satisfies:*

$$\mathbb{E}[f(X_t)] \lesssim f(X_0) \underbrace{e^{-\frac{\mu C}{\sigma_\gamma \sqrt{d}}t}}_{Decay} + \left(1 - e^{-\frac{\mu C}{\sigma_\gamma \sqrt{d}}t}\right) \underbrace{\frac{T\eta d^{\frac{3}{2}}LC\sigma_\gamma}{\mu}\left(\frac{\varepsilon^2}{BdT} + \frac{\Phi^2}{B^2}\right)\frac{1}{\varepsilon^2}}_{Privacy\text{-}Utility\ Trade\text{-}off}; \tag{5}$$

• *Phase 2: when no gradient is clipped, the loss satisfies:*

$$\mathbb{E}[f(X_t)] \lesssim f(X_0)e^{-\mu t} + \left(1 - e^{-\mu t}\right)\frac{T\eta dL}{\mu}\left(\frac{\varepsilon^2 \sigma_\gamma{}^2}{BT} + C^2\frac{\Phi^2}{B^2}\right)\frac{1}{\varepsilon^2}. \tag{6}$$

The decay rates are independent of $\varepsilon$: in Phase 2 they depend only on $\mu$, while in Phase 1 normalization spreads the signal over the sphere of radius $C$ (Vershynin, 2018, Ch. 3), giving a rate proportional to $C/(\sigma_\gamma \sqrt{d})$. In both phases, the *privacy-utility trade-off* term scales as $1/\varepsilon^2$ in the *high-privacy* regime.

We now turn to analyzing SDE dynamics assuming only $L$-smoothness of $f$. The following theorem presents a bound on the expected gradient norm *across* both phases **together**: We observe that the expected gradient norm admits the same $\mathcal{O}\left(1/\varepsilon^2\right)$ scaling in the *high-privacy* regime.

**Theorem 4.2** *Let $f$ be $L$-smooth, $K_1 := \max\{1, \frac{\sigma_\gamma \sqrt{d}}{C}\}$, and $K_2 := \max\{\frac{\sigma_\gamma^2}{B}, \frac{C^2}{Bd}\}$. Then,*

$$\mathbb{E}\left[\|\nabla f(X_{\tilde{t}})\|_2^2\right] \lesssim K_1\left(\frac{f(X_0)}{\eta T} + \eta dL\left(K_2 + \frac{(C\Phi)^2 T}{B^2 \varepsilon^2}\right)\right), \tag{7}$$

*where $\tilde{t}$ is a random time with uniform distribution on $[0, \tau]$.*

**Takeaway.** Thm 4.1 separates two effects: the *decay* terms, which encode the convergence speed, and the *privacy-utility* terms, which determine the asymptotic neighbourhood under DP. It shows that the convergence speed of DP-SGD is unaffected by the privacy budget $\varepsilon$: Fig. 2 empirically shows that, whenever DP-SGD does not diverge, its convergence speed is independent of $\varepsilon$. Additionally, when $\varepsilon \to 0$, the *privacy-utility trade-off* scales as $\mathcal{O}\left(1/\varepsilon^2\right)$, consistent with Thm 4.2. This insight is validated in Fig. 1: on a quadratic function (left panel), the empirical loss matches the theoretical values from Thm 4.1, and the same scaling is reproduced when training classifiers on IMDB and StackOverflow (center and right panels). The behavior persists on the test loss (Fig. C.7).

### 4.1.2 DP-SignSGD: THE *Privacy-Utility Trade-Off* IS $\mathcal{O}\left(1/\varepsilon\right)$

As for DP-SGD, we isolate the effect of $\varepsilon$ on the dynamics of DP-SignSGD and study the loss in each phase **separately**. Theorem B.14 covers the general case.

**Theorem 4.3** *Let $f$ be $\mu$-PL and $L$-smooth. Then, during*

• *Phase 1: when the gradient is clipped, the loss satisfies:*

$$\mathbb{E}[f(X_t)] \lesssim f(X_0) \underbrace{e^{\frac{-\mu B}{\sigma_\gamma \sqrt{dT}}\frac{\varepsilon}{\Phi}t}}_{Decay} + \left(1 - e^{\frac{-\mu B}{\sigma_\gamma \sqrt{dT}}\frac{\varepsilon}{\Phi}t}\right) \underbrace{\frac{\sqrt{T}\eta Ld^{\frac{3}{2}}\sigma_\gamma}{\mu B}\frac{\Phi}{\varepsilon}}_{Privacy\text{-}Utility\ Trade\text{-}off}; \tag{8}$$

• *Phase 2, i.e., when no gradient is clipped, the loss satisfies:*

$$\mathbb{E}[f(X_t)] \lesssim f(X_0) \underbrace{e^{\frac{-\mu \varepsilon t}{\sqrt{\varepsilon^2 \frac{\sigma_\gamma^2}{B} + \frac{C^2 \Phi^2}{B^2}}T}}}_{Decay} + \left(1 - e^{\frac{-\mu \varepsilon t}{\sqrt{\varepsilon^2 \frac{\sigma_\gamma^2}{B} + \frac{C^2 \Phi^2}{B^2}}T}}\right) \underbrace{\frac{\sqrt{T}\eta Ld}{\mu}\sqrt{\frac{\varepsilon^2 \sigma_\gamma^2}{BT} + \frac{C^2 \Phi^2}{B^2}}\frac{1}{\varepsilon}}_{Privacy\text{-}Utility\ Trade\text{-}off}. \tag{9}$$

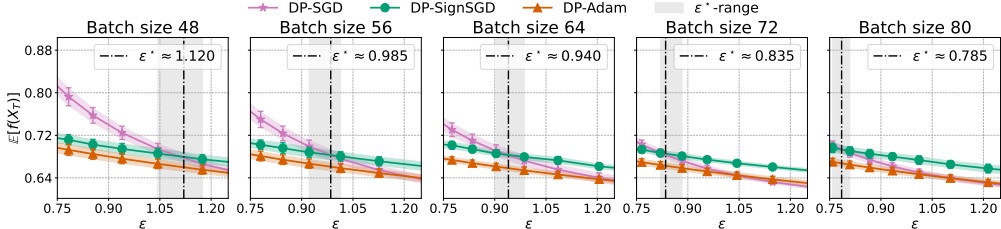

Figure 3: Logistic regression on IMDB Dataset: From left to right, we decrease the batch noise, i.e., increase the batch size, taking values $B \in \{48, 56, 64, 72, 80\}$: As per Theorem 4.5, the privacy threshold $\varepsilon^\star$ that determines when `DP-SignSGD` is more advantageous than `DP-SGD` shifts to the left. This confirms that if there is more noise due to the batch size, less privacy noise is needed for `DP-SignSGD` to be preferable over `DP-SGD`. For experimental details see Appendix C.4.

In the *high-privacy* regime, the decay rate is proportional to $\varepsilon$ in both phases (Eq. 8 and Eq. 9), unlike `DP-SGD`, where it is independent of $\varepsilon$ (Eq. 5 and Eq. 6). At the same time, the *privacy-utility trade-off* term in both phases scales as $\mathcal{O}\left(1/\varepsilon\right)$. This *might* be more favorable than the $\mathcal{O}\left(1/\varepsilon^2\right)$ scaling of `DP-SGD` in *high-privacy* regimes.

For $L$-smooth functions, the following result presents a bound on the expected gradient norm *across* both phases **together**. As the bound scales as $\mathcal{O}\left(1/\varepsilon\right)$, it suggests that adaptivity *might* mitigate the effect of large privacy noise on performance. Intuitively, the $\text{sign}[\cdot]$ effectively clips the privatized gradient signal, capping the update magnitude and reducing sensitivity to noise corruption.

**Theorem 4.4** *Let $f$ be $L$-smooth and $K_3 := \max\left\{\sqrt{\frac{\sigma_\gamma^2 \varepsilon^2}{BT} + \frac{C^2 \Phi^2}{B^2}}, \frac{\sigma_\gamma \Phi}{B}\sqrt{d}\right\}$. Then,*

$$\mathbb{E}\left[\|\nabla f(X_{\tilde{t}})\|_2^2\right] \lesssim K_3 \left(\frac{f(X_0)}{\eta\sqrt{T}} + \eta dL\sqrt{T}\right) \frac{1}{\varepsilon}, \tag{10}$$

*where $\tilde{t}$ is a random time with uniform distribution on $[0, \tau]$.*

**Takeaway:** Theorem 4.3 suggests that the privacy noise directly enters the convergence dynamics of `DP-SignSGD`, making its behavior qualitatively different from `DP-SGD`: The center column of Figure 2 confirms that `DP-SignSGD` converges faster for larger $\varepsilon$. Additionally, it also shows that it never diverges as drastically as `DP-SGD` for small $\varepsilon$. This is better shown in Figure 1, where we validate that the asymptotic loss scales with $\frac{1}{\varepsilon}$, while that of `DP-SGD` scales with $\frac{1}{\varepsilon^2}$. Therefore, adaptive methods are preferable in high-privacy settings, and all these insights are verified also for `DP-Adam` and generalize to the test loss (Figure C.7).

### 4.1.3 WHEN ADAPTIVITY REALLY MATTERS UNDER FIXED HYPERPARAMETERS.

In this subsection, we quantify when an adaptive method such as `DP-SignSGD` achieves better utility than `DP-SGD`. To this end, we compare *Privacy-Utility* terms of Phase 2 for both methods and derive conditions on the two sources of noise that govern the dynamics: the batch noise size $\sigma_\gamma$ and the privacy budget $\varepsilon$.

**Theorem 4.5** *If $\sigma_\gamma^2 \geq B$, then `DP-SignSGD` always achieves a better privacy-utility trade-off than `DP-SGD`. If $\sigma_\gamma^2 < B$, there exists a critical privacy level $\varepsilon^\star = \sqrt{\frac{C^2 TB}{n^2\left(B - \sigma_\gamma^2\right)} \log\left(\frac{1}{\delta}\right)}$ such that `DP-SignSGD` outperforms `DP-SGD` in utility whenever $\varepsilon < \varepsilon^\star$.*

**Takeaway:** This result makes the comparison explicit: $i$) Under **large batch noise** ($\sigma_\gamma^2 \geq B$), `DP-SignSGD` achieves a better utility than `DP-SGD`; $ii$) Under **small batch noise** ($\sigma_\gamma^2 < B$), the best optimizer depends on the privacy budget. For strict privacy ($\varepsilon < \varepsilon^\star$), `DP-SignSGD` has better utility, while for looser privacy ($\varepsilon > \varepsilon^\star$), `DP-SGD` achieves better overall performance. Thus, $\varepsilon^\star$ marks the threshold at which the advantage shifts from adaptive to non-adaptive methods when batch noise is small. By contrast, when batch noise is large, adaptive methods are already known to be more effective (Monzio Compagnoni et al., 2025b;c), and the effect of DP noise is only marginal relative to the intrinsic stochasticity of the gradients. We verify this result empirically in Figure 3: As we increase the batch size $B$, $\varepsilon^\star$ decreases, in accordance with our theoretical prediction.

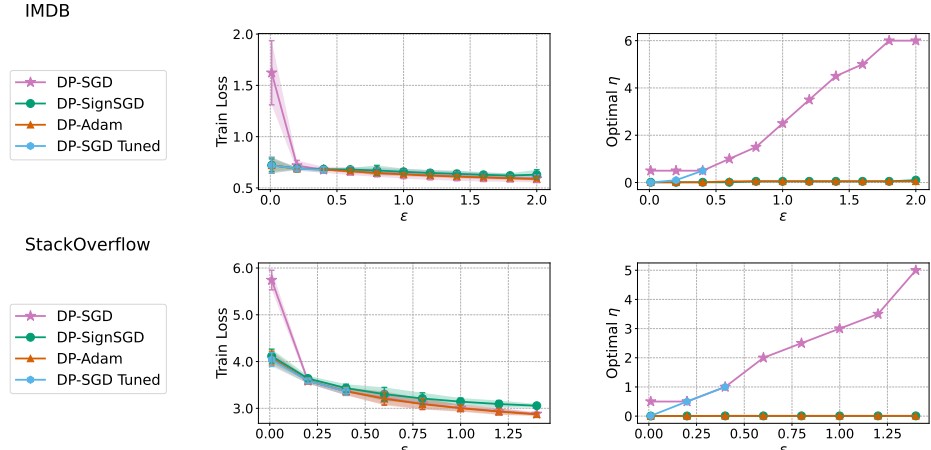

Figure 4: Empirical verification of Thm. 4.6 and Thm. 4.7 under Protocol B on the IMDB dataset **(Top Row)** and on the StackOverflow dataset **(Bottom Row)**. We tune $(\eta, C)$ of each optimizer for each $\varepsilon$ and confirm that: $i)$ all methods achieve comparable performance across privacy budgets; $ii)$ the optimal $\eta$ of DP-SGD scales linearly with $\varepsilon$, while that of adaptive methods is essentially $\varepsilon$-independent; $iii)$ failing to sweep over the "best" range of learning rates causes DP-SGD to severely underperform, whereas adaptive methods are resilient. On the **left**, DP-SGD degrades sharply for small $\varepsilon$. Indeed, the **right** panels shows that the selected optimal $\eta$ flattens out, while the theoretical one would have linearly decayed more: The "best" $\eta$ was simply missing from the grid. *A posteriori*, re-running the sweep with a larger grid (DP-SGD Tuned) recovers the scaling law and matches the performance of adaptive methods. For experimental details see Appendix C.5.

**Practical Implication.** If hyperparameter re-tuning is infeasible and the target regime involves stronger privacy constraints, e.g., lower privacy budget $\varepsilon$, or high stochasticity from small batches, adaptive methods are preferable. Otherwise, DP-SGD is the method of choice.

## 4.2 PROTOCOL B: BEST-TUNED HYPERPARAMETERS

We now mirror standard practice by allowing $(\eta, C)$ to be *tuned* over an extensive grid search for each target privacy budget $\varepsilon$. In contrast to Protocol A, this leads us to derive the theoretical optimal learning rates, which, just as in empirical tuning, are allowed to depend on $\varepsilon$ explicitly.

To select the optimal learning rate $\eta^\star$ for DP-SGD, we minimize the bound in Thm. 4.2 and consequently derive the implied optimal privacy-utility trade-off for DP-SGD in the $L$-smooth case.

**Theorem 4.6 (DP-SGD)** *Let* $\eta^\star = \min\left\{\sqrt{\frac{f(X_0)}{dLT\sigma_\gamma^2}}, \sqrt{\frac{f(X_0)}{dL}}\frac{\varepsilon n}{CT}\right\}$, *then the expected gradient norm bound of* DP-SGD *is* $\widetilde{\mathcal{O}}\left(\frac{C\sqrt{dLf(X_0)}}{\varepsilon n}\right)$, *as we ignore logarithmic terms and those decaying in* $T$.

This result aligns with the best-known privacy-utility trade-off obtained in prior works in these settings (Koloskova et al., 2023; Bassily et al., 2014). Importantly, we notice that the optimal learning rate of DP-SGD scales linearly in $\varepsilon$ and the resulting asymptotic performance scales like $1/\varepsilon$.

To derive the optimal learning rate $\eta^\star$ of DP-SignSGD, we minimize the bound in Theorem 4.4, and derive a privacy-utility trade-off in the $L$-smooth case.

**Theorem 4.7 (DP-SignSGD)** *Let* $\eta^\star = \sqrt{\frac{f(X_0)}{dLT}}$. *The expected asymptotic gradient norm bound of* DP-SignSGD *is* $\widetilde{\mathcal{O}}\left(\frac{C\sqrt{dLf(X_0)}}{\varepsilon n}\right)$, *as we ignore logarithmic terms and those decaying in* $T$.

Importantly, we observe that the asymptotic neighborhood of DP-SignSGD matches that of DP-SGD, while the optimal learning rate is independent of $\varepsilon$. This suggests that adaptivity automatically handles the privacy noise injection: This facilitates the transferability of optimal parameters to setups that require higher privacy. In contrast, DP-SGD needs retuning of the hyperparameters.

**Takeaway:** Our theory shows that while optimal learning rate scalings differ, the induced neighborhoods match. As shown in Fig.4, our experiments verify that: $i$) `DP-SGD`, `DP-SignSGD`, and `DP-Adam` exhibit similar asymptotic performance across a broad range of $\varepsilon$, even small values; $ii$) the optimal learning rate of `DP-SGD` is linear in $\varepsilon$, while that of adaptive methods is almost independent of it.

**Practical implication.** Hyperparameter searches are not free under DP: each evaluation consumes a portion of the privacy budget (Papernot & Steinke, 2022), making fine learning-rate grids costly. This asymmetrically impacts the two optimizers. For `DP-SGD`, the optimal step size scales linearly with $\varepsilon$ (Thm. 4.6), so the "right" $\eta^\star$ *moves* as privacy tightens. If a fixed sweep grid misses a value close to $\eta^\star$, the performance of `DP-SGD` can degrade sharply. This is illustrated in our experiments (Fig. 4): in the left panel, the performance of `DP-SGD` drops because the selected "optimal" $\eta$ plateaus instead of decaying linearly as predicted (right panel) — the true $\eta^\star$ was simply absent from the grid. By contrast, the optimal step size of `DP-SignSGD` (and empirically `DP-Adam`) is almost $\varepsilon$-invariant (Thm. 4.7), so a single well-chosen $\eta$ transfers across privacy levels with little or no re-tuning. This helps explain prior empirical reports that non-adaptive methods deteriorate more severely under stricter privacy (Zhou et al., 2020, Fig. 1), (Li et al., 2023, Fig. 5), (Asi et al., 2021, Fig. 2): a plausible cause is that their fixed grids did not track the $\varepsilon$-dependent $\eta^\star$ for `DP-SGD`. However, when both optimizers are carefully tuned, `DP-SGD` and `DP-Adam` match in performance in large-scale LLM fine-tuning (Li et al., 2022b, App. S).

## 5 CONCLUSION

We studied how differential privacy noise interacts with adaptive compared to non-adaptive optimization through the lens of SDEs: To our knowledge, this is the first SDE-based analysis of DP optimizers. Our results include explicit upper bounds on the expected loss and gradient norm, optimal learning rates, as well as the first characterization of stationary distributions for DP optimizers.

Under a *fixed-hyperparameter* scenario (Protocol A), the analysis reveals a sharp contrast: $i$) `DP-SGD` converges at a speed independent of the privacy budget $\varepsilon$ while incurring a $\mathcal{O}\left(1/\varepsilon^2\right)$ privacy-utility trade-off; $ii$) `DP-SignSGD` converges at a speed proportional to $\varepsilon$ while exhibiting a $\mathcal{O}\left(1/\varepsilon\right)$ privacy-utility trade-off. Additionally, when batch noise is large, adaptive methods dominate in terms of utility, as the effect of DP noise is marginal compared to the intrinsic stochasticity of the gradients, confirming known insights from non-private optimization. When batch noise is small, the preferable method depends on the privacy budget: for strict privacy, `DP-SignSGD` yields better utility, while for looser privacy, `DP-SGD` achieves better overall performance.

Under a *best-tuned* scenario (Protocol B), the picture changes: theory and experiments agree that the optimal learning rate of `DP-SGD` scales linearly with $\varepsilon$, whereas the optimal learning rate of `DP-SignSGD` (and empirically `DP-Adam`) is approximately $\varepsilon$-independent. With this tuning, the induced privacy-utility trade-offs match in order and the methods achieve comparable asymptotic performance, including at very small $\varepsilon$. A practical implication is that adaptive methods require less re-tuning if regulations mandate tighter privacy budgets.

We validated these theoretical insights on both synthetic and real datasets. Importantly, we also demonstrated that the qualitative behavior observed for `DP-SignSGD` extends empirically to `DP-Adam` and to test metrics, underscoring the strength and generality of our framework.

**Practitioner guidance.** Under higher privacy requirements, e.g., regulations mandate a smaller $\varepsilon$, if per-$\varepsilon$ re-tuning of the hyperparameters is impractical because retraining/tuning is expected to be costly (Protocol A), prefer an *adaptive* private optimizer such as `DP-SignSGD` (or `DP-Adam`): their performance scales more favorably as $\varepsilon$ decreases compared to `DP-SGD`.

When re-tuning is feasible (Protocol B): Both `DP-SGD` and adaptive methods can reach comparable asymptotic performance. However, hyperparameter searches are not free under DP: each sweep consumes additional privacy budget (Papernot & Steinke, 2022), making fine grids expensive. This creates an asymmetric risk: `DP-SGD` requires an $\varepsilon$-dependent learning rate ($\eta^\star \propto \varepsilon$), so if the sweep grid does not track this scaling, its performance can degrade sharply. In contrast, adaptive methods retain a portable, $\varepsilon$-independent learning rate, making them more robust and less costly to tune across privacy levels.

ACKNOWLEDGMENTS

Enea Monzio Compagnoni, Rustem Islamov, and Aurelien Lucchi acknowledge the financial support of the Swiss National Foundation, SNF grant No 207392. We acknowledge the use of OpenAI's ChatGPT as a writing assistant to help us rephrase and refine parts of the manuscript. All technical content, derivations, and scientific contributions remain the sole responsibility of the authors.

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

# Appendix

## CONTENTS

## A TECHNICAL RESULTS

In this section, we introduce some technical results used in the derivation of the SDEs.

**Lemma A.1** *Let $X \sim \mathcal{N}(\mu, \sigma^2 I_d)$ and fix a tolerance $\epsilon > 0$. If $\frac{\|\mu\|_2^2}{2\sigma^2(d+2)} < \epsilon$, for $d \to \infty$, we have that $\mathbb{E}\left(\frac{X}{\|X\|_2}\right) = \sqrt{\frac{1}{d}}\frac{\mu}{\sigma} + \varepsilon\mathcal{O}\left(\frac{1}{d^{3/2}}\right)$.*

**Proof:** Let us remember that if $X \sim \mathcal{N}(\mu, \sigma^2 I_d)$,

$$\mathbb{E}\left(\frac{X}{\|X\|_2^k}\right) = \frac{\Gamma\left(\frac{d}{2} + 1 - \frac{k}{2}\right)}{(2\sigma^2)^{k/2}\Gamma\left(\frac{d}{2} + 1\right)} {}_1F_1\left(\frac{k}{2}; \frac{d+2}{2}; -\frac{\|\mu\|_2^2}{2\sigma^2}\right)\mu, \tag{11}$$

where ${}_1F_1(a; b; z)$ is Kummer's confluent hypergeometric function. We know that

$$\lim_{d \to \infty} \frac{\Gamma\left(\frac{d}{2} + 1 - \frac{1}{2}\right)}{\Gamma\left(\frac{d}{2} + 1\right)} \overset{k=1}{\sim} \sqrt{\frac{2}{d}} + \mathcal{O}\left(\frac{1}{d^{3/2}}\right). \tag{12}$$

Let $z = \frac{\|\mu\|_2^2}{2\sigma^2}$. If $d > z$, by expanding the series, we have

$$ {}_1F_1\left(\frac{1}{2}; \frac{d}{2} + 1; -z\right) = \sum_{n \geq 0} \frac{a^{(n)}(-z)^n}{b^{(n)}n!} = 1 - \frac{z}{d+2} + \mathcal{O}\left(\frac{z}{d}\right)^2 < 1 - \epsilon. \tag{13}$$

Combining everything together, we obtain $\mathbb{E}\left(\frac{X}{\|X\|_2}\right) = \sqrt{\frac{1}{d}}\frac{\mu}{\sigma} + \epsilon\mathcal{O}\left(\frac{1}{d^{3/2}}\right)$. $\qquad\square$

**Lemma A.2** *Let $K(\nu) = \sqrt{\frac{2}{\nu}}\frac{\Gamma\left(\frac{\nu+1}{2}\right)}{\Gamma\left(\frac{\nu}{2}\right)}$ and $X \sim t_\nu(\mu, \sigma^2 I_d)$, for $\nu \geq 1$. Fix a tolerance $\epsilon > 0$: If $\frac{\|\mu\|_2^2}{2\sigma^2(d+2)} < \epsilon$, for $d \to \infty$, we have that $\mathbb{E}\left(\frac{X}{\|X\|_2}\right) = K(\nu)\sqrt{\frac{1}{d}}\frac{\mu}{\sigma} + \epsilon\mathcal{O}\left(\frac{1}{d^{3/2}}\right)$.*

**Proof:** One can write $X = \mu + \frac{\sigma Z}{\sqrt{S/\nu}}$, where $Z \sim \mathcal{N}(0, I_d)$ and $S \sim \chi^2_\nu$ are independent. Define $\tau = \frac{\sigma}{\sqrt{S/\nu}}$, then, conditioning on $S$ and applying Lemma A.1, we have

$$\mathbb{E}\left[\frac{X}{\|X\|_2}\bigg|S\right] = \sqrt{\frac{1}{d}}\frac{\mu}{\tau} + \epsilon\mathcal{O}\left(d^{-3/2}\right). \tag{14}$$

Remembering that $\mathbb{E}[\sqrt{S}] = \sqrt{2}\frac{\Gamma\left(\frac{\nu+1}{2}\right)}{\Gamma\left(\frac{\nu}{2}\right)}$, we have

$$\mathbb{E}\left[\frac{X}{\|X\|_2}\right] = \mathbb{E}_S\left[\left(\sqrt{\frac{1}{d}}\frac{\mu}{\tau} + \epsilon\mathcal{O}\left(d^{-3/2}\right)\right)\right] \tag{15}$$

$$= \left(\sqrt{\frac{1}{d}}\frac{\mu}{\sigma\sqrt{\nu}}\mathbb{E}_S[\sqrt{S}] + \epsilon\mathcal{O}\left(d^{-3/2}\right)\right) \tag{16}$$

$$= K(\nu)\sqrt{\frac{1}{d}}\frac{\mu}{\sigma} + \epsilon\mathcal{O}\left(\frac{1}{d^{3/2}}\right). \tag{17}$$

$\square$

**Remark A.1** *As discussed in the main paper, our analysis is based on the following two assumptions:*

*i) The number of trainable parameters is large, specifically $d = \Omega(10^4)$;*

*ii) The signal-to-noise ratio satisfies $\frac{\|\nabla f(x)\|_2^2}{2\sigma_\gamma^2} \ll d$.*

*First, they ensure that the approximation of the confluent hypergeometric function in Equation 13 is highly accurate. Second, neither condition is restrictive for modern deep learning models. The dimensionality assumption is trivially satisfied by contemporary architectures, which routinely have millions of parameters. Regarding the second assumption, Malladi et al. (2022) empirically measured the signal-to-noise ratio $\frac{\|\nabla f(x)\|_2^2}{2\sigma_\gamma^2}$ across a wide range of large-scale architectures and datasets, and consistently found values of at most $O(10^2)$. Thus, the regime in which our approximation is valid closely matches the regime observed in practice. This is further supported by our experimental results, which confirm our theoretical predictions across all models and tasks considered in this paper. In Figure A.1, we numerically evaluate the confluent hypergeometric function for varying values of $d$, and show that, for sufficiently large parameter counts, the approximation remains tight throughout the realistic signal-to-noise ratio range reported in (Malladi et al., 2022).*

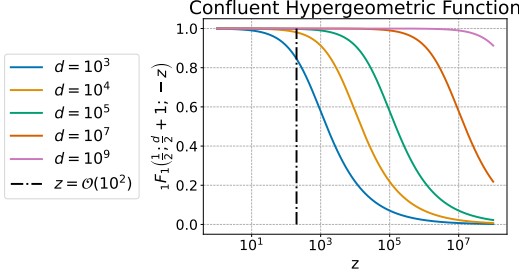

Figure A.1: Numerical validation of the approximation used in Equation 13. For several values of $d$, we plot the confluent hypergeometric function as a function of the signal-to-noise ratio $z$. In the realistic range observed in (Malladi et al., 2022), approximating this function by $1$ is extremely accurate.

# B   THEORETICAL FRAMEWORK

In this section, we introduce the theoretical framework, assumptions, and notations used to formally derive the SDE models used in this paper. We briefly recall the definition of $L$-smooth and $\mu$-PL functions. Then we introduce the set of functions of polynomial growth $G$.

**Definition B.1** *A function $f : \mathbb{R}^d \to \mathbb{R}$ is L-smooth if it is differentiable and its gradient is L-Lipschitz continuous, namely*

$$\|\nabla f(x) - \nabla f(y)\|_2 \leq L\|x - y\|_2 \; \forall x, y. \tag{18}$$

**Definition B.2** *A function $f : \mathbb{R}^d \to \mathbb{R}$ admitting a global minimum $x^*$ satisfies the Polyak-Łojasiewicz inequality if, for some $\mu > 0$ and for all $x \in \mathbb{R}^d$, it holds*

$$f(x) - f^* \leq \frac{1}{2\mu}\|\nabla f(x)\|_2^2. \tag{19}$$

*In this case, we say that the function $f$ is $\mu$-PL.*

**Definition B.3** *Let $G$ denote the set of continuous functions $g : \mathbb{R}^d \to \mathbb{R}$ of at most polynomial growth, namely such that there exist positive integers $k_1, k_2 > 0$ such that $|g(x)| < k_1(1 + \|x\|_2^2)^{k_2}$, for all $x \in \mathbb{R}^d$.*

To simplify the notation, we will write

$$b(x + \eta) = b_0(x) + \eta b_1(x) + \mathcal{O}(\eta^2),$$

whenever there exists $g \in G$, independent of $\eta$, such that

$$|b(x + \eta) - b_0(x) - \eta b_1(x)| \leq g(x)\eta^2.$$

We now introduce the definition of weak approximation, which formalizes in which sense the solution to an SDE, a continuous-time random process, models a discrete-time optimizer.

**Definition B.4** *Let $0 < \eta < 1$, $\tau > 0$ and $T = \lfloor \frac{\tau}{\eta} \rfloor$. We say that a continuous time process $X_t$ over $[0, \tau]$, is an order $\alpha$ weak approximation of a discrete process $x_k$, for $k = 0, \dots, N$, if for every $g \in G$, there exists $M$, independent of $\eta$, such that for all $k = 0, 1, \dots, T$*

$$|\mathbb{E}g(X_{k\eta}) - \mathbb{E}g(x_k)| \leq M\eta^\alpha.$$

This framework focuses on approximation in a *weak sense*, meaning in distribution rather than pathwise. Since $G$ contains all polynomials, all the moments of both processes become closer at a rate of $\eta^\alpha$ and thus their distributions. Thus, while the processes exhibit similar average behavior, their sample paths may differ significantly, justifying the term weak approximation.

**Remark B.1 (Validity of the SDE approximation)** *To guarantee that the SDE model is a first-order weak approximation of the optimizer dynamics in the sense of Definition 3.1, one shows that the first two moments of the one-step increments of the optimizer and of the SDE match up to $O(\eta^2)$, while all higher-order terms in the Taylor expansion are collected in an $O(\eta^2)$ remainder (see Appendix B). This implies that the discrepancy between the two processes scales as $O(\eta)$ for any test function of polynomial growth and any finite time horizon. The neglected $O(\eta^2)$ terms could, in principle, be retained by deriving higher-order SDEs. However, to the best of our knowledge, such second-order models have been derived (Li et al., 2017), but have not yet led to additional practical insight in the analysis of optimisation algorithms until recently Monzio Compagnoni et al. (2025a). Finally, Figure C.1 empirically compares the discrete algorithms with their SDE counterparts on quadratic and quartic objectives, confirming that for the step sizes used in our experiments, the first-order SDEs closely track the discrete dynamics.*

The key ingredient for deriving the SDE is given by the following result (see Theorem 1, (Li et al., 2017)), which provides sufficient conditions to get a weak approximation in terms of the single step increments of both $X_t$ and $x_k$. Before stating the theorem, we list the regularity assumption under which we are working.

**Assumption B.1** *Assume that the following conditions are satisfied:*

- $f, f_i \in \mathcal{C}_b^8(\mathbb{R}^d, \mathbb{R})$;

- $f, f_i$ *and its partial derivatives up to order 7 belong to $G$;*

- $\nabla f, \nabla f_i$ *satisfy the following Lipschitz condition: there exists $L > 0$ such that*

$$\|\nabla f(u) - \nabla f(v)\|_2 + \sum_{i=1}^d \|\nabla f_i(u) - \nabla f_i(v)\|_2 \le L\|u - v\|_2;$$

- $\nabla f, \nabla f_i$ *satisfy the following growth condition: there exists $M > 0$ such that*

$$\|\nabla f(x)\|_2 + \sum_{i=1}^n \|\nabla f_i(x)\|_2 \le M(1 + \|x\|_2).$$

**Assumption B.2** *Assume the* per-example *($\gamma$ has size 1) stochastic gradient admits the decomposition $\nabla f_\gamma(x) = \nabla f(x) + Z_\gamma(x)$, with centered, approximately isotropic noise $Z_\gamma(x)$. In Phase 1 (clipping regime), we model the per-example noise as heavy-tailed, e.g., $Z_\gamma(x) \sim \sigma_\gamma\, t_\nu(0, I_d)$, a multivariate Student-t with $\nu$ degrees of freedom (for $\nu = \infty$ we recover the Gaussian case; for $\nu \le 2$ the variance is unbounded). In Phase 2 (non-clipping regime), we consider the mini-batch average $\nabla f_\gamma(x) := \frac{1}{B}\sum_{\xi \in \gamma} \nabla f_\xi(x)$ and approximate its noise by $Z_\gamma(x) := \nabla f_\gamma(x) - \nabla f(x) \sim \frac{\sigma_\gamma}{\sqrt{B}}\mathcal{N}(0, I_d)$, reflecting the averaging effect of i.i.d. per-sample gradients.*

**Remark B.2** *The distinction between the two phases stems from how per-example clipping alters the effective noise seen by the optimizer. In Phase 2, clipping is inactive and the mini-batch average of i.i.d. per-sample gradients admits the usual Gaussian approximation with variance $\sigma_\gamma^2/B$. In Phase 1, clipping is applied* before *the average and the update is driven by normalized/truncated per-example gradients. While averaging still reduces the variance of the* average of clipped directions *by a factor $1/B$, the distributional shape is governed by the per-example gradient law and can deviate substantially from Gaussian. We therefore model the per-example noise with a multivariate Student-t, which captures heavy tails and yields tractable expressions, while recovering the Gaussian case as $\nu \to \infty$.*

**Lemma B.3** *Let $0 < \eta < 1$. Consider a stochastic process $X_t, t \ge 0$ satisfying the SDE*

$$dX_t = b(X_t)dt + \sqrt{\eta}\sigma(X_t)dW_t, \qquad X_0 = x \tag{20}$$

*where $b, \sigma$ together with their derivatives belong to $G$. Define the one-step difference $\Delta = X_\eta - x$, and indicate the $i$-th component of $\Delta$ with $\Delta_i$. Then we have*

1. $\mathbb{E}\Delta_i = b_i\eta + \frac{1}{2}\left[\sum_{j=1}^d b_j\partial_j b_i\right]\eta^2 + \mathcal{O}(\eta^3) \qquad \forall i = 1, \ldots, d;$

2. $\mathbb{E}\Delta_i\Delta_j = \left[b_i b_j + \sigma\sigma_{ij}^\top\right]\eta^2 + \mathcal{O}(\eta^3) \qquad \forall i, j = 1, \ldots, d;$

3. $\mathbb{E}\prod_{j=1}^s \Delta_{i_j} = \mathcal{O}(\eta^3) \qquad \forall s \ge 3,\ i_j = 1, \ldots, d.$

*All functions above are evaluated at $x$.*

**Theorem B.4** *Let $0 < \eta < 1$, $\tau > 0$ and set $T = \lfloor \tau/\eta \rfloor$. Let Assumption B.1 hold and let $X_t$ be a stochastic process as in Lemma B.3. Define $\bar\Delta = x_1 - x$ to be the increment of the discrete-time algorithm, and indicate the $i$-th component of $\bar\Delta$ with $\bar\Delta_i$. If in addition there exist $K_1, K_2, K_3, K_4 \in G$ so that*

1. $\left|\mathbb{E}\Delta_i - \mathbb{E}\bar\Delta_i\right| \le K_1(x)\eta^2, \qquad \forall i = 1, \ldots, d;$

2. $\left|\mathbb{E}\Delta_i\Delta_j - \mathbb{E}\bar\Delta_i\bar\Delta_j\right| \le K_2(x)\eta^2, \qquad \forall i, j = 1, \ldots, d;$

3. $\left|\mathbb{E}\prod_{j=1}^s \Delta_{i_j} - \mathbb{E}\prod_{j=1}^s \bar\Delta_{i_j}\right| \le K_3(x)\eta^2, \qquad \forall s \ge 3, \forall i_j = 1, \ldots, d;$

4. $\mathbb{E} \prod_{j=1}^{s} |\bar{\Delta}_{i_j}| \leq K_4(x)\eta^2, \qquad \forall i_j = 1, \ldots, d.$

*Then, there exists a constant $C$ so that for all $k = 0, 1, \ldots, N$ we have*

$$|\mathbb{E}g(X_{k\eta}) - \mathbb{E}g(x_k)| \leq C\eta. \tag{21}$$

*We say Eq. 20 is an order $1$ weak approximation of the update step of $x_k$.*

## B.1 DP-SGD

This subsection provides the formal derivation of the SDE model for DP-SGD (Theorem B.5), together with the formal versions of the loss and gradient-norm bounds used in the main paper (Theorems B.6 and B.7) and the quadratic stationary distribution (Theorem B.9).

**Idealized clipping regimes vs. the mixed regime.** Following the exposition in Section 4.1.1, we first analyze two idealized per-iteration regimes to make the dependence on the privacy budget $\varepsilon$ explicit: (i) *fully-clipped regime* ("Phase 1"), where all sampled per-example gradients exceed the clipping threshold and the update is dominated by normalized directions with magnitude on the order of $C$; and (ii) *unclipped regime* ("Phase 2"), where clipping is inactive and the update reduces to a standard noisy gradient step. These two regimes are used only to isolate how clipping changes the drift and diffusion terms. The fully realistic *mixed* case, where clipped and unclipped samples coexist within each mini-batch, is handled separately in Section B.1.1 (Theorem B.8).

**Theorem B.5** *Let $0 < \eta < 1$, $\tau > 0$ and set $T = \lfloor \tau/\eta \rfloor$ and $K(\nu) = \sqrt{\frac{2}{\nu}} \frac{\Gamma\left(\frac{\nu+1}{2}\right)}{\Gamma\left(\frac{\nu}{2}\right)}$. Let $x_k \in \mathbb{R}^d$ denote a sequence of DP-SGD iterations defined in Eq. 3. Assume Assumption B.1 and Assumption B.2. Let $X_t$ be the solution of the following SDEs with initial condition $X_0 = x_0$:*

• *Phase 1:*

$$dX_t = -\frac{CK(\nu)}{\sigma_\gamma \sqrt{d}} \nabla f(X_t)dt + \sqrt{\eta}\sqrt{\bar{\Sigma}(X_t)}dW_t, \tag{22}$$

*where $\bar{\Sigma}(x) = \frac{C^2}{B}\left(\mathbb{E}_\xi\left[\frac{\nabla f_\xi(x)\nabla f_\xi(x)^\top}{\|\nabla f_\xi(x)\|_2^2}\right] - \frac{K(\nu)^2}{\sigma_\gamma^2 d}\nabla f(x)\nabla f(x)^\top\right) + \frac{C^2 \sigma_{DP}^2}{B^2}I_d.$*

• *Phase 2:*

$$dX_t = -\nabla f(X_t)dt + \sqrt{\eta}\sqrt{\bar{\Sigma}(X_t)}dW_t, \tag{23}$$

*where $\bar{\Sigma}(x) = \left(\frac{\sigma_\gamma^2}{B} + \frac{C^2 \sigma_{DP}^2}{B^2}\right)I_d.$*

*Then, Eq. 22 and Eq. 23 are an order $1$ approximation of the discrete update of Phase 1 and Phase 2 of DP-SGD, respectively.*

**Proof:** • Phase 1: Let $Z_{DP} \sim \mathcal{N}\left(0, \frac{C^2 \sigma_{DP}^2}{B^2}I_d\right)$ be the differentially-private noise injected via the Gaussian mechanism and denote with $\bar{\Delta} = x_1 - x$ the one-step increment for Phase 1. In the fully-clipped regime, each sampled per-example gradient is clipped to norm $C$, hence

$$\bar{\Delta} = -\eta\left(\frac{C}{B}\sum_{\xi \in \gamma}\frac{\nabla f_\xi(x)}{\|\nabla f_\xi(x)\|_2} + Z_{DP}\right). \tag{24}$$

Applying Lemma A.2 with tolerance $\epsilon = \eta$ and by linearity of expectation we have

$$\mathbb{E}[\bar{\Delta}] = -\eta\,C\,\mathbb{E}_\xi\left[\frac{\nabla f_\xi(x)}{\|\nabla f_\xi(x)\|_2}\right] = -\eta\frac{CK(\nu)}{\sigma_\gamma\sqrt{d}}\nabla f(x) + \mathcal{O}(\eta^2). \tag{25}$$

To compute the covariance, define $Y_\xi(x) := \frac{\nabla f_\xi(x)}{\|\nabla f_\xi(x)\|_2}$ and $u_\gamma(x) := \frac{1}{B}\sum_{\xi \in \gamma}Y_\xi(x)$. Since the samples in $\gamma$ are i.i.d. conditional on $x$, we have $\text{Cov}(u_\gamma) = \frac{1}{B}\text{Cov}(Y_\xi)$, and $u_\gamma$ is independent of

$Z_{DP}$. Therefore

$$\text{Cov}(\bar{\Delta}) = \eta^2 \left( \text{Cov}(Cu_\gamma(x)) + \text{Cov}(Z_{DP}) \right) + \mathcal{O}(\eta^4) \tag{26}$$

$$= \eta^2 \left( \frac{C^2}{B} \text{Cov}(Y_\xi(x)) + \frac{C^2 \sigma_{DP}^2}{B^2} I_d \right) + \mathcal{O}(\eta^4) \tag{27}$$

$$= \eta^2 \left( \frac{C^2}{B} \mathbb{E}_\xi \left[ \frac{\nabla f_\xi(x) \nabla f_\xi(x)^\top}{\|\nabla f_\xi(x)\|_2^2} \right] - \frac{C^2 K(\nu)^2}{B \sigma_\gamma^2 d} \nabla f(x) \nabla f(x)^\top + \frac{C^2 \sigma_{DP}^2}{B^2} I_d \right) + \mathcal{O}(\eta^4). \tag{28}$$

Define now

$$b(x) := -\frac{C K(\nu)}{\sigma_\gamma \sqrt{d}} \nabla f(x) \tag{29}$$

$$\bar{\Sigma}(x) := \frac{C^2}{B} \left( \mathbb{E}_\xi \left[ \frac{\nabla f_\xi(x) \nabla f_\xi(x)^\top}{\|\nabla f_\xi(x)\|_2^2} \right] - \frac{K(\nu)^2}{\sigma_\gamma^2 d} \nabla f(x) \nabla f(x)^\top \right) + \frac{C^2 \sigma_{DP}^2}{B^2} I_d. \tag{30}$$

Then, from Lem B.3 and Thm. B.4 the claim follows.

• Phase 2: Following the same steps as above, one obtains:

$$\mathbb{E}[\bar{\Delta}] = -\eta \mathbb{E}_{\gamma, DP} \left[ \nabla f_\gamma(x) + Z_{DP} \right] = -\eta \nabla f(x), \tag{31}$$

and

$$\text{Cov}(\bar{\Delta}) = \eta^2 \mathbb{E} \left[ \left( \nabla f_\gamma(x) + Z_{DP} - \nabla f(x) \right) \left( \nabla f_\gamma(x) + Z_{DP} - \nabla f(x) \right)^\top \right] \tag{32}$$

$$= \eta^2 \mathbb{E} \left[ \left( \nabla f_\gamma(x) - \nabla f(x) \right) \left( \nabla f_\gamma(x) - \nabla f(x) \right)^\top \right] + \eta^2 \frac{C^2 \sigma_{DP}^2}{B^2} I_d \tag{33}$$

$$= \eta^2 \left( \frac{\sigma_\gamma^2}{B} + \frac{C^2 \sigma_{DP}^2}{B^2} \right) I_d. \tag{34}$$

Define

$$b(x) := -\nabla f(x); \tag{35}$$

$$\bar{\Sigma}(x) := \left( \frac{\sigma_\gamma^2}{B} + \frac{C^2 \sigma_{DP}^2}{B^2} \right) I_d; \tag{36}$$

Finally, from Lem B.3 and Thm. B.4 the claim follows. $\qquad \square$

**Theorem B.6** *Let $f$ be $L$-smooth and $\mu$-PL. Then, for $t \in [0, \tau]$, we have that*

• *Phase 1, i.e., when the gradient is clipped, the loss satisfies:*

$$\mathbb{E}[f(X_t)] \lesssim f(X_0) e^{-\frac{\mu C}{\sigma_\gamma \sqrt{d}} t} + \left( 1 - e^{-\frac{\mu C}{\sigma_\gamma \sqrt{d}} t} \right) \frac{T \eta d^{\frac{3}{2}} L C \sigma_\gamma}{\mu} \left( \frac{\varepsilon^2}{BdT} + \frac{\Phi^2}{B^2} \right) \frac{1}{\varepsilon^2}; \tag{37}$$

• *Phase 2, i.e., when the gradient is not clipped, the loss satisfies:*

$$\mathbb{E}[f(X_t)] \lesssim f(X_0) e^{-\mu t} + \left( 1 - e^{-\mu t} \right) \frac{T \eta d L}{\mu} \left( \frac{\varepsilon^2 \sigma_\gamma^2}{BT} + C^2 \frac{\Phi^2}{B^2} \right) \frac{1}{\varepsilon^2}. \tag{38}$$

**Proof:** • Phase 1: By construction we have

$$\text{Tr}\left( \bar{\Sigma}(x) \right) \leq \frac{C^2}{B} + d \frac{C^2 \sigma_{DP}^2}{B^2}. \tag{39}$$

Since $f$ is $\mu$-PL and $L$-smooth it follows that $2\mu f(x) \leq \|\nabla f(x)\|_2^2$ and $\nabla^2 f(x) \preceq L I_d$. Hence, by applying the Itô formula we have

$$df(X_t) = -\frac{C K(\nu)}{\sigma_\gamma \sqrt{d}} \|\nabla f(X_t)\|_2^2 dt + \frac{\eta}{2} \text{Tr}\left( \nabla^2 f(X_t) \bar{\Sigma}(X_t) \right) dt + \mathcal{O}(\text{Noise}) \tag{40}$$

$$\leq -2\mu \frac{C K(\nu)}{\sigma_\gamma \sqrt{d}} f(X_t) dt + \frac{\eta d L}{2} \left( \frac{C^2}{Bd} + \frac{C^2 \sigma_{DP}^2}{B^2} \right) dt + \mathcal{O}(\text{Noise}). \tag{41}$$

Therefore,

$$\mathbb{E}[f(X_t)] \le f(X_0)e^{-2\mu\frac{K(\nu)C}{\sigma_\gamma\sqrt{d}}t} + \left(1 - e^{-2\mu\frac{K(\nu)C}{\sigma_\gamma\sqrt{d}}t}\right)\frac{\eta d^{\frac{3}{2}}L\sigma_\gamma}{4\mu C K(\nu)}\left(\frac{C^2}{Bd} + \frac{C^2\sigma_{DP}^2}{B^2}\right). \tag{42}$$

Let us now remind that

$$\sigma_{DP} = \frac{q\sqrt{T\log(1/\delta)}}{\varepsilon}, \tag{43}$$

then

$$\mathbb{E}[f(X_t)] \le f(X_0)e^{-2\mu\frac{K(\nu)C}{\sigma_\gamma\sqrt{d}}t} + \left(1 - e^{-2\mu\frac{K(\nu)C}{\sigma_\gamma\sqrt{d}}t}\right)\frac{\eta d^{\frac{3}{2}}LC\sigma_\gamma}{4\mu K(\nu)}\left(\frac{1}{Bd} + \frac{Tq^2\log(1/\delta)}{B^2\varepsilon^2}\right). \tag{44}$$

- Phase 2: Similarly to Phase 1, we have

$$\mathrm{Tr}\left(\bar{\Sigma}(x)\right) = d\left(\frac{\sigma_\gamma^2}{B} + \frac{C^2\sigma_{DP}^2}{B^2}\right). \tag{45}$$

Again using the fact that $f$ is $\mu$-PL and $L$-smooth and by applying the Itô formula, one obtains

$$df(X_t) \le -\|\nabla f(X_t)\|_2^2 dt + \frac{\eta dL}{2}\left(\frac{\sigma_\gamma^2}{B} + \frac{C^2\sigma_{DP}^2}{B^2}\right) + \mathcal{O}(\text{Noise}), \tag{46}$$

from which we have

$$\mathbb{E}[f(X_t)] \le f(X_0)e^{-2\mu t} + \left(1 - e^{-2\mu t}\right)\frac{\eta dL}{4\mu}\left(\frac{\sigma_\gamma^2}{B} + \frac{C^2\sigma_{DP}^2}{B^2}\right). \tag{47}$$

Hence, by expanding $\sigma_{DP}$

$$\mathbb{E}[f(X_t)] \le f(X_0)e^{-2\mu t} + \left(1 - e^{-2\mu t}\right)\frac{\eta dL}{4\mu}\left(\frac{\sigma_\gamma^2}{B} + \frac{C^2q^2T\log(1/\delta)}{B^2\varepsilon^2}\right). \tag{48}$$

Finally, let $\Phi = q\sqrt{\log(1/\delta)}$ and suppress all problem-independent constants, such as $2, \pi, K(\nu)$, to obtain the claim.

$\square$

**Theorem B.7** *Let $f$ be L-smooth and define*

$$K_1 := \max\left\{1, \frac{\sigma_\gamma\sqrt{d}}{CK(\nu)}\right\} \quad and \quad K_2 := \max\left\{\frac{C^2}{Bd}, \frac{\sigma_\gamma^2}{B}\right\}. \tag{49}$$

*then*

$$\mathbb{E}\left[\|\nabla f(X_{\tilde{t}})\|_2^2\right] \lesssim K_1\left(\frac{f(X_0)}{\eta T} + \frac{\eta dL}{2}\left(K_2 + \frac{C^2\left(\frac{q}{B}\right)^2 T\log(1/\delta)}{\varepsilon^2}\right)\right), \tag{50}$$

*where $\tilde{t} \sim \mathrm{Unif}(0, \tau)$.*

**Proof:** Since $f$ is $L$-smooth and by applying the Itô formula to Phase 1 we have:

$$df(X_t) \le -\frac{CK(\nu)}{\sigma_\gamma\sqrt{d}}\|\nabla f(X_t)\|_2^2 dt + \frac{\eta}{2}\mathrm{Tr}\left(L\bar{\Sigma}(X_t)\right)dt + \mathcal{O}(\text{Noise}) \tag{51}$$

$$\le -\frac{CK(\nu)}{\sigma_\gamma\sqrt{d}}\|\nabla f(X_t)\|_2^2 dt + \frac{\eta dL}{2}\left(\frac{C^2}{Bd} + \frac{C^2\sigma_{DP}^2}{B^2}\right)dt + \mathcal{O}(\text{Noise}). \tag{52}$$

Similarly, in Phase 2 we obtain

$$df(X_t) \le -\|\nabla f(X_t)\|_2^2 dt + \frac{\eta}{2}\mathrm{Tr}\left(L\bar{\Sigma}(X_t)\right)dt + \mathcal{O}(\text{Noise}) \tag{53}$$

$$\le -\|\nabla f(X_t)\|_2^2 dt + \frac{\eta dL}{2}\left(\frac{\sigma_\gamma^2}{B} + \frac{C^2\sigma_{DP}^2}{B^2}\right)dt + \mathcal{O}(\text{Noise}). \tag{54}$$

Let $K_1$ and $K_2$ as in Eq. 49. Then, by integrating and taking the expectation, we have

$$\mathbb{E}\int_0^\tau \|\nabla f(X_t)\|_2^2 dt \leq K_1\left(f(X_0) - f(X_\tau) + \frac{\tau\eta dL}{2}\left(K_2 + \frac{C^2\sigma_{DP}^2}{B^2}\right)\right) \tag{55}$$

$$\Longrightarrow \mathbb{E}\int_0^\tau \frac{1}{\tau}\|\nabla f(X_t)\|_2^2 dt \leq K_1\left(\frac{f(X_0) - f(X_\tau)}{\tau} + \frac{\eta dL}{2}\left(K_2 + \frac{C^2\sigma_{DP}^2}{B^2}\right)\right) \tag{56}$$

$$\Longrightarrow \mathbb{E}\left[\|\nabla f(X_{\tilde{t}})\|_2^2\right] \leq K_1\left(\frac{2\varepsilon^2(f(X_0) - f(X_\tau)) + \eta^2 dLTK_2}{2\eta T} + \frac{\eta dLTC^2q^2\log(1/\delta)}{B^2}\right)\frac{1}{\varepsilon^2}, \tag{57}$$

where the last step follows from the Law of the Unconscious Statistician and $\tilde{t} \sim \mathrm{Unif}(0,\tau)$. Finally, by suppressing problem-independent constants, $2, \pi$, we obtain the claim.

$\square$

### B.1.1 MIXED-PHASE GRADIENT BOUND

In this section, we extend the two-phase SDE derivation to a single mixed setting. This is important because, at any point during training, some per-example gradients may exceed the clipping threshold while others remain below it. We show that in this scenario the same bound holds as in Theorem B.7, where it was previously derived under a worst-case approach.

**Theorem B.8** *Let $f : \mathbb{R}^d \to \mathbb{R}$ be $L$-smooth. Then, we can write the SDE of* `DP-SGD` *as*

$$dX_t = b_{\mathrm{mix}}(X_t)\,dt + \sqrt{\eta}\,\Sigma_{\mathrm{mix}}(X_t)^{1/2}dW_t, \tag{57}$$

*where the drift and covariance satisfy, for all $x$,*

$$\langle \nabla f(x), b_{\mathrm{mix}}(x)\rangle \leq -\frac{1}{K_1}\|\nabla f(x)\|^2, \tag{58}$$

$$\mathrm{Tr}\,\Sigma_{\mathrm{mix}}(x) \leq d\left(K_2 + \frac{C^2\sigma_{\mathrm{DP}}^2}{B^2}\right), \tag{59}$$

*where $K_1$, $K_2$ are defined in Equation 49. Therefore, for $\tilde{t} \sim \mathrm{Unif}(0,\tau)$,*

$$\mathbb{E}\|\nabla f(X_{\tilde{t}})\|_2^2 \leq K_1\left(\frac{f(X_0)}{\eta T} + \frac{\eta dL}{2}\left(K_2 + \frac{C^2\sigma_{\mathrm{DP}}^2}{B^2}\right)\right), \tag{60}$$

*i.e., the same $L$-smooth convergence bound as in Theorem B.7 holds for* any mixture *of clipped and unclipped samples in each mini-batch.*

**Proof:** We proceed in three steps: i) drift under mixed clipping, ii) covariance under mixed clipping using an explicit decomposition of $G_k$ into clipped and unclipped parts, and iii) Itô's formula and the final bound.

• Step 1: Drift of the mixed batch. The DP-SGD update can be written as

$$x_{k+1} = x_k - \eta\left(G_k + \frac{1}{B}Z_{\mathrm{DP}}\right), \quad \text{where} \quad G_k := \frac{1}{B}\sum_{i \in \gamma_k}\mathcal{C}[\nabla f_i(x_k)]. \tag{61}$$

Let

$$S_{1,k} := \{i \in \gamma_k : \|\nabla f_i(x_k)\| \geq C\}, \qquad S_{2,k} := \{i \in \gamma_k : \|\nabla f_i(x_k)\| < C\}, \tag{62}$$

$$B_k := |S_{1,k}|, \qquad p_k := \frac{B_k}{B} \in [0,1]. \tag{63}$$

Intuitively, $p_k$ represents the probability of a sample being in Phase 1. Define the per-sample contributions

$$Y_i := C(\nabla f_i(x_k)), \quad i \in S_{1,k}, \qquad X_i := \nabla f_i(x_k), \quad i \in S_{2,k}, \tag{64}$$

and the corresponding batch averages

$$g_k^{(1)} := \frac{1}{B}\sum_{i \in S_{1,k}} Y_i, \qquad g_k^{(2)} := \frac{1}{B}\sum_{i \in S_{2,k}} X_i, \tag{65}$$

so that

$$G_k = g_k^{(1)} + g_k^{(2)}. \tag{66}$$

In the same way as in the proof of Theorem B.5, we have that

$$\mathbb{E}[Y_i] = a_1 \nabla f(x_k), \qquad a_1 := \frac{CK(\nu)}{\sigma_\gamma \sqrt{d}}, \tag{67}$$

and

$$\mathbb{E}[X_i] = \nabla f(x_k). \tag{68}$$

Conditioned on the sets $S_{1,k}, S_{2,k}$, the $Y_i$ are i.i.d. over $S_{1,k}$ and the $X_i$ are i.i.d. over $S_{2,k}$, so

$$\mathbb{E}[g_k^{(1)} \mid S_{1,k}] = \frac{1}{B} \sum_{i \in S_{1,k}} \mathbb{E}[Y_i] = \frac{B_k}{B} \mathbb{E}[Y_i] = p_k a_1 \nabla f(x_k), \tag{69}$$

$$\mathbb{E}[g_k^{(2)} \mid S_{2,k}] = \frac{1}{B} \sum_{i \in S_{2,k}} \mathbb{E}[X_i] = \frac{B - B_k}{B} \mathbb{E}[X_i] = (1 - p_k) \nabla f(x_k). \tag{70}$$

Thus

$$\mathbb{E}[G_k \mid S_{1,k}, S_{2,k}] = p_k a_1 \nabla f(x_k) + (1 - p_k) \nabla f(x_k). \tag{71}$$

Recall the definition of $K_1$ from Equation 49

$$K_1 := \max\left\{1, \frac{\sigma_\gamma \sqrt{d}}{CK(\nu)}\right\}. \tag{72}$$

Then, it holds $a_1 = CK(\nu)/(\sigma_\gamma \sqrt{d}) \geq 1/K_1$ and $a_2 := 1 \geq 1/K_1$. Since $a_{\mathrm{mix}}(p_k)$ is a convex combination of $a_1$ and $a_2$,

$$a_{\mathrm{mix}}(p_k) = p_k a_1 + (1 - p_k) a_2 \geq \min\{a_1, a_2\} \geq \frac{1}{K_1} \qquad \forall \, p_k \in [0, 1]. \tag{73}$$

Therefore, the drift in the SDE limit satisfies

$$b_{\mathrm{mix}}(x) = -a_{\mathrm{mix}}(p(x)) \nabla f(x), \qquad \langle \nabla f(x), b_{\mathrm{mix}}(x) \rangle \leq -\frac{1}{K_1} \|\nabla f(x)\|^2. \tag{74}$$

• Step 2: Covariance of the mixed batch. We now compute the gradient noise covariance and show it is a convex combination of the pure-phase covariances, which are already derived in Theorem B.5. Define the centered contributions

$$U_i := Y_i - \mathbb{E}[Y_i], \quad i \in S_{1,k}, \qquad V_j := X_j - \mathbb{E}[X_j], \quad j \in S_{2,k}. \tag{75}$$

Then

$$g_k^{(1)} - \mathbb{E}[g_k^{(1)} \mid S_{1,k}] = \frac{1}{B} \sum_{i \in S_{1,k}} U_i, \qquad g_k^{(2)} - \mathbb{E}[g_k^{(2)} \mid S_{2,k}] = \frac{1}{B} \sum_{j \in S_{2,k}} V_j. \tag{76}$$

Hence

$$G_k - \mathbb{E}[G_k \mid S_{1,k}, S_{2,k}] = \frac{1}{B} \sum_{i \in S_{1,k}} U_i + \frac{1}{B} \sum_{j \in S_{2,k}} V_j. \tag{77}$$

Let

$$\Sigma_1^{\mathrm{single}}(x_k) := \mathrm{Cov}(Y_i) = \mathrm{Cov}(U_i), \qquad \Sigma_2^{\mathrm{single}}(x_k) := \mathrm{Cov}(X_j) = \mathrm{Cov}(V_j). \tag{78}$$

be the covariances of a single data-point. Conditioned on the sets $S_{1,k}, S_{2,k}$, the random vectors $\{U_i : i \in S_{1,k}\}$ and $\{V_j : j \in S_{2,k}\}$ are independent and zero-mean. Thus

$$\mathrm{Cov}(G_k \mid S_{1,k}, S_{2,k}) = \mathrm{Cov}\left(\frac{1}{B} \sum_{i \in S_{1,k}} U_i + \frac{1}{B} \sum_{j \in S_{2,k}} V_j \,\Big|\, S_{1,k}, S_{2,k}\right) \tag{79}$$

$$= \frac{1}{B^2} \mathrm{Cov}\left(\sum_{i \in S_{1,k}} U_i\right) + \frac{1}{B^2} \mathrm{Cov}\left(\sum_{j \in S_{2,k}} V_j\right), \tag{80}$$

where cross terms vanish by independence. Using i.i.d. within each group, we have

$$\text{Cov}\Big(\sum_{i \in S_{1,k}} U_i\Big) = B_k \Sigma_1^{\text{single}}(x_k), \qquad \text{Cov}\Big(\sum_{j \in S_{2,k}} V_j\Big) = (B - B_k)\Sigma_2^{\text{single}}(x_k), \qquad (81)$$

therefore

$$\Sigma_{\text{grad}}(x_k; S_{1,k}, S_{2,k}) := \text{Cov}(G_k \mid x_k, S_{1,k}, S_{2,k}) = \frac{B_k}{B^2}\Sigma_1^{\text{single}}(x_k) + \frac{B - B_k}{B^2}\Sigma_2^{\text{single}}(x_k). \quad (82)$$

Since $p_k = B_k/B$ and $1 - p_k = (B - B_k)/B$, we obtain

$$\Sigma_{\text{grad}}(x_k; S_{1,k}, S_{2,k}) = \frac{p_k}{B}\Sigma_1^{\text{single}}(x_k) + \frac{1 - p_k}{B}\Sigma_2^{\text{single}}(x_k). \qquad (83)$$

In the pure-phase SDEs of Theorem B.5, the *batch-level* (gradient and DP) covariances are given by

$$\Sigma_1(\bar{x}) = \frac{1}{B}\Sigma_1^{\text{single}}(x) + \frac{C^2\sigma_{\text{DP}}^2}{B^2}I_d, \qquad \Sigma_2(\bar{x}) = \frac{1}{B}\Sigma_2^{\text{single}}(x) + \frac{C^2\sigma_{\text{DP}}^2}{B^2}I_d. \qquad (84)$$

From equation 83, the *gradient* part of the mixed-phase covariance is

$$\Sigma_{\text{grad}}(x_k; S_{1,k}, S_{2,k}) = p_k\left(\Sigma_1(x_k) - \frac{C^2\sigma_{\text{DP}}^2}{B^2}I_d\right) + (1 - p_k)\left(\Sigma_2(x_k) - \frac{C^2\sigma_{\text{DP}}^2}{B^2}I_d\right). \quad (85)$$

Adding the DP noise term $\frac{C^2\sigma_{\text{DP}}^2}{B^2}I_d$ back in, the *full* covariance of the DP-SGD increment in the mixed batch is

$$\Sigma_{\text{mix}}(x_k; S_{1,k}, S_{2,k}) = \Sigma_{\text{grad}}(x_k; S_{1,k}, S_{2,k}) + \frac{C^2\sigma_{\text{DP}}^2}{B^2}I_d \qquad (86)$$

$$= p_k\Sigma_1(x_k) + (1 - p_k)\Sigma_2(x_k). \qquad (87)$$

Thus, at the SDE level, the mixed-phase covariance is exactly a convex combination of the pure-phase covariances $\Sigma_1$ and $\Sigma_2$. From Theorem B.6, we have the trace bounds

$$\text{Tr}\,\Sigma_1(\bar{x}) \leq \frac{C^2}{B} + d\frac{C^2\sigma_{\text{DP}}^2}{B^2}, \qquad (88)$$

$$\text{Tr}\,\Sigma_2(\bar{x}) = d\left(\frac{\sigma_\gamma^2}{B} + \frac{C^2\sigma_{\text{DP}}^2}{B^2}\right). \qquad (89)$$

Let $K_2$ as in Equation 49 we can write, for $r \in \{1, 2\}$,

$$\text{Tr}\,\Sigma_r(\bar{x}) \leq d\left(K_2 + \frac{C^2\sigma_{\text{DP}}^2}{B^2}\right). \qquad (90)$$

Using equation 87, for any $p_k \in [0, 1]$,

$$\text{Tr}\,\Sigma_{\text{mix}}(x_k) = p_k\,\text{Tr}\,\Sigma_1(x_k) + (1 - p_k)\,\text{Tr}\,\Sigma_2(x_k) \leq d\left(K_2 + \frac{C^2\sigma_{\text{DP}}^2}{B^2}\right). \qquad (91)$$

Hence, the mixed-phase covariance satisfies exactly the same worst-case trace bound as the pure-phase covariances.

• Step 3: Itô bound and convergence. Finally, we can rewrite the SDE of `DP-SGD` as follows:

$$dX_t = b_{\text{mix}}(X_t)\,dt + \sqrt{\eta}\,\Sigma_{\text{mix}}(X_t)^{1/2}dW_t, \qquad (92)$$

where, for all $x$,

$$\langle\nabla f(x), b_{\text{mix}}(x)\rangle \leq -\frac{1}{K_1}\|\nabla f(x)\|^2, \qquad (93)$$

$$\text{Tr}\,\Sigma_{\text{mix}}(x) \leq d\left(K_2 + \frac{C^2\sigma_{\text{DP}}^2}{B^2}\right). \qquad (94)$$

Since $f$ is $L$-smooth, $\nabla^2 f(x) \preceq LI_d$. By Itô's formula,

$$df(X_t) = \langle\nabla f(X_t), b_{\text{mix}}(X_t)\rangle\,dt + \frac{\eta}{2}\text{Tr}\big(\nabla^2 f(X_t)\Sigma_{\text{mix}}(X_t)\big)\,dt + \mathcal{O}(\text{Noise}). \qquad (95)$$

Using the drift and covariance bounds,

$$df(X_t) \leq -\frac{1}{K_1} \|\nabla f(X_t)\|_2^2 \, dt + \frac{\eta d L}{2} \left( K_2 + \frac{C^2 \sigma_{\mathrm{DP}}^2}{B^2} \right) dt + \mathcal{O}(\text{Noise}). \tag{96}$$

Integrating from 0 to $\tau := \eta T$,

$$f(X_\tau) - f(X_0) \leq -\frac{1}{K_1} \int_0^\tau \|\nabla f(X_t)\|_2^2 \, dt + \frac{\eta d L}{2} \left( K_2 + \frac{C^2 \sigma_{\mathrm{DP}}^2}{B^2} \right) \tau + \mathcal{O}(\text{Noise}). \tag{97}$$

Rearranging,

$$\frac{1}{K_1} \int_0^\tau \|\nabla f(X_t)\|_2^2 \, dt \leq f(X_0) - f(X_\tau) + \frac{\eta d L}{2} \left( K_2 + \frac{C^2 \sigma_{\mathrm{DP}}^2}{B^2} \right) \tau + \mathcal{O}(\text{Noise}). \tag{98}$$

Taking expectations,

$$\frac{1}{K_1} \mathbb{E} \int_0^\tau \|\nabla f(X_t)\|_2^2 \, dt \leq \mathbb{E}[f(X_0) - f(X_\tau)] + \frac{\eta d L}{2} \left( K_2 + \frac{C^2 \sigma_{\mathrm{DP}}^2}{B^2} \right) \tau. \tag{99}$$

Let $\tilde{t} \sim \mathrm{Unif}(0, \tau)$. Then, by the Law of the Unconscious Statistician,

$$\mathbb{E}\|\nabla f(X_{\tilde{t}})\|^2 = \frac{1}{\tau} \mathbb{E} \int_0^\tau \|\nabla f(X_t)\|_2^2 \, dt \leq K_1 \left( \frac{f(X_0)}{\tau} + \frac{\eta d L}{2} \left( K_2 + \frac{C^2 \sigma_{\mathrm{DP}}^2}{B^2} \right) \right). \tag{100}$$

Since $\tau = \eta T$, this is exactly the gradient-norm bound as in Theorem B.7, with the same constants $K_1$, $K_2$, now rigorously shown to hold under arbitrary mixtures of clipped and unclipped samples at each iteration.

$\square$

We now derive the stationary distribution of DP-SGD at convergence: We empirically validate this result in Figure C.6.

**Theorem B.9** Let $f(x) = \frac{1}{2} x^\top H x$ where $H = \mathrm{diag}(\lambda_1, \ldots, \lambda_d)$. The stationary distribution at convergence of DP-SGD is

$$(\mathbb{E}[X_\tau], \mathrm{Cov}(X_\tau)) = \left( X_0 e^{-H\tau}, \frac{T\eta}{2\varepsilon^2} \left( \frac{\varepsilon^2 \sigma_\gamma^2}{BT} + \frac{C^2 q^2 \log(1/\delta)}{B^2} \right) (1 - e^{-2H\tau}) H^{-1} \right). \tag{101}$$

**Proof:** Since $H$ is diagonal, we can isolate each component. Furthermore, since $f(\cdot)$ is quadratic we can rewrite the SDE as:

$$dX_{t,i} = -\lambda_i X_{t,i} dt + \sqrt{\eta} \sqrt{\frac{\sigma_\gamma^2}{B} + \frac{C^2 \sigma_{DP}^2}{B^2}} dW_{t,i}. \tag{102}$$

We have immediately that

$$\mathbb{E}[X_{t,i}] = X_{0,i} e^{-\lambda_i t}. \tag{103}$$

Applying the Itô isometry, we obtain:

$$\mathbb{E}[(X_{t,i} - \mathbb{E}[X_{t,i}])^2]$$

$$= \eta \mathbb{E} \left[ \left( \int_0^t \left( e^{-\lambda_i(t-s)} \sqrt{\frac{\sigma_\gamma^2}{B} + \frac{C^2 \sigma_{DP}^2}{B^2}} dW_s \right)^\top \left( e^{-\lambda_i(t-s)} \sqrt{\frac{\sigma_\gamma^2}{B} + \frac{C^2 \sigma_{DP}^2}{B^2}} dW_s \right) \right] \right]$$

$$= \eta \left( \frac{\sigma_\gamma^2}{B} + \frac{C^2 \sigma_{DP}^2}{B^2} \right) \int_0^t e^{-2\lambda_i(t-s)} ds$$

$$= \frac{\eta}{2\lambda_i} \left( \frac{\sigma_\gamma^2}{B} + \frac{C^2 q^2 T \log(1/\delta)}{B^2 \varepsilon^2} \right) (1 - e^{-2\lambda_i t})$$

$$= \frac{T\eta}{2\varepsilon^2 \lambda_i} \left( \frac{\varepsilon^2 \sigma_\gamma^2}{BT} + \frac{C^2 q^2 \log(1/\delta)}{B^2} \right) (1 - e^{-2\lambda_i t}).$$

$\square$

## B.2 DP-SIGNSGD

This subsection provides the formal derivation of the SDE model for DP-SignSGD (Theorem B.10), together with the formal loss and gradient-norm bounds used in the main paper (Theorems B.12 and B.13) and the quadratic stationary distribution (Theorem B.15).

**Idealized clipping regimes vs. the mixed regime.** As in the main paper (Section 4.1.2), we first analyze two idealized regimes: (i) *fully-clipped regime* ("Phase 1"), where sampled per-example gradients are clipped and the sign update acts on a privatized, normalized direction; and (ii) *unclipped regime* ("Phase 2"), where clipping is inactive and the privatized gradient is approximately Gaussian after averaging. These regimes are used only to isolate how $\varepsilon$ enters the drift and diffusion. The realistic *mixed* regime (a mixture of clipped and unclipped samples within each minibatch) is treated in the mixed-phase analysis below (Theorem B.14).

**Theorem B.10** *Let $0 < \eta < 1$, $\tau > 0$ and set $T = \lfloor \tau/\eta \rfloor$ and $K(\nu) = \sqrt{\frac{2}{\nu}} \frac{\Gamma\left(\frac{\nu+1}{2}\right)}{\Gamma\left(\frac{\nu}{2}\right)}$, $\nu \geq 1$. Let $x_k \in \mathbb{R}^d$ denote a sequence of DP-SignSGD iterations defined in Eq. 4. Assume Assumption B.1 and Assumption B.2. Let $X_t$ be the solution of the following SDEs with initial condition $X_0 = x_0$:*

• *Phase 1:*

$$dX_t = -\mathbb{E}_\gamma\left[\mathrm{Erf}\left(\frac{B}{\sigma_{DP}\sqrt{2}} u_\gamma(X_t)\right)\right] dt + \sqrt{\eta}\sqrt{\bar{\Sigma}(X_t)} dW_t, \tag{104}$$

*where, for a mini-batch $\gamma$ of size $B$, we define the averaged normalized direction*

$$u_\gamma(x) := \frac{1}{B} \sum_{\xi \in \gamma} \frac{\nabla f_\xi(x)}{\|\nabla f_\xi(x)\|_2}, \tag{105}$$

*and we keep the diffusion covariance implicit as*

$$\bar{\Sigma}(x) := \mathrm{Cov}_{\gamma,DP}(\mathrm{sign}\left(Cu_\gamma(x) + Z_{DP}\right)), \qquad Z_{DP} \sim \mathcal{N}\left(0, \frac{C^2 \sigma_{DP}^2}{B^2} I_d\right). \tag{106}$$

• *Phase 2:*

$$dX_t = -\mathrm{Erf}\left(\frac{\nabla f(X_t)}{\sqrt{2\left(\frac{C^2 \sigma_{DP}^2}{B^2} + \frac{\sigma_\gamma^2}{B}\right)}}\right) dt + \sqrt{\eta}\sqrt{\bar{\Sigma}(X_t)} dW_t, \tag{107}$$

*where $\bar{\Sigma}(x) = I_d - \mathrm{Erf}\left(\frac{\nabla f(x)}{\sqrt{2\left(\frac{C^2 \sigma_{DP}^2}{B^2} + \frac{\sigma_\gamma^2}{B}\right)}}\right)^2$ and $\mathrm{Erf}(\cdot)$ is applied component-wise.*

*Then, Eq. 104 and Eq. 107 are an order $1$ approximation of the discrete update of Phase 1 and Phase 2 of DP-SignSGD, respectively.*

**Proof:** The proof is virtually identical to that of Theorem B.5. Hence, we highlight only the necessary details for each phase. Let $\bar{\Delta} = x_1 - x$ be the one-step increment.

• Phase 1: In the fully-clipped regime, the privatized direction used by DP-SignSGD is

$$g(x) = \frac{C}{B} \sum_{\xi \in \gamma} \frac{\nabla f_\xi(x)}{\|\nabla f_\xi(x)\|_2} + Z_{DP} = C u_\gamma(x) + Z_{DP}, \tag{108}$$

with $u_\gamma$ as in Eq. 105 and $Z_{DP} \sim \mathcal{N}\left(0, \frac{C^2 \sigma_{DP}^2}{B^2} I_d\right)$. Therefore

$$\mathbb{E}[\bar{\Delta}] = -\eta \mathbb{E}_{\gamma,DP}\left[\mathrm{sign}\left(Cu_\gamma(x) + Z_{DP}\right)\right]. \tag{109}$$

Remember that, for any scalar random variable $Y$, we have $\mathbb{E}[\mathrm{sign}(Y)] = 1 - 2\mathbb{P}(Y < 0)$, and for $Y \sim \mathcal{N}(0,1)$ it holds $\Phi(y) = \frac{1}{2}\left(1 + \mathrm{Erf}\left(\frac{y}{\sqrt{2}}\right)\right)$. Fix $i \in \{1, \ldots, d\}$. Conditioned on $\gamma$,

the random variable $Cu_{\gamma,i}(x) + Z_{DP,i}$ is Gaussian with mean $Cu_{\gamma,i}(x)$ and variance $C^2\sigma_{DP}^2/B^2$, hence

$$\mathbb{E}_{DP}[\text{sign}\left(Cu_{\gamma,i}(x) + Z_{DP,i}\right) | \gamma] = 1 - 2\Phi\left(-\frac{B}{\sigma_{DP}}u_{\gamma,i}(x)\right)$$

$$= \text{Erf}\left(\frac{B}{\sigma_{DP}\sqrt{2}}u_{\gamma,i}(x)\right). \tag{110}$$

Collecting the coordinates and taking expectation over $\gamma$ yields

$$\mathbb{E}[\bar{\Delta}] = -\eta\mathbb{E}_{\gamma}\left[\text{Erf}\left(\frac{B}{\sigma_{DP}\sqrt{2}}u_{\gamma}(x)\right)\right]. \tag{111}$$

For the second moment, define $S(x) := \text{sign}(Cu_{\gamma}(x) + Z_{DP})$ and $m(x) := \mathbb{E}[S(x)]$. Then

$$\text{Cov}(\bar{\Delta}) = \eta^2\text{Cov}(S(x)) =: \eta^2\bar{\Sigma}(x). \tag{112}$$

We emphasize that we do *not* assume the coordinates of $S(x)$ to be independent after averaging over $\gamma$ (off-diagonal entries of $\bar{\Sigma}(x)$ may be non-zero). Nonetheless, since $S_i(x) \in \{\pm 1\}$, we have for each coordinate

$$\text{Var}(S_i(x)) = 1 - m_i(x)^2, \tag{113}$$

and therefore the trace satisfies

$$\text{Tr}(\bar{\Sigma}(x)) = \sum_{i=1}^{d}\text{Var}(S_i(x)) = d - \|m(x)\|_2^2 \le d. \tag{114}$$

This trace bound is the only property of the diffusion used later in Lyapunov arguments. Define now

$$b(x) := -\mathbb{E}_{\gamma}\left[\text{Erf}\left(\frac{B}{\sigma_{DP}\sqrt{2}}u_{\gamma}(x)\right)\right], \tag{115}$$

$$\bar{\Sigma}(x) := \text{Cov}_{\gamma,DP}(\text{sign}\left(Cu_{\gamma}(x) + Z_{DP}\right)). \tag{116}$$

Then, from Lem B.3 and Thm. B.4 the claim for Phase 1 follows.

• Phase 2: Remember that, from Assumption B.2, $\nabla f_{\gamma} = \nabla f + Z_{\gamma}$, where $Z_{\gamma} \sim \mathcal{N}\left(0, \frac{\sigma_{\gamma}^2}{B}\right)$. We calculate the expected increment

$$\mathbb{E}[\bar{\Delta}] = -\eta\mathbb{E}\left[\text{sign}(\nabla f_{\gamma}(x) + Z_{DP})\right] \tag{117}$$

$$= -\eta\mathbb{E}[\text{sign}(\nabla f(x) + Z_{\gamma} + Z_{DP})] \tag{118}$$

$$= -\eta\,\text{Erf}\left(\frac{\nabla f(x)}{\sqrt{2\left(\frac{C^2\sigma_{DP}^2}{B^2} + \frac{\sigma_{\gamma}^2}{B}\right)}}\right). \tag{119}$$

Instead, the covariance becomes

$$\text{Cov}(\bar{\Delta})_{ij} = \eta^2\mathbb{E}_{\gamma,DP}\left[\left(\text{sign}(\nabla f_{\gamma} + Z_{DP}) - \text{Erf}\left(\frac{\nabla f(x)}{\sqrt{2\left(\frac{C^2\sigma_{DP}^2}{B^2} + \frac{\sigma_{\gamma}^2}{B}\right)}}\right)\right)_i \right. \tag{120}$$

$$\left.\left(\text{sign}(\nabla f_{\gamma} + Z_{DP}) - \text{Erf}\left(\frac{\nabla f(x)}{\sqrt{2\left(\frac{C^2\sigma_{DP}^2}{B^2} + \frac{\sigma_{\gamma}^2}{B}\right)}}\right)\right)_j\right] \tag{121}$$

$$= \eta^2\mathbb{E}_{\gamma,DP}\left[\text{sign}(\nabla f_{\gamma} + Z_{DP})_i\,\text{sign}(\nabla f_{\gamma} + Z_{DP})_j\right] \tag{122}$$

$$- \eta^2\,\text{Erf}\left(\frac{\partial_i f(x)}{\sqrt{2\left(\frac{C^2\sigma_{DP}^2}{B^2} + \frac{\sigma_{\gamma}^2}{B}\right)}}\right)\text{Erf}\left(\frac{\partial_j f(x)}{\sqrt{2\left(\frac{C^2\sigma_{DP}^2}{B^2} + \frac{\sigma_{\gamma}^2}{B}\right)}}\right). \tag{123}$$

If $i = j$, we have

$$\text{Cov}(\bar{\Delta})_{ii} = \eta^2 \left( 1 - \text{Erf} \left( \frac{\partial_i f(x)}{\sqrt{2 \left( \frac{C^2 \sigma_{DP}^2}{B^2} + \frac{\sigma_\gamma^2}{B} \right)}} \right)^2 \right) ; \tag{124}$$

while if $i \neq j$

$$\text{Cov}(\bar{\Delta})_{ij} = \eta^2 \, \text{Erf} \left( \frac{\partial_i f(x)}{\sqrt{2 \left( \frac{C^2 \sigma_{DP}^2}{B^2} + \frac{\sigma_\gamma^2}{B} \right)}} \right) \text{Erf} \left( \frac{\partial_j f(x)}{\sqrt{2 \left( \frac{C^2 \sigma_{DP}^2}{B^2} + \frac{\sigma_\gamma^2}{B} \right)}} \right) \tag{125}$$

$$- \eta^2 \, \text{Erf} \left( \frac{\partial_i f(x)}{\sqrt{2 \left( \frac{C^2 \sigma_{DP}^2}{B^2} + \frac{\sigma_\gamma^2}{B} \right)}} \right) \text{Erf} \left( \frac{\partial_j f(x)}{\sqrt{2 \left( \frac{C^2 \sigma_{DP}^2}{B^2} + \frac{\sigma_\gamma^2}{B} \right)}} \right) = 0, \tag{126}$$

Define now

$$b(x) = - \text{Erf} \left( \frac{\nabla f(x)}{\sqrt{2 \left( \frac{C^2 \sigma_{DP}^2}{B^2} + \frac{\sigma_\gamma^2}{B} \right)}} \right) \tag{127}$$

$$\bar{\Sigma}(x) = I_d - \text{Erf} \left( \frac{\nabla f(x)}{\sqrt{2 \left( \frac{C^2 \sigma_{DP}^2}{B^2} + \frac{\sigma_\gamma^2}{B} \right)}} \right)^2 . \tag{128}$$

Then, from Lem B.3 and Thm. B.4 the claim follows.

$\square$

**Corollary B.11** *Under our assumptions, using the linear approximation* $\text{Erf}(z) \sim \frac{2}{\sqrt{\pi}} z$ *in a neighborhood of* $0$ *together with Lemma A.2, the drift terms in Eq. 104 and Eq. 107 simplify as follows. We keep the diffusion covariance* implicit *(since our Lyapunov bounds depend on it only through its trace).*

• *Phase 1:*

$$dX_t = -\sqrt{\frac{2}{d\pi}} \frac{BK(\nu)}{\sigma_{DP}\sigma_\gamma} \nabla f(X_t) \, dt + \sqrt{\eta} \, \bar{\Sigma}_1(X_t)^{1/2} \, dW_t, \tag{129}$$

*where* $\bar{\Sigma}_1(x) = \text{Cov}_{\gamma,DP}(\text{sign}\,(Cu_\gamma(x) + Z_{DP}))$ *with* $u_\gamma$ *as in Eq. 105 and* $Z_{DP}$ *as in Eq. 106.*

• *Phase 2:*

$$dX_t = -\sqrt{\frac{2}{\pi}} \frac{1}{\sqrt{\frac{C^2 \sigma_{DP}^2}{B^2} + \frac{\sigma_\gamma^2}{B}}} \nabla f(X_t) \, dt + \sqrt{\eta} \, \bar{\Sigma}_2(X_t)^{1/2} \, dW_t, \tag{130}$$

*where* $\bar{\Sigma}_2(x) := I_d - \text{Erf} \left( \frac{\nabla f(x)}{\sqrt{2 \left( \frac{C^2 \sigma_{DP}^2}{B^2} + \frac{\sigma_\gamma^2}{B} \right)}} \right)^2 .$

*Moreover, the diffusion satisfies*

$$\text{Tr}(\bar{\Sigma}_j(x)) \leq d, \tag{131}$$

*which is the only property of the diffusion used in our trace-based Lyapunov bounds.*

**Proof:** Let us recall that

$$u_\gamma(x) := \frac{1}{B} \sum_{\xi \in \gamma} \frac{\nabla f_\xi(x)}{\|\nabla f_\xi(x)\|_2}. \tag{132}$$

Then, $|u_{\gamma,i}(x)| \ll 1$ for all $i = 1, \ldots, d$ by definition. Additionally, let us consider the term $\frac{|\partial_i f(x)|}{\sqrt{2\left(\frac{C^2 \sigma_{DP}^2}{B^2} + \sigma_\gamma^2/B\right)}}$, which appears in Eq. 107 in the argument of the error function Erf. That can be interpreted as a Signal-to-Noise Ratio (SNR), and it is clear that if this is large, the descent is stronger, while if it is small, the descent is weaker. To derive our results, we put ourselves in the worst-case-scenario where the SNR is small, e.g. $\frac{|\partial_i f(x)|}{\sqrt{2\left(\frac{C^2 \sigma_{DP}^2}{B^2} + \sigma_\gamma^2/B\right)}} \ll 1$. We add that this is particularly true when $\varepsilon \to 0$, which is the high-privacy regime we study in this paper.

- Phase 1: Since $|u_{\gamma,i}(x)| \ll 1$, we can use $\mathrm{Erf}(z) \sim \frac{2}{\sqrt{\pi}} z$ and write

$$
\begin{aligned}
\mathbb{E}_\gamma \left[ \mathrm{Erf} \left( \frac{B}{\sigma_{DP}\sqrt{2}} u_\gamma(x) \right) \right] &= \mathbb{E}_\gamma \left[ \sqrt{\frac{2}{\pi}} \frac{B}{\sigma_{DP}} u_\gamma(x) \right] \\
&= \sqrt{\frac{2}{\pi}} \frac{B}{\sigma_{DP}} \mathbb{E}_\xi \left[ \frac{\nabla f_\xi(x)}{\|\nabla f_\xi(x)\|_2} \right] \\
&= \sqrt{\frac{2}{d\pi}} \frac{B K(\nu)}{\sigma_{DP} \sigma_\gamma} \nabla f(x), \tag{133}
\end{aligned}
$$

where in the last step we used Lemma A.2. Therefore, the drift in Eq. 104 becomes the one stated in Eq. 129.

For the diffusion term we keep $\bar{\Sigma}_1(x) = \mathrm{Cov}_{\gamma,DP}(S_1(x))$ implicit, where $S_1(x) := \mathrm{sign}(C u_\gamma(x) + Z_{DP})$ and $m_1(x) := \mathbb{E}[S_1(x)]$. Since $S_{1,i}(x) \in \{\pm 1\}$, we have $\mathrm{Var}(S_{1,i}(x)) = 1 - m_{1,i}(x)^2$, hence $\mathrm{Tr}(\bar{\Sigma}_1(x)) = d - \|m_1(x)\|_2^2 \le d$.

- Phase 2: Since $\left| \frac{\partial_i f(x)}{\sqrt{2\left(\frac{C^2 \sigma_{DP}^2}{B^2} + \sigma_\gamma^2/B\right)}} \right| \ll 1$ for $i = 1, \ldots, d$, we can use the same linear approximation of the error function. In detail, one has

$$\mathrm{Erf}\left( \frac{\nabla f(x)}{\sqrt{2\left(\frac{C^2 \sigma_{DP}^2}{B^2} + \frac{\sigma_\gamma^2}{B}\right)}} \right) = \frac{2}{\sqrt{\pi}} \frac{\nabla f(x)}{\sqrt{2\left(\frac{C^2 \sigma_{DP}^2}{B^2} + \frac{\sigma_\gamma^2}{B}\right)}} = \sqrt{\frac{2}{\pi}} \frac{1}{\sqrt{\frac{C^2 \sigma_{DP}^2}{B^2} + \frac{\sigma_\gamma^2}{B}}} \nabla f(x). \tag{134}$$

Therefore, Eq. 107 becomes Eq. 130. The diffusion covariance $\bar{\Sigma}_2(x) = I_d - S_2$ with $S_2(x) := \mathrm{Erf}\left( \frac{\nabla f(x)}{\sqrt{2\left(\frac{C^2 \sigma_{DP}^2}{B^2} + \frac{\sigma_\gamma^2}{B}\right)}} \right)^2$ also satisfies $\mathrm{Tr}(\bar{\Sigma}_2(x)) = d - \|S_2(x)\|_2^2 \le d$ by the same argument. $\square$

**Remark B.3** *In the loss and gradient-norm bounds that follow, the diffusion enters only through the trace term $\mathrm{Tr}(\nabla^2 f(X_t) \bar{\Sigma}(X_t)) \le L \, \mathrm{Tr}(\bar{\Sigma}(X_t))$. Since $\bar{\Sigma}(x) = \mathrm{Cov}(S(x))$ for some $S(x) \in \{\pm 1\}^d$, we always have $\mathrm{Tr}(\bar{\Sigma}(x)) \le d$; hence we keep $\bar{\Sigma}$ implicit and do not require any diagonal approximation or coordinate-independence assumption.*

**Theorem B.12** *Let $f$ be $L$-smooth and $\mu$-PL. Then, for $t \in [0, \tau]$, we have that*

- *Phase 1, i.e., when the gradient is clipped, the loss satisfies:*

$$\mathbb{E}[f(X_t)] \lesssim f(X_0) e^{\frac{-\mu B}{\sigma_\gamma \sqrt{dT}} \frac{\varepsilon}{\Phi} t} + \left( 1 - e^{\frac{-\mu B}{\sigma_\gamma \sqrt{dT}} \frac{\varepsilon}{\Phi} t} \right) \frac{\sqrt{T} \eta L d^{\frac{3}{2}} \sigma_\gamma}{\mu B} \frac{\Phi}{\varepsilon}; \tag{135}$$

- *Phase 2, i.e., when the gradient is not clipped, the loss satisfies:*

$$\mathbb{E}[f(X_t)] \lesssim f(X_0)e^{\frac{-\mu\varepsilon t}{\sqrt{\varepsilon^2\frac{\sigma_\gamma^2}{B}+\frac{C^2\Phi^2}{B^2}T}}} + \left(1 - e^{\frac{-\mu\varepsilon t}{\sqrt{\varepsilon^2\frac{\sigma_\gamma^2}{B}+\frac{C^2\Phi^2}{B^2}T}}}\right)\frac{\sqrt{T}\eta Ld}{\mu}\sqrt{\frac{\varepsilon^2\sigma_\gamma^2}{BT}+\frac{C^2\Phi^2}{B^2}}\frac{1}{\varepsilon}. \quad (136)$$

**Proof:** First of all, observe that in both phases $\bar{\Sigma}(x) = \mathrm{Cov}(S(x))$ for some $S(x) \in \{\pm 1\}^d$. In particular, $\mathrm{Tr}(\bar{\Sigma}(x)) \leq d$ (and thus $\mathrm{Tr}(\nabla^2 f(X_t)\bar{\Sigma}(X_t)) \leq Ld$ when $\nabla^2 f \preceq LI_d$).

- Phase 1: Since $f$ is $\mu$-PL and $L$-smooth it follows that $2\mu f(x) \leq \|\nabla f(x)\|^2$ and $\nabla^2 f(x) \preceq LI_d$. Then, By applying the Itô formula, we have

$$df(X_t) \leq -\sqrt{\frac{2}{d\pi T}}\frac{K(\nu)}{\sigma_\gamma}\frac{B\varepsilon}{q\log(1/\delta)}\|\nabla f(X_t)\|_2^2 dt + \frac{\eta}{2}\mathrm{Tr}\left(\nabla^2 f(X_t)\bar{\Sigma}(X_t)\right)dt + \mathcal{O}(\text{Noise}) \tag{137}$$

$$\leq -2\mu\sqrt{\frac{2}{d\pi T}}\frac{K(\nu)}{\sigma_\gamma}\frac{B\varepsilon}{q\log(1/\delta)}f(X_t)dt + \frac{\eta dL}{2}dt + \mathcal{O}(\text{Noise}). \tag{138}$$

Therefore,

$$\mathbb{E}[f(X_t)] \leq f(X_0)e^{-2\mu\left(\sqrt{\frac{2}{d\pi T}}\frac{K(\nu)}{\sigma_\gamma}\frac{B\varepsilon}{q\log(1/\delta)}\right)t} \tag{139}$$

$$+ \left(1 - e^{-2\mu\left(\sqrt{\frac{2}{d\pi T}}\frac{K(\nu)}{\sigma_\gamma}\frac{B\varepsilon}{q\log(1/\delta)}\right)t}\right)\sqrt{\frac{\pi T}{2}}\frac{\eta d^{\frac{3}{2}}L\sigma_\gamma}{4\mu K(\nu)}\frac{q\log(1/\delta)}{B\varepsilon}. \tag{140}$$

- Phase 2: As for Phase 1, by applying the Itô formula one has

$$df(X_t) \leq -\sqrt{\frac{2}{\pi}}\frac{1}{\sqrt{\varepsilon^2 B^{-1}\sigma_\gamma^2 + B^{-2}C^2 q^2\log(1/\delta)T}}\varepsilon\|\nabla f(X_t)\|_2^2 dt \tag{141}$$

$$+ \frac{\eta}{2}\mathrm{Tr}\left(\nabla^2 f(X_t)\bar{\Sigma}(X_t)\right)dt + \mathcal{O}(\text{Noise}) \tag{142}$$

$$\leq -2\mu\sqrt{\frac{2}{\pi}}\frac{1}{\sqrt{\varepsilon^2 B^{-1}\sigma_\gamma^2 + B^{-2}C^2 q^2\log(1/\delta)T}}\varepsilon f(X_t)dt + \frac{\eta dL}{2}dt + \mathcal{O}(\text{Noise}). \tag{143}$$

Therefore

$$\mathbb{E}[f(X_t)] \leq f(X_0)e^{-\sqrt{\frac{2}{\pi}}\frac{2\mu}{\sqrt{\varepsilon^2 B^{-1}\sigma_\gamma^2 + B^{-2}C^2 q^2\log(1/\delta)T}}\varepsilon t} \tag{144}$$

$$+ \left(1 - e^{-\sqrt{\frac{2}{\pi}}\frac{2\mu}{\sqrt{\varepsilon^2 B^{-1}\sigma_\gamma^2 + B^{-2}C^2 q^2\log(1/\delta)T}}\varepsilon t}\right)\sqrt{\frac{\pi T}{2}}\frac{\eta dL}{4\mu}\sqrt{\frac{\varepsilon^2\sigma_\gamma^2}{BT}+\frac{C^2 q^2\log(1/\delta)}{B^2}}\frac{1}{\varepsilon}.$$

Finally, by suppressing problem-independent constants, such as $2, \pi, K(\nu)$, the thesis follows.

$\square$

**Theorem B.13** *Let $f$ be an $L$-smooth function. Define*

$$K_3 = \max\left\{\sqrt{\frac{d\pi}{2}}\frac{\sigma_\gamma q\sqrt{\log(1/\delta)}}{BK(\nu)}, \sqrt{\frac{\pi}{2}}\sqrt{\frac{\varepsilon^2\sigma_\gamma^2}{BT}+\frac{C^2 q^2\log(1/\delta)}{B^2}}\right\}. \tag{145}$$

*Then*

$$\mathbb{E}\left[\|\nabla f(X_{\tilde{t}})\|_2^2\right] \lesssim K_3\left(\frac{f(X_0)}{\eta\sqrt{T}}+\eta dL\sqrt{T}\right)\frac{1}{\varepsilon}, \tag{146}$$

*where $\tilde{t} \sim \mathrm{Unif}(0,\tau)$.*

**Proof:** Since in both phases $\bar{\Sigma}(x) = \mathrm{Cov}(S(x))$ for some $S(x) \in \{\pm 1\}^d$, we have $\mathrm{Tr}(\bar{\Sigma}(x)) \leq d$, hence $\mathrm{Tr}(\nabla^2 f(X_t)\bar{\Sigma}(X_t)) \leq Ld$ when $\nabla^2 f \preceq LI_d$. Therefore, for a worst-case analysis the drift is the only term worth comparing. Let then $K_3$ as in Eq. 145. Applying the Itô formula to the worst-case SDE we have

$$df(X_t) \leq -\varepsilon(\sqrt{T}K_3)^{-1}\|\nabla f(X_t)\|_2^2 dt + \frac{\eta}{2}\mathrm{Tr}\left(\nabla^2 f(X_t)\bar{\Sigma}(X_t)\right) dt + \mathcal{O}(\text{Noise}) \tag{147}$$

$$\leq -\varepsilon(\sqrt{T}K_3)^{-1}\|\nabla f(X_t)\|_2^2 dt + \frac{\eta dL}{2} dt + \mathcal{O}(\text{Noise}). \tag{148}$$

Then, by integrating and taking the expectation

$$\mathbb{E}\int_0^\tau \|\nabla f(X_t)\|_2^2 dt \leq K_3\sqrt{T}\left(f(X_0) + \frac{\eta dL\tau}{2}\right)\varepsilon^{-1} \tag{149}$$

$$\Longrightarrow \mathbb{E}\int_0^\tau \frac{1}{\tau}\|\nabla f(X_t)\|_2^2 dt \leq \frac{K_3}{\eta\sqrt{T}}\left(f(X_0) + \frac{\eta dL\tau}{2}\right)\varepsilon^{-1} \tag{150}$$

$$\Longrightarrow \mathbb{E}\left[\|\nabla f(X_{\tilde{t}})\|_2^2\right] \leq K_3\left(\frac{f(X_0)}{\eta\sqrt{T}} + \frac{\eta dL\sqrt{T}}{2}\right)\frac{1}{\varepsilon}. \tag{151}$$

where in the last step we used the Law of the Unconscious Statistician and $\tilde{t} \sim \mathrm{Unif}(0,\tau)$. Finally, by suppressing problem-independent constants, we get the thesis. $\qquad\square$

### B.2.1 MIXED-REGIME GRADIENT BOUND

Analogously to Section B.1.1, we extend the idealized fully-clipped/unclipped analysis to the realistic *mixed* regime in which each mini-batch contains both clipped and unclipped samples. The following result shows that the same $\mathcal{O}(1/\varepsilon)$ gradient-norm scaling as in Theorem B.13 continues to hold in this mixed regime (up to constants).

**Theorem B.14** *Let $f : \mathbb{R}^d \to \mathbb{R}$ be L-smooth. In the mixed regime, we can write the SDE of* `DP-SignSGD` *as*

$$dX_t = b_{\text{mix}}(X_t)\, dt + \sqrt{\eta}\, \bar{\Sigma}_{\text{mix}}(X_t)^{1/2} dW_t, \tag{152}$$

*where*

$$b_{\text{mix}}(x) = -\mathbb{E}\left[\mathrm{Erf}\left(\frac{B}{C\sigma_{DP}\sqrt{2}}G(x)\right)\right], \tag{153}$$

$$\bar{\Sigma}_{\text{mix}}(x) := \mathrm{Cov}_{\gamma,DP}\left(\mathrm{sign}\left(G(x) + \frac{1}{B}Z_{DP}\right)\right), \tag{154}$$

*with*

$$G(x) = \frac{1}{B}\sum_{\xi\in\gamma}\mathcal{C}[\nabla f_\xi(x)], \qquad Z_{DP} \sim \mathcal{N}(0, C^2\sigma_{DP}^2 I_d). \tag{155}$$

*In particular, letting $m_{\text{mix}}(x) := \mathbb{E}\left[\mathrm{Erf}\left(\frac{B}{C\sigma_{DP}\sqrt{2}}G(x)\right)\right]$, we have $\mathrm{Tr}(\bar{\Sigma}_{\text{mix}}(x)) = d - \|m_{\text{mix}}(x)\|_2^2 \leq d$.*

*Define*

$$K_4 = \max\left\{\sqrt{\frac{\pi d}{2}}\frac{\sigma_\gamma q\sqrt{\log(1/\delta)}}{BK(\nu)}, \sqrt{\frac{\pi}{2}}\frac{Cq\sqrt{\log(1/\delta)}}{B}\right\}. \tag{156}$$

*Then*

$$\mathbb{E}\left[\|\nabla f(X_{\tilde{t}})\|_2^2\right] \leq K_4\left(\frac{f(X_0)}{\eta\sqrt{T}} + \frac{\eta dL\sqrt{T}}{2}\right)\frac{1}{\varepsilon}, \tag{157}$$

*where $\tilde{t} \sim \mathrm{Unif}(0,\tau)$.*

**Remark B.4** *By construction we have $K_4 \leq K_3$, so Theorem B.14 provides a formally tighter upper bound than Theorem B.13. However, note that the first term in the definitions of $K_3$ and $K_4$ (Equations 145 and 156, respectively) scales as $\sqrt{d}$. Since $d$ is assumed to be large, this term typically dominates the maximum in both constants. As a consequence, in the high-dimensional regime of interest we effectively have $K_3 = K_4$, and the improvement from the mixed-phase analysis is negligible in practice.*

**Proof:** We divide the proof into two steps: i) SDE derivation, ii) gradient bound.

• Step 1: SDE derivation. Write one update of `DP-SignSGD` in the mixed regime as

$$x_{k+1} = x_k - \eta\, S_k, \qquad S_k := \text{sign}\left(G_k + \frac{1}{B}Z_{\text{DP}}\right), \tag{158}$$

where

$$G_k := \frac{1}{B}\sum_{\xi\in\gamma_k}\mathcal{C}[\nabla f_\xi(x_k)], \qquad Z_{\text{DP}} \sim \mathcal{N}(0, C^2\sigma_{DP}^2 I_d). \tag{159}$$

Conditioned on $\gamma_k$, each coordinate of $G_k + \frac{1}{B}Z_{\text{DP}}$ is Gaussian with mean $(G_k)_i$ and variance $C^2\sigma_{DP}^2/B^2$, hence

$$\mathbb{E}_{DP}[(S_k)_i \,|\, \gamma_k] = \text{Erf}\left(\frac{B}{C\sigma_{DP}\sqrt{2}}(G_k)_i\right). \tag{160}$$

Collecting the coordinates and taking expectation over $\gamma_k$ gives

$$\mathbb{E}[x_{k+1} - x_k] = -\eta\,\mathbb{E}\left[\text{Erf}\left(\frac{B}{C\sigma_{DP}\sqrt{2}}G_k\right)\right] =: \eta\, b_{\text{mix}}(x_k). \tag{161}$$

For the second moment, define

$$\bar{\Sigma}_{\text{mix}}(x) := \text{Cov}_{\gamma,DP}(S_k) \quad \text{with } x = x_k. \tag{162}$$

Then

$$\text{Cov}(x_{k+1} - x_k) = \eta^2\bar{\Sigma}_{\text{mix}}(x_k). \tag{163}$$

Since $S_k \in \{\pm 1\}^d$, we have $\text{Tr}(\bar{\Sigma}_{\text{mix}}(x)) \leq d$. Therefore, by Lem B.3 and Thm. B.4 we obtain the SDE

$$dX_t = b_{\text{mix}}(X_t)\, dt + \sqrt{\eta}\,\bar{\Sigma}_{\text{mix}}(X_t)^{1/2}dW_t. \tag{164}$$

• Step 2: Gradient bound. As argued in the proof of Corollary B.11, $\left|\frac{B}{C\sigma_{DP}\sqrt{2}}G(x)\right| \ll 1$ (componentwise) without loss of generality. Using $\text{Erf}(z) \sim \frac{2}{\sqrt{\pi}}z$ in a neighborhood of 0, we obtain the linearized drift

$$b_{\text{mix}}(x) \approx -\sqrt{\frac{2}{\pi}}\frac{B}{C\sigma_{DP}}\,\mathbb{E}[G(x)]. \tag{165}$$

In the mixed regime, we can write (for some $p_k \in [0,1]$ encoding the expected clipping rate at $x_k$)

$$\mathbb{E}[G(x)] = p_k\, C\,\mathbb{E}_\xi\left[\frac{\nabla f_\xi(x)}{\|\nabla f_\xi(x)\|_2}\right] + (1 - p_k)\,\mathbb{E}_\xi[\nabla f_\xi(x)]. \tag{166}$$

Using Lemma A.2 for the clipped component and unbiasedness for the unclipped component yields

$$\nabla f(x)^\top\mathbb{E}[G(x)] = \left(p_k\frac{CK(\nu)}{\sigma_\gamma\sqrt{d}} + (1 - p_k)\right)\|\nabla f(x)\|_2^2. \tag{167}$$

Therefore, applying the Itô formula and using $\text{Tr}(\nabla^2 f(X_t)\bar{\Sigma}_{\text{mix}}(X_t)) \leq L\,\text{Tr}(\bar{\Sigma}_{\text{mix}}(X_t)) \leq Ld$, we have

$$df(X_t) \leq -\sqrt{\frac{2}{\pi}}\frac{B}{C\sigma_{DP}}\nabla f(X_t)^\top\mathbb{E}[G(X_t)]\, dt + \frac{\eta}{2}\,\text{Tr}\left(\nabla^2 f(X_t)\bar{\Sigma}_{\text{mix}}(X_t)\right) dt + \mathcal{O}(\text{Noise})$$

$$\leq -(p_k a_1 + (1 - p_k)a_2)\,\|\nabla f(X_t)\|_2^2\, dt + \frac{\eta dL}{2}dt + \mathcal{O}(\text{Noise}), \tag{168}$$

where $a_1 = \sqrt{\frac{2}{\pi}}\frac{B}{\sigma_{DP}}\frac{K(\nu)}{\sigma_\gamma\sqrt{d}}$ and $a_2 = \sqrt{\frac{2}{\pi}}\frac{B}{C\sigma_{DP}}$.

Expanding $\sigma_{DP}$ and defining $K_4$ as in Eq. 156, we have

$$\varepsilon\sqrt{T^{-1}}K_4^{-1} \leq p_k a_1 + (1 - p_k)a_2, \qquad \forall p_k \in [0,1]. \tag{169}$$

Thus

$$df(X_t) \leq -\varepsilon\sqrt{T^{-1}}K_4^{-1}\|\nabla f(X_t)\|_2^2\, dt + \frac{\eta dL}{2}dt + \mathcal{O}(\text{Noise}). \tag{170}$$

Taking expectations and integrating over $[0, \tau]$ gives

$$\mathbb{E} \int_0^\tau \|\nabla f(X_t)\|_2^2 \, dt \leq \varepsilon^{-1} \sqrt{T} \, K_4 \left( f(X_0) + \frac{\eta dL\tau}{2} \right) \tag{171}$$

$$\implies \mathbb{E} \int_0^\tau \frac{1}{\tau} \|\nabla f(X_t)\|_2^2 dt \leq \frac{K_4}{\eta \sqrt{T}} \left( f(X_0) + \frac{\eta dL\tau}{2} \right) \varepsilon^{-1} \tag{172}$$

$$\implies \mathbb{E} \left[ \|\nabla f(X_{\tilde{t}})\|_2^2 \right] \leq K_4 \left( \frac{f(X_0)}{\eta \sqrt{T}} + \frac{\eta dL\sqrt{T}}{2} \right) \frac{1}{\varepsilon}, \tag{173}$$

where in the last step we used the Law of the Unconscious Statistician and $\tilde{t} \sim \text{Unif}(0, \tau)$.

$\square$

Finally, we derive the stationary distribution of `DP-SignSGD`: We empirically validate it in Fig. C.6.

**Theorem B.15** *Let $f(x) = \frac{1}{2} x^\top H x$ where $H = \text{diag}(\lambda_1, \dots, \lambda_d)$. The stationary distribution of Phase 2 is*

$$\mathbb{E}[X_T] = X_0 e^{-KH\tau}; \tag{174}$$

$$\text{Cov}(X_T) = X_0^2 e^{-2KH\tau} \left( e^{-\eta K^2 H\tau} - 1 \right) \tag{175}$$

$$+ \eta \left( 2KH + \eta H^2 K^2 \right)^{-1} \left( 1 - e^{-(2KH + \eta K^2 H^2)\tau} \right) \tag{176}$$

*where $K = \sqrt{\frac{2}{\pi}} \frac{1}{\sqrt{\varepsilon^2 B^{-1} \sigma_\gamma^2 + B^{-2} C^2 q^2 \log(1/\delta)T}} \varepsilon$.*

**Proof:** Since $H$ is diagonal, we can work component-wise. Let us remember the SDE:

$$dX_{t,i} = -\sqrt{\frac{2}{\pi}} \frac{1}{\sqrt{\frac{C^2 \sigma_{DP}^2}{B^2} + \frac{\sigma_\gamma^2}{B}}} \lambda_i X_{t,i} dt + \sqrt{\eta} \sqrt{1 - \frac{2}{\pi \left( \frac{C^2 \sigma_{DP}^2}{B^2} + \frac{\sigma_\gamma^2}{B} \right)} \lambda_i^2 X_{t,i}^2} dW_t. \tag{177}$$

To ease the notation, we write $K = \sqrt{\frac{2}{\pi}} \frac{1}{\sqrt{\varepsilon^2 B^{-1} \sigma_\gamma^2 + B^{-2} C^2 q^2 \log(1/\delta)T}} \varepsilon$. Hence, we can write $X_{t,i}$ in closed form as

$$X_{t,i} = x_{0,i} e^{-K\lambda_i t} + \sqrt{\eta} \int_0^t e^{-K\lambda_i (t-s)} \sqrt{1 - K^2 \lambda_i^2 X_{t,i}^2} dW_s. \tag{178}$$

Due to the properties of the stochastic integral, we immediately have

$$\mathbb{E}[X_{t,i}] = X_{0,i} e^{-\sqrt{\frac{2}{\pi}} \frac{1}{\sqrt{\varepsilon^2 B^{-1} \sigma_\gamma^2 + B^{-2} C^2 q^2 \log(1/\delta)T}} \varepsilon \lambda_i t}. \tag{179}$$

Using the Itô formula on $g(x) = x^2$, we have

$$d\left(X_{t,i}^2\right) = -2K\lambda_i X_{t,i}^2 dt + \frac{\eta}{2} 2 dt - \frac{\eta}{2} 2K^2 \lambda_i^2 X_{t,i}^2 dt + \mathcal{O}(\text{Noise}) \tag{180}$$

$$\implies \mathbb{E}[X_{t,i}^2] = X_{0,i}^2 e^{-(2K\lambda_i + \eta K^2 \lambda_i^2)t} + \frac{\eta}{2K\lambda_i + \eta \lambda_i^2 K^2} \left( 1 - e^{-(2K\lambda_i + \eta K^2 \lambda_i^2)t} \right), \tag{181}$$

therefore

$$\text{Cov}(X_{t,i}) = \mathbb{E}[X_{t,i}^2] - \mathbb{E}[X_{t,i}]^2 \tag{182}$$

$$= X_{0,i}^2 e^{-(2K\lambda_i + \eta K^2 \lambda_i^2)t} + \frac{\eta}{2K\lambda_i + \eta \lambda_i^2 K^2} \left( 1 - e^{-(2K\lambda_i + \eta K^2 \lambda_i^2)t} \right) - X_{0,i}^2 e^{-2K\lambda_i t}$$

$$= X_{0,i}^2 e^{-2K\lambda_i t} \left( e^{-\eta K^2 \lambda_i^2 t} - 1 \right) + \frac{\eta}{2K\lambda_i + \eta \lambda_i^2 K^2} \left( 1 - e^{-(2K\lambda_i + \eta K^2 \lambda_i^2)t} \right). \tag{183}$$

$\square$

Finally, we present a result that allows us to determine which of `DP-SignSGD` and `DP-SGD` is more advantageous depending on the training setting.

**Corollary B.16** *If $\frac{\sigma_\gamma^2}{B} \geq 1$, then* DP-SignSGD *always achieves a better privacy-utility trade-off than* DP-SGD, *though its convergence is slower. If $\frac{\sigma_\gamma^2}{B} < 1$, there exists a critical privacy level*

$$\varepsilon^\star = \sqrt{\frac{C^2 TB}{n^2 \left(B - \sigma_\gamma^2\right)} \log\left(\frac{1}{\delta}\right)}, \tag{184}$$

*such that* DP-SignSGD *outperforms* DP-SGD *in utility whenever $\varepsilon < \varepsilon^\star$, but still converges more slowly than* DP-SGD.

**Proof:** The Phase 2 asymptotic terms at $t = T$ are

$$A_{\text{SGD}} = \frac{T\eta dL}{\mu}\left(\frac{\varepsilon^2 \sigma_\gamma^2}{TB} + C^2 \frac{\Phi^2}{B^2}\right)\frac{1}{\varepsilon^2}, \qquad A_{\text{Sign}} = \frac{\sqrt{T}\eta dL}{\mu}\sqrt{\frac{\varepsilon^2 \sigma_\gamma^2}{TB} + C^2 \frac{\Phi^2}{B^2}}\,\frac{1}{\varepsilon}. \tag{185}$$

We compare $A_{\text{Sign}} < A_{\text{SGD}}$. Cancelling the common factor $\frac{\eta dL}{\mu}$ gives

$$\frac{\sqrt{T}}{\varepsilon}\sqrt{\frac{\varepsilon^2 \sigma_\gamma^2}{TB} + C^2 \frac{\Phi^2}{B^2}} \;<\; \frac{T}{\varepsilon^2}\left(\frac{\varepsilon^2 \sigma_\gamma^2}{TB} + C^2 \frac{\Phi^2}{B^2}\right). \tag{186}$$

Multiplying by $\varepsilon^2$ and dividing by the positive square root yields

$$\varepsilon\sqrt{T} \;<\; T\sqrt{\frac{\varepsilon^2 \sigma_\gamma^2}{TB} + C^2 \frac{\Phi^2}{B^2}}. \tag{187}$$

All quantities are non-negative, so squaring preserves the inequality:

$$\varepsilon^2 T \;<\; T^2\left(\frac{\varepsilon^2 \sigma_\gamma^2}{TB} + C^2 \frac{\Phi^2}{B^2}\right) \iff \left(1 - \frac{\sigma_\gamma^2}{B}\right)\varepsilon^2 \;<\; C^2 \frac{\Phi^2}{B^2}T. \tag{188}$$

Using $\frac{\Phi}{B} = \frac{1}{n}\sqrt{\log(1/\delta)}$ gives

$$\left(1 - \frac{\sigma_\gamma^2}{B}\right)\varepsilon^2 \;<\; \frac{C^2}{n^2}T\log\left(\frac{1}{\delta}\right). \tag{189}$$

If $\frac{\sigma_\gamma^2}{B} \geq 1$, the left coefficient is non-positive and the inequality holds for all $\varepsilon > 0$. If $\frac{\sigma_\gamma^2}{B} < 1$, solving for $\varepsilon$ yields

$$\varepsilon < \sqrt{\frac{C^2 TB}{n^2 \left(B - \sigma_\gamma^2\right)} \log\left(\frac{1}{\delta}\right)} = \varepsilon^\star, \tag{190}$$

which proves the claim. $\qquad\square$

Interestingly, by keeping $\eta$ and $C$ depend on the optimizer, we get

$$\sqrt{T}\eta_{\text{sign}}\sqrt{\frac{\sigma_\gamma^2}{BT} + \frac{C_{\text{sign}}^2 \Phi^2}{B^2 \varepsilon^2}} \;<\; T\eta_{\text{sgd}}\left(\frac{\sigma_\gamma^2}{BT} + \frac{C_{\text{sgd}}^2 \Phi^2}{B^2 \varepsilon^2}\right). \tag{191}$$

We observe that if $\sigma_\gamma \to \infty$, DP-SignSGD is always better than DP-SGD, while if $\sigma_\gamma \to 0$, there is always a threshold $\varepsilon^\star$. Since the algebraic expressions are complex, we believe this is enough to show that our insight is much more general than the case derived here and presented in the main paper.

## C  EXPERIMENTAL DETAILS AND ADDITIONAL RESULTS

Our empirical analysis is based on the official GitHub repository https://github.com/kenziyuliu/DP2 released with the Google paper (Li et al., 2023). In particular we consider the two following classification problems:

**IMDB** (Maas et al., 2011) is a sentiment analysis dataset for movie reviews, posed as a binary classification task. It contains 25,000 training samples and 25,000 test samples, with each review represented using a vocabulary of 10,000 words. We train a logistic regression model with 10,001 parameters.

**StackOverflow** (Kaggle, 2022), (TensorFlow Federated, 2022) is a large-scale text dataset derived from Stack Overflow questions and answers. Following the setup in (TensorFlow Federated, 2022), we consider the task of predicting the tag(s) associated with a given sentence, but we restrict our experiments to the standard centralized training setting rather than the federated one. We randomly select 246,092 sentences for training and 61,719 for testing, each represented with 10,000 features. The task is cast as a 500-class classification problem, yielding a model with approximately 5 million parameters.

**MovieLens-100k** (Harper & Konstan, 2015) is a widely used movie-rating dataset for recommendation systems. It consists of 100,000 ratings provided by 943 users across 1,682 movies ($\approx 6\%$ non-zero entries). We consider a non-convex matrix factorization problem with an embedding dimension of 100, resulting in 262,500 parameters. Each observed rating is treated as an individual "record" for differential privacy, and the non-zero entries are randomly split into training and evaluation sets.

**Optimizers.** We train both classification problems using `DP-SGD`, `DP-SignSGD` and `DP-Adam`. For $k \geq 0$, learning rate $\eta$, variance $\sigma_{\text{DP}}^2$, and batches $\gamma_k$ of size $B$ modeled as i.i.d. uniform random variables taking values in $\{1, \ldots, n\}$. Let $g_k$ be the private gradient, defined as

$$g_k := \frac{1}{B} \sum_{i \in \gamma_k} \mathcal{C}[\nabla f_i(x_k)] + \frac{1}{B} \mathcal{N}(0, C^2 \sigma_{\text{DP}}^2 I_d) \tag{192}$$

and $\mathcal{C}[\cdot]$ be the clipping function

$$\mathcal{C}[x] = \min\left\{\frac{C}{\|x\|_2}, 1\right\} x. \tag{193}$$

The iterates of `DP-SGD` are defined as

$$x_{k+1} = x_k - \eta g_k, \tag{194}$$

while those of `DP-SignSGD` are defined as

$$x_{k+1} = x_k - \eta \operatorname{sign}[g_k], \tag{195}$$

where $\operatorname{sign}[\cdot]$ is applied component-wise. The update rule of `DP-Adam` is defined as follows:

$$\begin{aligned}
m_{k+1} &= \beta_1 m_k + (1 - \beta_1) g_k, & \hat{m}_{k+1} &= \frac{m_{k+1}}{1 - \beta_1^{k+1}}, \\
v_{k+1} &= \beta_2 v_k + (1 - \beta_2)(g_k \odot g_k), & \hat{v}_{k+1} &= \frac{v_{k+1}}{1 - \beta_2^{k+1}}, \\
x_{k+1} &= x_k - \eta \frac{\hat{m}_{k+1}}{\sqrt{\hat{v}_{k+1}} + \epsilon},
\end{aligned} \tag{196}$$

where $g_k$ is the privatized stochastic gradient and is defined in Equation 192.

**Hyper-parameters.** Unless stated otherwise, we fix the following hyperparameters in our experiments: for IMDB, StackOverflow and MovieLens respectively, we train for $100, 50, 50$ epochs with batch size $B = 64$. The choice of batch size follows the setting in (Li et al., 2023). We also aimed to avoid introducing unnecessary variability, keeping the focus on the direction suggested by our theoretical results. Finally, we set $\delta = 10^{-5}, 10^{-6}, 10^{-6}$, corresponding to the rule $\delta = 10^{-k}$, where $k$ is the smallest integer such that $10^{-k} \leq 1/n$ for the training dataset size $n$.

**Protocol A.** We perform a grid search on *learning rate* $\eta = \{0.001, 0.01, 0.1, 1, 3, 5, 10\}$ and *clipping threshold* $C = \{0.1, 0.25, 0.5, 1, 2, 3, 5\}$ for `DP-SGD`, `DP-SignSGD` and `DP-Adam` on both datasets, using $\sigma_{DP} = 1$: this gives $\varepsilon = 2.712$ and $\varepsilon = 0.424$ for IMDB and StackOverflow respectively. For MovieLens we use the same hyperparameters grid, but for the noise multiplier we use instead $\sigma_{DP} = 0.5$, which yields $\varepsilon = 1.329$. We summarize the best set of hyperparameters for each method on both datasets in Table C.1.

| Dataset | DP-SGD | DP-SignSGD | DP-Adam |
|---|---|---|---|
| IMDB | (5, 0.5) | (0.1, 0.5) | (0.1, 0.5) |
| StackOverflow | (3, 0.25) | (0.01, 0.5) | (0.01, 0.5) |
| MovieLens | (0.1, 2) | (0.001, 3) | (0.001, 1) |

Table C.1: Tuned hyperparameters for different methods across the two datasets. The values refer to (learning rate, clipping parameter); For DP-Adam we also used $\beta_1 = 0.9, \beta_2 = 0.999$ and adaptivity $\epsilon = 10^{-8}$ in both cases.

**Protocol B.** For each noise multiplier, we tune a new pair of learning rate and clipping parameter by performing a grid search. **IMDB**: For DP-SignSGD and DP-Adam, we consider the following learning rates $\eta = \{0.01, 0.05, 0.10, 0.15, 0.22, 0.27, 0.33, 0.38, 0.44, 0.50\}$ and clipping thresholds $C = \{0.05, 0.1, 0.25, 0.5\}$, while for DP-SGD we consider a different range of learning rates $\eta = \{0.5, 0.7, 1.0, 1.5, 2.0, 2.5, 3.0, 3.5, 4.0, 4.5, 5.0, 5.5, 6.0\}$ and $C = \{0.1, 0.25, 0.5\}$. This tuning is designed to identify the best hyperparameters across a broad range of privacy budgets $\varepsilon = \{0.01, 0.2, 0.4, 0.6, 0.8, 1.0, 1.2, 1.4, 1.6, 1.8, 2.0\}$, which correspond to the following noise multipliers: $\{271.23, 13.56, 6.78, 4.52, 3.39, 2.71, 2.26, 1.94, 1.70, 1.51, 1.36\}$. **StackOverflow**: For DP-Adam we consider the following learning rates $\{0.001, 0.003, 0.005, 0.01, 0.03, 0.05, 0.1, 0.5\}$, for DP-SignSGD we add $\{0.008, 0.015, 0.02, 0.03, 0.04\}$ to the list, while for DP-SGD we consider a different range of learning rates $\eta = \{0.1, 0.5, 1.0, 2.0, 2.5, 3.0, 3.5, 4.0, 5.0\}$. For the clipping thresholds we consider $C = \{0.05, 0.1, 0.25, 0.35, 0.5, 1.0\}$ for every method. This tuning is designed to identify the best hyperparameters across a broad range of privacy budgets $\varepsilon = \{0.01, 0.2, 0.4, 0.6, 0.8, 1.0, 1.2, 1.4\}$, which correspond to the following noise multipliers: $\{42.384, 2.119, 1.060, 0.706, 0.530, 0.424, 0.353, 0.303\}$. **Movielens**: For DP-SignSGD and DP-Adam, we consider the following learning rates $\eta = \{0.0001, 0.0003, 0.001, 0.003, 0.01, 0.03, 0.1, 0.3, 1, 3\}$ and clipping thresholds $C = \{0.1, 0.25, 0.5, 1.0, 2.0, 3.0\}$, while for DP-SGD we consider a different range of learning rates $\eta = \{0.01, 0.03, 0.05, 0.1, 0.3, 0.5, 1\}$ and $C = \{0.1, 0.25, 0.5, 1.0, 2.0, 3.0\}$. This tuning is designed to identify the best hyperparameters across a broad range of privacy budgets $\varepsilon = \{0.01, 0.05, 0.1, 0.2, 0.5, 1\}$, which correspond to the following noise multipliers: $\{66.48, 13.30, 6.65, 3.32, 1.33, 0.66\}$.

## C.1 DP-SGD AND DP-SignSGD: SDE VALIDATION (FIGURE C.1).

In this section, we describe how we validated the SDE models derived in Theorem B.5 and Theorem B.10 (Figure C.1). In line with works in the literature Monzio Compagnoni et al. (2025c;b), we optimize a quadratic and a quartic function. We run both DP-SGD and DP-SignSGD, calculating the full gradient and injecting noise as described in Assumption B.2. Similarly, we integrate our SDEs using the Euler-Maruyama algorithm (See, e.g., (Monzio Compagnoni et al., 2025c), Algorithm 1) with $\Delta t = \eta$. Results are averaged over 200 repetitions. For each of the two functions, the details are presented in the following paragraphs.

**Quadratic function**: We consider the quadratic function $f(x) = \frac{1}{2}x^\top H x$, with $H = 0.1 \, \mathrm{diag}(2, 1, \ldots, 1)$, in dimension $d = 1024$. The clipping parameter is set to $C = 5$, and each algorithm is run for $T = 10000$ iterations. The gradient noise scale is $\sigma_\gamma = 1/\sqrt{d}$. The learning rate is $\eta = 0.1$ for DP-SGD and $\eta = 0.01$ for DP-SignSGD. The differential privacy parameters are $(\varepsilon, \delta, q) = (5, 10^{-4}, 10^{-4})$, corresponding to a noise multiplier of $\sigma_{DP} = 0.03$. The initial point is sampled as $x_0 = \frac{50}{\sqrt{d}}\mathcal{N}(0, I_d)$, using an independent seed for each method. In this experiment and in all the following where we use a Student's $t$, we use $\nu = 1$.

**Quartic function**: We also test on the quartic function $f(x) = \frac{1}{2}\sum_{i=0}^{d-1} H_{ii}x_i^2 + \frac{\lambda}{4}\sum_{i=0}^{d-1} x_i^4 - \frac{\xi}{3}\sum_{i=0}^{d-1} x_i^3$, where $H = \mathrm{diag}(-2, 1, \ldots, 1)$, $\lambda = 0.5$, and $\xi = 0.1$. The problem dimension, clipping, and number of iterations are the same: $d = 1024, C = 5, T = 10000$, with gradient noise $\sigma_\gamma = 1/\sqrt{d}$. Both methods use a learning rate of $\eta = 0.01$. The differential privacy parameters are $(\varepsilon, \delta, q) = (5, 10^{-4}, 10^{-4})$ for DP-SGD and $(5, 10^{-4}, 2 \times 10^{-4})$ for DP-SignSGD, corresponding

to noise multipliers $\sigma_{DP} = 0.03$ and $\sigma_{DP} = 0.06$, respectively. Initialization is $x_0 = \frac{50}{\sqrt{d}}\mathcal{N}(0, I_d)$ for DP-SGD and $y_0 = -x_0$ for DP-SignSGD.

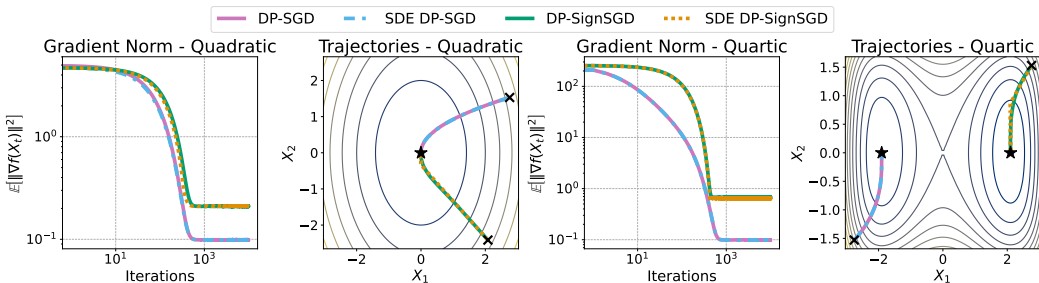

Figure C.1: Consistent with Theorem B.5 and Theorem B.10, we empirically validate that the SDEs of DP-SGD and DP-SignSGD model their respective optimizers. For a convex quadratic function (**left two panels**) and a nonconvex quartic function (**right two panels**), the SDEs accurately track both the trajectories and the gradient norm of the corresponding algorithms, averaged over 200 runs.

## C.2 ASYMPTOTIC LOSS BOUND (FIGURES 1, C.2 AND C.7)

This section refers to Figure 1, Figure C.2 and Figure C.7. We consider three different scenarios: A quadratic function, IMDB, StackOverflow and MovieLens. Each setup is optimized using DP-SGD, DP-SignSGD, and DP-Adam, and we plot the final averaged training loss across a range of privacy levels. In the left panel, we include the exact bounds from Theorem 4.1 and Theorem 4.3 to show agreement with theory; in the central and right panels, we compare the final losses with the trends in $\varepsilon$ predicted by the same theorems. Experimental details are as follows.

**Quadratic**: $f(x) = \frac{1}{2}x^\top H x$, $H = 10I_d$; $d = 1024$, $C = 5$, $T = 50000$, $\sigma_\gamma = 0.01$; learning rate $\eta = 0.01 \cdot \eta_t$ with $\eta_t = (1 + \eta t)^{-0.6}$ (a decaying schedule for stability; our theory assumes constant $\eta$, so we use this toy experiment for qualitative validation); Adam parameters: $\beta_1 = 0.9, \beta_2 = 0.999, \epsilon = 10^{-8}$. We used 8 noise multipliers, linearly spaced from 0 to 2, which with $q = 10^{-4}, \delta = 10^{-4}$ correspond to $\varepsilon \in \{\infty, 6.78, 2.38, 1.19, 0.79, 0.59, 0.48, 0.40, 0.34\}$.

**IMDB**: Hyperparameters are given in Table C.1. We performed 10 runs for each noise multiplier $\{0.5, 1.0, 2.0, 4.0, 6.0, 8.0, 10.0, 12.0\}$, yielding the following values for $\varepsilon$ $\{5.425, 2.712, 1.356, 0.678, 0.452, 0.339, 0.271, 0.226\}$, respectively. We report the average training and test loss of the final epoch with confidence bounds (Figure 1 and Figure C.7).

**StackOverflow**: Hyperparameters are given in Table C.1. We performed 3 runs using for each noise multipliers $\{0.1, 0.3, 0.5, 1.0, 2.0, 4.0, 6.0, 8.0\}$, yielding the following values for $\varepsilon$ $\{4.238, 1.413, 0.848, 0.424, 0.212, 0.106, 0.071, 0.053\}$, respectively. We report the average training and test loss of the final epoch with confidence bounds (Figure 1 and Figure C.7).

**MovieLens**: Hyperparameters are given in Table C.1. We performed 10 runs for each noise multiplier $\{0.06, 0.08, 0.1, 0.2, 0.3, 0.5, 1.0, 1.5\}$, yielding the following values for $\varepsilon$ $\{11.079, 8.310, 6.648, 3.324, 2.216, 1.329, 0.665, 0.443\}$, respectively. We report the average training and test loss of the final epoch with confidence bounds (Figure C.2).

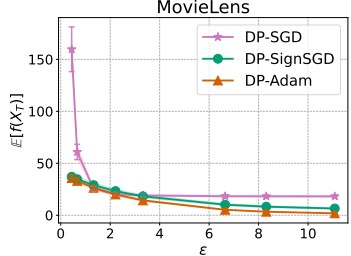

Figure C.2: Empirical validation of the privacy-utility trade-off predicted by Thm. 4.1 and Thm. 4.3, comparing `DP-SGD`, `DP-SignSGD`, and `DP-Adam`: Our focus is on verifying the functional dependence of the asymptotic loss levels in terms of $\varepsilon$. Train MSE for matrix factorization on MovieLens-100k, confirm the same pattern: the utility of `DP-SGD` scales as $\frac{1}{\varepsilon^2}$, while the utility of `DP-SignSGD` scales linearly as $\frac{1}{\varepsilon}$. We observe that the insights obtained for `DP-SignSGD` extend to `DP-Adam` as well as to the test loss.

### C.3  Convergence Speed Analysis (Figure 2 and C.3)

This section refers to Figure 2 and Figure C.3. We consider three different scenarios: IMDB, Stack-Overflow and MovieLens. Each setup is optimized using `DP-SGD`, `DP-SignSGD`, and `DP-Adam` and six different privacy levels: We plot the average trajectories of the training losses and observe that, when it converges, the convergence speed of `DP-SGD` does not depend on the level of privacy, while the two adaptive methods are more resilient to the demands of high levels of privacy, but their convergence speed changes for every $\varepsilon$, as predicted in Theorem 4.3.

**IMDB**: Hyperparameters are given in Table C.1. We performed 10 runs for each noise multiplier $\{0.8, 1.0, 1.2, 1.6, 4.0, 6.0\}$ and corresponding epsilons $\{3.390, 2.712, 2.260, 1.695, 0.678, 0.452\}$. We report the average trajectories of the training loss with confidence bounds (Figure 2).

**StackOverflow**: Hyperparameters are given in Table C.1. We performed 3 runs for each noise multiplier $\{0.37, 0.5, 0.64, 1.19, 1.46, 1.73\}$ and corresponding epsilons $\{1.146, 0.848, 0.662, 0.356, 0.290, 0.245\}$. We report the average trajectories of the training loss with confidence bounds (Figure 2).

**MovieLens**: Hyperparameters are given in Table C.1. We performed 10 runs for each noise multiplier $\{0.06, 0.1, 0.2, 0.3, 0.5, 1.0\}$ and corresponding epsilons $\{11.080, 6.648, 3.324, 2.216, 1.330, 0.665\}$. We report the average trajectories of the training loss with confidence bounds (Figure C.3).

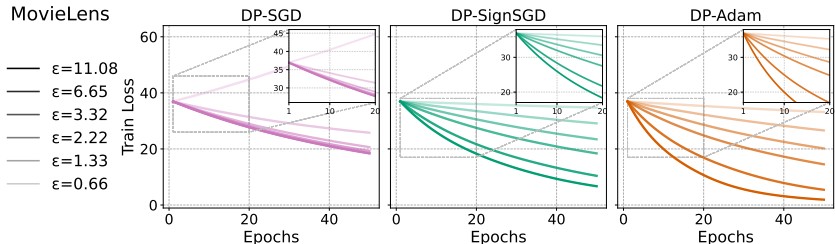

Figure C.3: Empirical validation of the convergence speeds predicted by Thm. 4.1 and Thm. 4.3. We compare `DP-SGD`, `DP-SignSGD`, and `DP-Adam` as we optimize a non-convex matrix factorization problem on the MovieLens dataset. We verify that when `DP-SGD` converges, its speed is unaffected by $\varepsilon$. As expected, it diverges when $\varepsilon$ is too small. Regarding `DP-SignSGD` and `DP-Adam`, they are faster when $\varepsilon$ is large and never diverge even when this is small.

## C.4 WHEN ADAPTIVITY REALLY MATTERS (FIGURE 3 AND FIGURE C.4)

This section refers to Figure 3 and Figure C.4. Each setup is optimized using `DP-SGD`, `DP-SignSGD`, and `DP-Adam`. We consider different batch sizes and for each we plot the final loss values for different privacy levels, similarly to Section C.2. We highlight the possible range of $\varepsilon^\star$ and a dash-dotted line to mark its approximate value, suggested by each graph. As predicted by Theorem 4.5, the empirical value of $\varepsilon^\star$ shifts left as we increase the batch size. Experimental details are as follows.

**IMDB**: Hyperparameters are given in Table C.1. We select a wide range of noise multipliers: $\{0.5, 1.0, 1.2, 1.5, 1.8, 2.0, 2.2, 2.5, 2.8, 3.0, 3.2, 3.5, 3.8, 4.0, 4.5, 5.0, 6.0, 8.0, 10.0, 12.0\}$ and increasing batch sizes $B = \{48, 56, 64, 72, 80\}$. The corresponding epsilons are

$B = 48$: $\{4.698, 2.349, 1.879, 1.566, 1.342, 1.174, 1.044, 0.940, 0.854, 0.783, 0.723, 0.671, 0.626, 0.587, 0.522, 0.470, 0.391, 0.294, 0.235, 0.196\}$;

$B = 56$: $\{5.070, 2.535, 2.028, 1.690, 1.449, 1.268, 1.127, 1.014, 0.922, 0.845, 0.780, 0.724, 0.676, 0.634, 0.563, 0.507, 0.423, 0.317, 0.254, 0.211\}$;

$B = 64$: $\{5.425, 2.712, 2.170, 1.808, 1.550, 1.356, 1.205, 1.085, 0.986, 0.904, 0.835, 0.775, 0.723, 0.678, 0.603, 0.542, 0.452, 0.339, 0.271, 0.226\}$;

$B = 72$: $\{5.740, 2.870, 2.296, 1.913, 1.640, 1.435, 1.276, 1.148, 1.044, 0.957, 0.883, 0.820, 0.765, 0.717, 0.638, 0.574, 0.478, 0.359, 0.287, 0.239\}$;

$B = 80$: $\{6.070, 3.035, 2.428, 2.023, 1.734, 1.517, 1.349, 1.214, 1.104, 1.012, 0.934, 0.867, 0.809, 0.759, 0.674, 0.607, 0.506, 0.379, 0.303, 0.253\}$.

For each batch size, we performed 10 runs and plotted the average final value of the Train Loss and the empirical $\varepsilon^\star$: these observed values follow the direction indicated in Thm. 4.5. For visualization purposes, we show only a smaller window of $\varepsilon$ values satisfying $0.75 \leq \varepsilon \leq 1.25$.

**StackOverflow**: Due to the higher computational cost required, with our limited resources we managed to select only a restricted range of noise multipliers: $\{0.1, 0.3, 0.5, 1.0, 2.0, 4.0, 6.0, 8.0\}$ and batch sizes: $\{48, 56, 64\}$. The corresponding epsilons are

$B = 48$: $\{1.223, 0.734, 0.367, 0.184, 0.092, 0.061, 0.046\}$;

$B = 56$: $\{1.322, 0.793, 0.396, 0.198, 0.099, 0.066, 0.050\}$;

$B = 64$: $\{1.413, 0.848, 0.424, 0.212, 0.106, 0.071, 0.053\}$.

For each batch size, we performed 3 runs and plotted the average final value of the Train Loss and the empirical $\varepsilon^\star$: these observed values follow the direction indicated in Thm. 4.5. For visualization purposes, we show only a smaller window of $\varepsilon$ values satisfying $0.15 \leq \varepsilon \leq 0.75$.

**MovieLens**: Hyperparameters are given in Table C.1. We select a wide range of noise multipliers: $\{0.15, 0.25, 0.35, 0.45, 0.55, 0.65, 0.75, 0.85, 0.95\}$ and batch sizes $\{56, 64, 72\}$. The corresponding epsilons are

$B = 56$: $\{4.146, 2.487, 1.777, 1.382, 1.131, 0.957, 0.829, 0.732, 0.655\}$;

$B = 64$: $\{4.432, 2.659, 1.899, 1.477, 1.209, 1.023, 0.886, 0.782, 0.700\}$;

$B = 72$: $\{4.700, 2.820, 2.014, 1.567, 1.282, 1.085, 0.940, 0.829, 0.742\}$.

For each batch size, we performed 10 runs and plotted the average final value of the Train Loss and the empirical $\varepsilon^\star$: these observed values follow the direction indicated in Thm. 4.5. For visualization purposes, we show only a smaller window of $\varepsilon$ values satisfying $0.75 \leq \varepsilon \leq 1.5$.

## C.5 BEST-TUNED HYPERPARAMETERS (FIGURES 4 AND C.5)

This section refers to Figure 4 and Figure C.5. On top of the hyperparameter sweep performed described in Section C, we additionally tune `DP-SGD` for the smaller values of $\varepsilon$. As predicted by Theorem 4.6, the optimal learning rate for `DP-SGD` scales with $\varepsilon$, while those of the adaptive methods are almost constant. Furthermore, we observe that once we reach the limits of the hyperparameter grid, `DP-SGD` loses performance drastically.

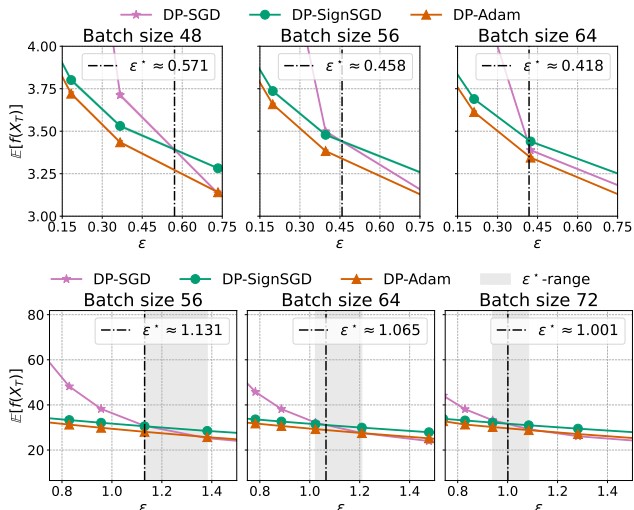

Figure C.4: (**Top Row**) StackOverflow and (**Bottom Row**) MovieLens: From left to right, we decrease the batch noise, i.e., increase the batch size, taking values $B = 48, 56, 64$: As per Theorem 4.5, the privacy threshold $\varepsilon^\star$ that determines when DP-SignSGD is more advantageous than DP-SGD shifts to the left. This confirms that if there is more noise due to the batch size, less privacy noise is needed for DP-SignSGD to be preferable over DP-SGD.

**IMDB**: We additionally tune DP-SGD using $\eta = \{0.001, 0.005, 0.01, 0.05, 0.1, 0.5\}$ and $C = \{0.1, 0.25, 0.5\}$ and add the corresponding values using the cyan line. On the left, we plot the average of the final 5 train loss values and confidence bound for each method against the privacy budget $\varepsilon$; On the right, we focus on the scaling of the optimal learning rate with respect to $\varepsilon$.

**StackOverflow**: We additionally tune DP-SGD using $\eta = \{0.001, 0.01, 0.05\}$ and add the corresponding values using the cyan line. As above, on the left, we plot the average of the final 5 training loss values and confidence bounds for each method against the privacy budget $\varepsilon$; on the right, we focus on the scaling of the optimal learning rate with respect to $\varepsilon$.

**MovieLens**: We additionally tune DP-SGD using $\eta = \{0.0001, 0.0003, 0.001, 0.003\}$ and add the corresponding values using the cyan line. On the left, we plot the average of the final 5 train loss values and confidence bound for each method against the privacy budget $\varepsilon$; On the right, we focus on the scaling of the optimal learning rate with respect to $\varepsilon$.

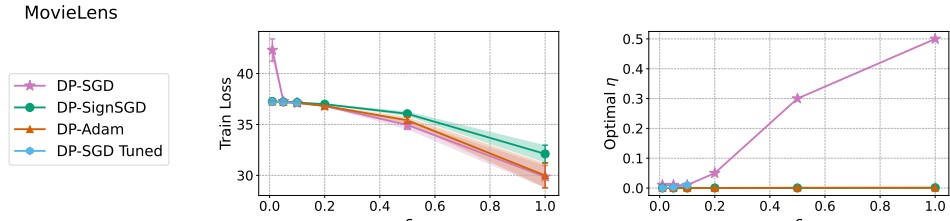

Figure C.5: Empirical verification of Thm. 4.6 and Thm. 4.7 under Protocol B on the MovieLens dataset. We tune $(\eta, C)$ of each optimizer for each $\varepsilon$ and confirm that: $i)$ all methods achieve comparable performance across privacy budgets; $ii)$ the optimal $\eta$ of DP-SGD scales linearly with $\varepsilon$, while that of adaptive methods is essentially $\varepsilon$-independent; $iii)$ failing to sweep over the "best" range of learning rates causes DP-SGD to severely underperform, whereas adaptive methods are resilient. On the **left**, DP-SGD degrades sharply for small $\varepsilon$. Indeed, the **right** panels shows that the selected optimal $\eta$ flattens out, while the theoretical one would have linearly decayed more: The "best" $\eta$ was simply missing from the grid. *A posteriori*, re-running the sweep with a larger grid (DP-SGD Tuned) recovers the scaling law and matches the performance of adaptive methods.

## C.6 STATIONARY DISTRIBUTIONS

In this paragraph, we describe how we validated the convergence behavior predicted in Theorem B.9 and Theorem B.15. To produce Figure C.6, we run both `DP-SGD` and `DP-SignSGD` on $f(x) = \frac{1}{2}x^\top H x$, where $H = \text{diag}(2, 1)$, $x_0 = (0.01, 0.005)$, $\eta = 0.001$, $\sigma_\gamma = \sigma_{DP} = 0.1$, $C = 5$. We average over 20000 runs and plot the evolution of the moments compared to the theoretical prediction provided in Theorem B.9 and Theorem B.15.

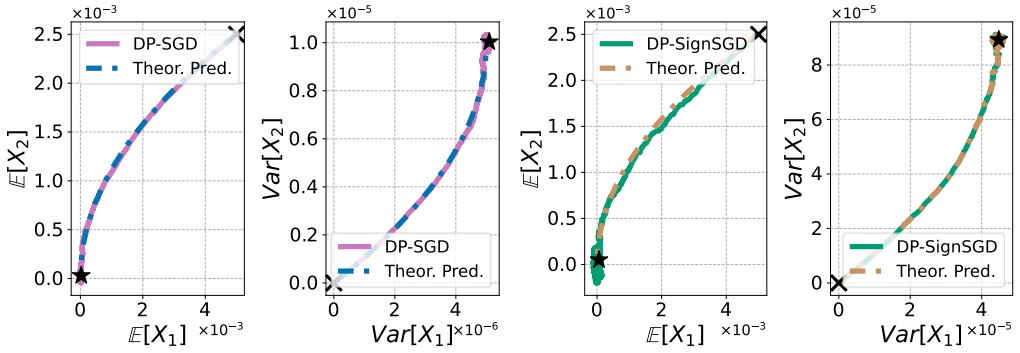

Figure C.6: The empirical dynamics of the first and second moments of the iterates $X_t$ of `DP-SGD` (left two panels) and of `DP-SignSGD` (right two panels) match that prescribed in Theorem B.9 and Theorem B.15, respectively.

## C.7 ADDITIONAL RESULTS — TEST LOSS

Interestingly, the insights provided in Theorem 4.1 and Theorem 4.3 regarding both the asymptotic bound and the convergence speed extend, in practice, also to the test loss. In the same set-up of Section C.2, we plot the asymptotic values of the Test Loss and interpolate with $\mathcal{O}(1/\varepsilon)$ and $\mathcal{O}(1/\varepsilon^2)$ to show that they match the predicted scaling.

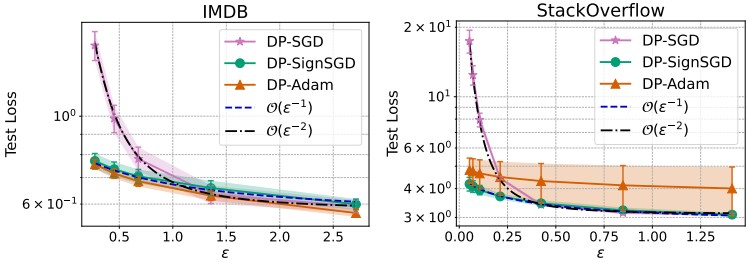

Figure C.7: Privacy-utility trade-off on the *test loss*, comparing `DP-SGD`, `DP-SignSGD`, and `DP-Adam`. **Left:** Logistic regression on the IMDB dataset. **Right:** Logistic regression on the StackOverflow dataset. In both cases, the empirical scalings predicted by Thm. 4.1 and Thm. 4.3 carry over from training to test: `DP-SGD` follows the $\frac{1}{\varepsilon^2}$ trend, while adaptive methods follow the $\frac{1}{\varepsilon}$ trend. This demonstrates that not only do our theoretical insights generalize to the widely used `DP-Adam`, but also extend from *training* to *test* loss.

Similarly, in the same set-up as Section C.3, we plot the trajectories of the Test Loss (Fig. C.8): we observe that once again the convergence speed of `DP-SGD` is not affected by the choice of $\varepsilon$, while adaptive methods clearly present different $\varepsilon$-dependent rates.

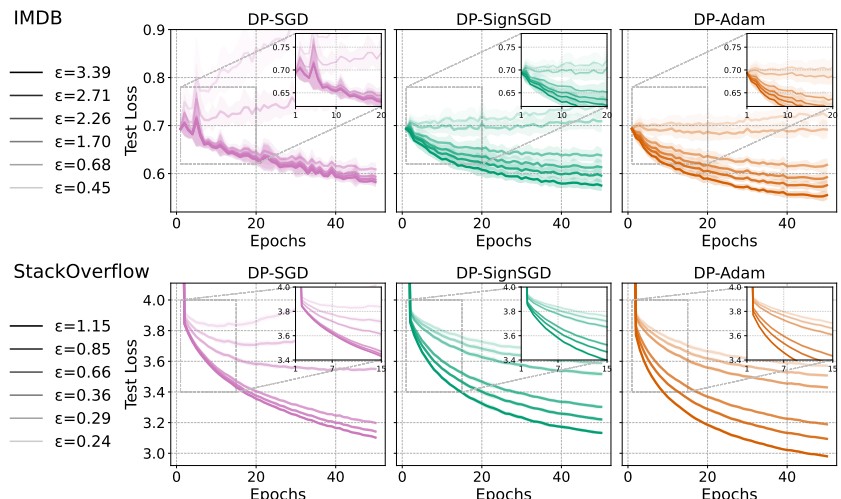

Figure C.8: We compare the Test Loss of `DP-SGD`, `DP-SignSGD`, and `DP-Adam` as we train a logistic regression on the IMDB dataset (**Top Row**) and on the StackOverflow dataset (**Bottom Row**).

# D LIMITATIONS

As highlighted by Li et al. (2021), the approximation capability of SDEs can break down when the learning rate $\eta$ is large or when certain regularity assumptions on $\nabla f$ and the noise covariance matrix are not fulfilled. Although such limitations can, in principle, be alleviated by employing higher-order weak approximations, our position is that the essential function of SDEs is to provide a simplified yet faithful description of the discrete dynamics that offers practical insight. We do not anticipate that raising the approximation order beyond what is required to capture curvature-dependent effects would deliver substantial additional benefits.

We stress that our SDE formulations have been thoroughly validated empirically: the derived SDEs closely track their corresponding optimizers across a wide range of architectures, including MLPs, CNNs, ResNets, and ViTs (Paquette et al., 2021; Malladi et al., 2022; Monzio Compagnoni et al., 2024; 2025b;c; Xiao et al., 2025; Marshall et al., 2025).

