# OpenReview forum: "Adaptive Methods Are Preferable in High Privacy Settings: An SDE Perspective"
_ICLR.cc/2026/Conference — ICLR 2026 Poster_

### Official Review · Reviewer_g3be · 2025-10-31

**Soundness:** 2
**Presentation:** 2
**Contribution:** 2
**Rating:** 4
**Confidence:** 4

**Summary:**

The main methodological contribution is to approximate the discrete optimization algorithm with a stochastic differential equation (SDE). It focuses on DP-SGD and DP-SignSGD under per-example clipping and provides the first theoretical framework that connects SDE with convergence dynamics.

**Strengths:**

The paper presents the first SDE-based theoretical framework for analyzing differentially private optimizers, bridging discrete training dynamics with continuous-time analysis. It provides clear scaling laws for DP-SGD $O(1/\epsilon^2)$ and DP-SignSGD $O(1/\epsilon)$, offering strong insight into how adaptivity interacts with privacy noise. Theoretical predictions are thoroughly validated by experiments across multiple datasets. The work also gives practical guidance for tuning and transferring DP optimizers under different privacy budgets.

**Weaknesses:**

While the SDE framework is well motivated, the paper does not clearly justify why the continuous approximation is valid or which small terms are neglected. A short discussion explaining that the discretization error between the algorithm and its SDE counterpart can be ignored would make this assumption more convincing. Additionally, although the authors argue that higher-order SDE approximations are unnecessary, a simple analysis or experiment illustrating how the first-order approximation deviates from the discrete dynamics would strengthen the theoretical rigor. Clarifying these points would improve both the transparency and completeness of the analysis.

**Questions:**

1.The paper could provide more heuristic insight into why DP-SignSGD, which can be seen as a post-processing of DP-SGD and thus should not improve efficiency, nevertheless achieves better empirical and theoretical utility. In particular, what property of the sign operation makes it more robust to DP noise?

2.Additionally, the role of the clipping bound C deserves clarification: while a larger C should intuitively reduce gradient bias, the theory suggests that a larger C moves the iterates further from the optimum.

3.The two-phase assumption (clipped vs. unclipped regime) is central to the analysis but not well justified; a short discussion of why it is valid to analyze the optimization dynamics separately in the clipped and unclipped phases would improve the theoretical clarity of the paper -- it should noted that in a single batch of samples, some gradient would be clipped and some would not but we will only have one update at the end. The analysis in the main theorem is confusing.

4.The paper also lacks comparison with SOTA DP-SGD benchmark, e.g., "unlocking high-accuracy differentially private image classification through scale", "a theory to instruct differentially-private learning via clipping bias reduction", and "differentially private image classification
by learning priors from random processes".

---

> ### Author Response · Authors · 2025-11-19
> **Many Thanks - Addressed Weaknesses and Questions**
>
> We sincerely thank the Reviewer g3be for their interesting review. We appreciate that the reviewer recognised the **novelty** of our analysis and how it offers 1) **"strong insight into how adaptivity interacts with privacy noise"**, 2) **"Theoretical predictions are thoroughly validated by experiments across multiple datasets"**, and 3) practical guidance for practitioners.
>
> We hope that the following clarifications will help re-evaluate our submission and potentially adjust the score.
>
>
> ## Weaknesses
>
> 1. **"The paper does not clearly justify why the continuous approximation is valid or which small terms are neglected"**
> We thank the reviewer for pointing out this: We will improve the presentation in our paper. This work follows the **standard weak-approximation framework** [1] used in prior analyses of SGD based on SDEs (see our Related Works). As stated in Definition 3.1 of the main paper, the error between the SDE and the discrete-time iterates scales with the learning rate $\eta$. The key ingredients **ensuring validity** are:
>
>    - **1.a Order-1 weak approximation:** Theorems B.5 and B.10 explicitly verify the conditions of Theorem B.4, showing that the first two moments of the discrete updates match those of the SDE up to **$O(\eta^2)$**; see Eqs. (25 and 31)–(32 and 33) and (109 and 124)–(125 126) for DP-SGD and DP-SignSGD, respectively. Therefore **we do formally guarantee** that our SDEs attain an $O(\eta)$-error w.r.t. the discrete time optimizer.
>
>    - **1.b Higher-order terms:** All neglected terms are of **higher-order $O(\eta^2)$** in the Taylor expansion of one-step increments, made explicit in Lemma B.3 (items 1–3) and in the proofs of Theorems B.5 and B.10. This is in line with the fact that we derive an **Order-1 weak approximation**.
>
>     Empirically, Figure C.1 (already present in the original submission) shows that the SDEs **faithfully reproduce the discrete dynamics** for both convex quadratic and nonconvex quartic functions, **empirically validating** that the order-1 approximation is accurate in the regimes we study. These **experiments** are in line with, or **more extensive** than, those found in most papers using SDEs to model optimizers: See Appendix A of [2] for a comprehensive literature review on this experimental aspect.
>
>    Thus, the continuous-time model is not simply assumed: it is **quantitatively derived, conceptually grounded, and rigorously justified**. It provides a **verified weak limit** whose validity we check **both analytically and empirically**.
>
>    Additionally, while the higher-order SDE for SGD was derived in [1], it has, to the best of our knowledge, **never found practical application** in the literature. We therefore believe the current level of approximation is both standard [4-6] and fully sufficient for the phenomena we analyze.
>
>    To conclude, we believe all the guarantees the Reviewer was asking for (both theoretical and experimental) are mentioned in our answer above. We are happy to answer any additional questions.
>
> ---
>
> ## Questions
>
> 1. **What is the role of the Sign operator?**
> This is an important conceptual question. By discarding gradient magnitudes, the sign operator makes the update inherently robust to the DP-noise injection, and renders the method scale-invariant. We outline these effects in more detail next:
>     - 1.a The key point is that *post-processing* preserves **privacy guarantees** and influeces **optimization performance**. In the same spirit, classic SignSGD [3] is also a simple post-processing of SGD: The sign operator makes it so SignSGD exhibits markedly different convergence behavior than SGD [3]. We prove that this is also the case in DP training.
>     - 1.b Intuitively, the DP noise added to DP-SGD can induce high-variance perturbations in the iterates, leading to solutions of lower quality. In contrast, the sign operation effectively clips the privatized gradient signal, capping the update magnitude and reducing sensitivity to noise corruption. This phenomenon is visible in Figure 1 and is fully explained in Theorems 4.1–4.4.
>     - 1.c Regarding which optimizer achieves a better "theoretical utility", we want to highlight that:
>         - **i)** under Protocol A, we show that *DP-SignSGD is better than SGD* **only** in high-privacy or small-batch (large gradient noise) regimes;
>         - **ii)** Under Protocol B, e.g. **under optimal hyperparameters**, DP-SGD (Theorem 4.6) and DP-SignSGD (Theorem 4.7) achieve the **same asymptotic privacy-utility tradeoff**, which is fully in line with [9] and [10, 11], which show that SGD and Adam achieve comparable performance **under optimal hyperparameters** in DP and non-DP LLM training.

---

> > ### Author Response · Authors · 2025-11-19
> > **Continuation & References**
> >
> > 2. **The role of the clipping bound C deserves clarification**
> > The apparent contradiction arises because **two different effects** depend on $C$:
> >     - **2.a Clipping bias (Phase 1):**
> >      A larger $C$ reduces the distortion of the stochastic gradient, which aligns with intuition.
> >     - **2.b DP noise scale:**
> >      The DP noise injected is proportional to **$C$**, so increasing $C$ *also increases the privacy noise scale*. In the Phase 2 SDE bounds for DP-SGD (Eq. 6) and DP-SignSGD (Eq. 9), the privacy-utility term always contains a **$C^2 \sigma_{\text{DP}}^2 / B^2$** contribution.
> >
> >    Thus, increasing $C$ simultaneously *reduces gradient bias but amplifies privacy noise*. **This is fully consistent with recent analyses of clipping bias versus DP noise** (e.g., [8]).
> >
> > 3. **Does your theory cover the fact that some gradients would be clipped and some would not?**
> > We thank the Reviewer for raising this important point. We are fully aware that different samples within a mini-batch may simultaneously fall into different regimes (clipped vs. unclipped). To address this formally, we have added a complete proof (see **Theorem B.8** for DPSGD and **Theorem B.14** for DPSignSGD in the revised appendix): For the sake of completeness, we added detailed proofs which are clearly a straightforward combination of the steps necessary for those on the individual phases. These results show that the same convergence bounds originally derived in Theorem 4.2 and Theorem 4.4 for the $L$-smooth case continue to hold under this mixed-phase setting. This is expected, as in our original derivations, we took a worst-case approach to handle the two phases.
> >
> >    We emphasize that our decision to present the “clipped’’ and “unclipped’’ dynamics separately in Theorems 4.1 and 4.3 was purely pedagogical: separating the two regimes makes the role of privacy noise and the impact of clipping substantially easier to interpret. The full mixed-phase proof confirms that these results hold without requiring the batch to be entirely in one phase. We hope this clarification resolves the Reviewer’s concern.
> >
> > 4. **Missing SOTA benchmarks**
> > We respectfully push back on the concern regarding missing comparisons with recent DP-SGD benchmarks. Our experimental setting is drawn **directly from the official Google DP2 repository** (GitHub: https://github.com/kenziyuliu/DP2), accompanying the paper [7] (ICLR 2023). That work introduced *new* DP optimizers and was **accepted at a top-tier venue (ICLR 2023) using this experimental setup**. In other words, these experiments have already been deemed sufficient by the DP community to validate *algorithmic innovations* at a top conference such as ICLR.
> >
> >    In our case, we do not aim to propose a new optimizer nor to compete on SOTA image-classification performance. Our goal is to **explain** the behavior of existing DP optimizers through an SDE framework. Since our experiments mirror those used to evaluate new methods in [7], we believe that they are more than adequate for our more modest purpose of validating the **theoretical insights** on existing methods.
> >
> >    Crucially, the experiments included already confirm every phenomenon predicted by our analysis. Adding SOTA benchmark comparisons is also well beyond our available compute by orders of magnitude, and would not test any additional aspect of the theory nor change the validation of the SDE predictions. Nevertheless, we have **added a new experiment on the StackOverflow dataset**: Its results are perfectly in line with our theoretical predictions -- see Figure C.3 in Appendix C.5. We hope this addresses the Reviewer's concern to some extent.
> >
> > ---
> >
> > ## References
> >
> > [1] Li et al., **Stochastic Modified Equations and Dynamics of Stochastic Gradient Algorithms I: Mathematical Foundations**, Journal of Machine Learning 2019
> >
> > [2] Compagnoni et al., **Adaptive Methods through the Lens of SDEs: Theoretical Insights on the Role of Noise**, ICLR 2025
> >
> > [3] Bernstein et al., **signSGD: Compressed Optimisation for Non-Convex Problems**, ICML 2018
> >
> > [4] Compagnoni et al., **Unbiased and Sign Compression in Distributed Learning: Comparing Noise Resilience via SDEs**, AISTATS 2025 (*Oral*)
> >
> > [5] Jastrzębski et al., **Three Factors Influencing Minima in SGD**, ICANN 2018
> >
> > [6] Xiao et al., **Exact Risk Curves of SignSGD in High Dimensions**, ICML 2025
> >
> > [7] Li et al., **Differentially Private Adaptive Optimization with Delayed Preconditioners**, ICLR 2023
> >
> > [8] Koloskova et al., **Revisiting Gradient Clipping: Stochastic bias and tight convergence guarantees**, ICML 2023
> > [9] Li et al., **Large Language Models Can Be Strong Differentially Private Learners**, ICML 2022
> >
> > [10] Srećković et al., **Is your batch size the problem? Revisiting the Adam-SGD gap in language modeling**, 2025
> >
> > [11] Marek et al, **Small Batch Size Training for Language Models: When Vanilla SGD Works, and Why Gradient Accumulation Is Wasteful**, NeurIPS 2025.

---

### Official Review · Reviewer_B6gH · 2025-10-31

**Soundness:** 3
**Presentation:** 3
**Contribution:** 4
**Rating:** 6
**Confidence:** 3

**Summary:**

The paper studies DP stochastic gradient descent and its variants via modelling the DP-SGD dynamics via stochastic differential equations. It develops continuous-time approximations for adaptive algorithms and distinguishes between two training “phases”: an initial exploration phase and a later convergence phase. The analysis aims to explain how noise injection from differential privacy interacts with the optimization dynamics, and how the privacy parameters and optimizers' hyperparameters (clipping bound, batch size and learning rate) influence convergence. The authors attempt to unify previous heuristics (DP-SGD, DP-SignSGD etc.) under a common SDE framework, claiming to provide an interpretable connection between privacy, optimization noise, and generalization. The paper also reports empirical results to illustrate the phase-transition behavior. There is a very interesting phenomenon that is supported both by theory and experiments: DP-SGD has $\varepsilon^{-2}$ behaviour for small $\varepsilon$-values whereas DP-Adam and DP-SignSGD have $\varepsilon^{-1}$ behaviour.

**Strengths:**

- The paper is well written, seems to be of very high quality.

- The fact that the SDE view is able to capture the experimental behaviour that DP-SGD has $\varepsilon^{-2}$ behaviour for small $\varepsilon$-values whereas DP-Adam and DP-SignSGD have $\varepsilon^{-1}$ behaviour (see Figure 1) is very impressive.

- The SDE view is well motivated and also commonly considered in the literature (e.g., Blei et al. 2018).

**Weaknesses:**

- The paper focuses on only on few adaptive optimizers, and I am a bit surprised about their choices: DP-Adam (Adam with DP gradients) and DP-SignSGD (which is not that well-known). The reason might be that the analysis is amenable for them (questions below), and I think the contribution is very valuable neertheless.

- Due to the fact that very few adaptive optimizers seem to actually fit into this SDE framework (or can be seen as discretizations of SDEs, meaning that the weakly converge to them in the vanishing step size limit), I have a feeling that this framework does not actually help in designing new hyperparameter adaptive DP optimizers. The paper seems thus to give an analytical explanation of certain differences in the optimizers' behaviors ( $\varepsilon^{-1}$ vs.  $\varepsilon^{-2}$ error behaviour).

**Questions:**

DP-Adam refers to the algorithm considered by (Balles and Hennig, 2018), i.e., to the plain Adam with DP-SGD gradients, right? Why not to analyze other adaptive optimizers like the versions of DP-Adam that are tailored for DP (e.g., Tang and Lécuyer, 2023, or Li et al., 2023, the references you also list)? Were DP-Adam and DP-SignSGD chosen because the analysis is amenable for them?

Recently, certain filtering methods have turned out to give good privacy-utility trade-offs, see e.g.

Zhang, X., Bu, Z., Balle, B., Hong, M., Razaviyayn, M., & Mirrokni, V. DiSK: Differentially Private Optimizer with Simplified Kalman Filter for Noise Reduction. In The Thirteenth International Conference on Learning Representations 2025.

Could this SDE view allow analyzing those methods as well?

---

> ### Author Response · Authors · 2025-11-19
> **Many Thanks - Addressed Weaknesses and Questions**
>
> We truly thank the reviewer B6gH for their positive review. We are pleased to read that the reviewer found our work **"well written"** and **"of very high quality"** and appreciate how they found our contribution to be **excellent** with **"impressive"** insights. We are pleased to hear that the Reviewer agrees that our **"SDE view is well motivated and also commonly considered in the literature"**.
>
> We hope that the following clarifications will help re-evaluate our submission and potentially adjust the score.
>
>
> ## Weaknesses
>
> 1. **Why focus on these optimizers (DP-SGD, DP-SignSGD, and DP-Adam)?**
> Yes, the reviewer is correct. Our choice reflects the fact that the algorithms we study are more amenable to mathematical analysis while remaining closely aligned with practical methods. More precisely, this work presents **the first SDE-based analysis of DP adaptive optimizers**, and we therefore chose optimizers whose dynamics remain mathematically tractable. Since SignSGD has a long history as a proxy for Adam in the non-DP literature [2–5],  we study DP-SignSGD **as a proxy for the popular DP-Adam**, and we complement the theory with **empirical experiments**, confirming that the qualitative insights obtained through the SDE of DP-SignSGD also extend to DP-Adam. This reinforces that the resulting insights are both broad and practically relevant.
>
> 2. **Very few adaptive optimizers fit into this SDE framework & no novel methods**
> Our paper focuses on explaining the difference between DP-SGD, DP-SignSGD, and DP-Adam. Understanding these differences is **key to practically guide** the selection of the correct method for a specific setup, e.g., high-privacy or low-budget. Additionally, insights derived from this kind of analysis could lead to the design of improved optimizers, which is indeed a promising research direction: We view our work as an important foundation for such future developments.
>
>     Regarding the scope of the SDE framework, we respectfully clarify that this is broader than suggested. In fact, many modern adaptive optimizers have been analyzed through SDE techniques, including RMSProp, Adam, AdamW, SignSGD, and some distributed extensions (see Appendix A of [1] and references therein).
>
> ---
>
> ## Questions
>
> We number the answers following the Reviewer’s ordering.
>
> 1. **DP-Adam refers to the algorithm inspired by (Balles and Hennig, 2018)?**
> Yes, throughout the paper, DP-Adam refers to **plain Adam where the stochastic gradients are replaced by DP-SGD gradients**, consistent with the conventions used in the DP literature: We will further clarify this in the paper.
>
> 2. **“Why not include DP-specific Adam variants in your analysis?”**
> As discussed in Weaknesses 1, our goal is to provide the first SDE-based analysis of DP adaptive optimizers. For this purpose, we focused on the most standard baselines: **DP-SGD** [7] and **DP-Adam** [2,6]. Analyzing other variants is feasible but requires a **separate, dedicated treatment**, which we defer to future works. As noted above, a broad family of modern optimizers has already been analyzed using SDE techniques, each requiring its technical innovations.
>
> 3. **Could the SDE view be used to analyze filtering-based DP optimizers (e.g., DiSK)?**
>    In principle, yes: While deriving the SDE is likely doable and would take heavy inspiration from [8], the convergence analysis would require extensive analysis of its own, which lies beyond the scope of this work.
>
> ---
>
> ## References
>
> [1] Compagnoni et al., **Adaptive Methods through the Lens of SDEs: Theoretical Insights on the Role of Noise**, ICLR 2025
>
> [2] Balles et al., **Dissecting Adam: The Sign, Magnitude and Variance of Stochastic Gradients**, PMLR 2018
>
> [3] Zou et al., **Understanding the Generalization of Adam in Learning Neural Networks with Proper Regularization**, OPT 2021
>
> [4] Peng et al., **Simple Convergence Proof of Adam From a Sign-like Descent Perspective**, 2025
>
> [5] Li et al., **On the Optimization and Generalization of Two-layer Transformers with Sign Gradient Descent**, ICLR 2025
>
> [6] Ganesh et al., **On Design Principles for Private Adaptive Optimizers**, 2025
>
> [7] Abadi et al., **Deep Learning with Differential Privacy**, ACM SIGSAC 2016
>
> [8] Su et al., A Differential Equation for Modeling Nesterov's Accelerated Gradient Method: Theory and Insights, NIPS 2014

---

### Official Review · Reviewer_a1Sd · 2025-11-01

**Soundness:** 4
**Presentation:** 3
**Contribution:** 3
**Rating:** 6
**Confidence:** 4

**Summary:**

-  This paper investigates differentially private learning through the lens of a stochastic differential equation. Based on theoretical understanding, the paper investigates how the DP-SGD, DP-SignSGD, and its variant DP-Adam perform in various privacy budgets. The authors argue that DP-SignSGD is epsilon-dependent, which reduces the burden of parameter tuning in DPDL.

**Strengths:**

-	The paper investigates the optimization process of differentially private learning in terms of SDE, which has not been actively investigated.
-	The authors provide a theoretical analysis of why DP-SGD and DP-SignSGD differ in training dynamics, especially with hyperparameter setups.
-	Based on their observations, the authors argue two protocols that cover both fixed and tuning parameters.

**Weaknesses:**

Please refer to the Questions section.

**Questions:**

-	The paper investigates the difference between DP-SGD and DP-SignSGD in terms of differentially private deep learning. Is there any related work on non-private optimization, or does this comparison solely rely on a DP sense?
-	For protocol A, how did the authors choose the parameters? For private learning, the clipping value is also as important as the learning rate. Can the authors provide more results while varying hyperparameters in both protocols A and B?
-	The paper’s analysis is based on that the optimal performances of DP-SGD and DP-SignSGD are almost similar (without considering parameter search). However, as far as the reviewer knows, the current methods prefer DP-SGD compared to DP-SignSGD. Does DP-SignSGD still provide comparable results with private fine-tuning or bigger architectures? Refer to [1] or recent tuning methods in larger vision or language-based DP papers.

    [1] Unlocking High-Accuracy Differentially Private Image Classification through Scale, 2022.

-	What about the case of DP-SGD-based Adam, instead of DP-SignSGD-based Adam?

---

> ### Author Response · Authors · 2025-11-19
> **Many Thanks - Addressed Weaknesses and Questions**
>
> We sincerely thank the Reviewer a1Sd for the positive observations, underlining the **novelty** and **soundness** of our **theoretical SDE-based analysis** for DP optimization, as well as how our work addresses **practitioner-oriented scenarios**.
>
> We hope that the following clarifications will help re-evaluate our submission and potentially adjust the score.
>
> ## Questions:
>
> 1. **Non-DP comparison between SignSGD and SGD**
> A comparison between SignSGD and SGD in the **non-DP** setting has indeed been studied: see [1], which analyzes adaptive methods through the lens of SDEs, including SignSGD, SGD, AdamW, and related optimizers.
>
> 2. **Hyperparameter tuning and fairness of the experimental comparison**
> We split our answer into subpoints for clarity.
>
> - **2a. Choice of hyperparameters (learning rate and clipping)**
>   Protocols A and B are defined as in lines 60 to 65:
>
>      - **ii)** In Protocol A, we fix the privacy budget $\varepsilon$ as per [2] and perform an extensive grid search to find the optimal $(\eta, C)$ for this specific $\varepsilon$. Then, we **keep** $(\eta, C)$ fixed and study the dependency of the loss as we vary $\varepsilon$;
>      - **ii)** In Protocol B, for each value of $\varepsilon$, we perform extensive grid search to find the optimal $(\eta,C)$.
>
>     As detailed in lines 1917–1952, in **both Protocol A and Protocol B** we perform a **grid search jointly over learning rates and clipping thresholds**. In particular, for each optimizer and each protocol, we sweep over a broad range of learning rates and clipping values, following the methodology of [2] and using the openly available DP2 codebase from Google (GitHub: https://github.com/kenziyuliu/DP2). We always select the **best-performing configuration** within this grid, which ensures a fair comparison between DP-SGD and DP-SignSGD for both protocols.
>
> - **2b. On varying hyperparameters and additional results**
>   We agree that the clipping value is as important as the learning rate in private learning. In our experiments, we aimed to tune both the clipping threshold and the learning rate as fairly as possible, often considering search grids even wider than [2], as detailed in Appendix C of our manuscript. We made sure that the optimal hyperparameter does not lie on the edge of the grid, extending our search space if necessary.
>
>   An illustrative example is provided in the table below, which reports the grid over $(\eta, C)$ used for DP-Adam on IMDB, from which we selected $(\eta = 0.1, C = 0.5)$ as the optimal configuration for Figures 1, 2, and 3.
>
>
> | **lr \ C** | 0.1 | 0.25 | **0.5** | 1.0 |
> |--------|-----------------------|-----------------------|-----------------------|-----------------------|
> | 0.0001 | (0.691, 0.541) | (0.691, 0.541) | (0.691, 0.549) | (0.692, 0.561) |
> | 0.001  | (0.684, 0.541) | (0.684, 0.541) | (0.680, 0.602) | (0.686, 0.605) |
> | 0.01   | (0.698, 0.599) | (0.657, 0.621) | (0.631, 0.657) | (0.651, 0.648) |
> | **0.1**    | (0.640, 0.709) | (0.577, 0.719) | **(0.561, 0.720)** | (0.603, 0.682) |
> | 1.0    | (1.858, 0.684) | (1.884, 0.673) | (1.917, 0.663) | (1.988, 0.648) |
>
>
> Due to the computational constraints, we did not consider even wider grids. However, we believe our choice of grids is adequate as it is wider than [2]. If the Reviewer has a specific concern about a particular region of the hyperparameter space (e.g., much larger/smaller clipping values or learning rates), we would be glad to take this into account and, where feasible, expand our sweeps accordingly.
>
> 3. **DP-SGD vs. DP-SignSGD vs. DP-Adam**
> Thank you for raising this point. First, we emphasize that since studying SignSGD as a proxy for Adam is standard in the literature [6-9], we introduced DP-SignSGD in our paper **primarily as a simplified proxy** for understanding **DP-Adam**, which is a widely adopted optimizer in DP deep learning (see [11-13]).
>
>     To reply to the question: As we discuss in our paper, [12] showed that DP-SGD and DP-Adam do achieve comparable performances if hyperparameters are appropriately tuned in DP LLM training. Similarly, the same has been shown to hold also on non-DP settings in LLM training [3,5]. This empirical evidences support our claims that under optimal hyperparameter tuning, DP-SGD and DP-SignSGD should achieve comparable performance.
>
>     Finally, while the **paper** presented by the Reviewer did prefer DP-SGD, they tackle vision tasks, where SGD with momentum is canonically preferred to Adam in non-DP learning. Therefore, it is not surprising that they opted for DP-SGD over DP-Adam. On the same note, such a **paper** cites [10], which in turn preferred to use DP-Adam to train DP-BERT. Therefore, which of DP-SGD and DP-Adam is preferable remains an open question, and **our results provide some insight into this phenomenon**.

---

> > ### Author Response · Authors · 2025-11-19
> > **Continuation & References**
> >
> > 4. We thank the Reviewer for this question. In our work, the algorithms we study are **DP-SGD** and **DP-SignSGD**, defined in Definition 3.3 (lines 238–253), and **DP-Adam**, which corresponds to Algorithm 5 in [4]. In particular, our DP-Adam implementation is **DP-SGD–based**: it uses the DP-SGD gradient as input to the Adam updates.
> > Designing an Adam-style method based on **DP-SignSGD** instead would likely lead to an algorithm very similar to **Signum** [14], which one might naturally call **DP-Signum**. Given the strong performance of Signum in the non-private setting, we agree that such a DP-Signum variant would be an interesting alternative to explore in DP training. However, it is beyond the scope of our work, where we focus on the two popular algorithms, DP-SGD and DP-Adam.
> >
> > ---
> >
> > ## References
> >
> > [1] Compagnoni et al., **Adaptive Methods through the Lens of SDEs: Theoretical Insights on the Role of Noise**, ICLR 2025
> >
> > [2] Li et al., **Differentially Private Adaptive Optimization with Delayed Preconditioners**, ICLR 2023
> >
> > [3] Srećković et al., **Is your batch size the problem? Revisiting the Adam-SGD gap in language modeling**, 2025
> >
> > [4] Ganesh et al., **On Design Principles for Private Adaptive Optimizers**, 2025
> >
> > [5] Marek et al, **Small Batch Size Training for Language Models: When Vanilla SGD Works, and Why Gradient Accumulation Is Wasteful**, 	NeurIPS 2025.
> >
> > [6] Balles et al., **Dissecting Adam: The Sign, Magnitude and Variance of Stochastic Gradients**, PMLR 2018
> >
> > [7] Zou et al., **Understanding the Generalization of Adam in Learning Neural Networks with Proper Regularization**, OPT 2021
> >
> > [8] Peng et al., **Simple Convergence Proof of Adam From a Sign-like Descent Perspective**, 2025
> >
> > [9] Li et al., **On the Optimization and Generalization of Two-layer Transformers with Sign Gradient Descent**, ICLR 2025
> >
> > [10] Anil et al, **Large-scale differentially private BERT**, 2021.
> >
> > [11] Zhou et al., **Private Stochastic Non-Convex Optimization: Adaptive Algorithms and Tighter Generalization Bounds**, 2020
> >
> > [12] Li et al., **Large Language Models Can Be Strong Differentially Private Learners**, ICML 2022
> >
> > [13] Gylberth et al., **Differentially private optimization algorithms for deep neural networks**, ICACSIS 2017
> >
> > [14] Bernstein at al., **signSGD: Compressed Optimisation for Non-Convex Problems**, ICML 2018

---

### Official Review · Reviewer_4NeB · 2025-11-07

**Soundness:** 2
**Presentation:** 2
**Contribution:** 2
**Rating:** 4
**Confidence:** 3

**Summary:**

Thru discussing how DP noise interacts with adaptivity in optimization, DP-SGD and DP-SignSGD are proposed in this work, where DP-SGD is shown to be converged at a speed independent of ε, DP-SignSGD is with convergence speed scales linearly in ε. Under optimal learning rates, both methods reach comparable theoretical asymptotic performance, while this leaves potential issues in practice.

**Strengths:**

- SDE-based analysis of differentially private optimizers, using this framework to expose how DP noise interacts with adaptivity and batch noise.
- DP-SGD is shown ito be converged at a speed independent of ε.
- DP-SignSGD: its convergence speed scales linearly in ε, while its privacy-utility trade-off scales as O (1/ε)

**Weaknesses:**

- The assumptions on SNR (signal-to-noise ratio) are built on linear approximations that are only valid in a high-noise, low-signal regime.
- A general Student-t distribution for batch noise is used to capture heavy tails, while it is not used consistently in assumption B.2..
- The experimental validation for Protocol B on the StackOverflow dataset is missing.

**Questions:**

- The theoretical analysis is derived for DP-SignSGD, while the conclusions are empirically with DP-Adam. Please provide more discussions and valiations.
- The experimental validation for Protocol B on the StackOverflow dataset needs to be provided.

---

> ### Author Response · Authors · 2025-11-19
> **Many Thanks - Addressed Weaknesses and Questions**
>
> We sincerely thank Reviewer 4NeB for their thorough review.
>
> We hope that the following clarifications will help re-evaluate our submission and potentially adjust the score.
>
> ## Weaknesses
>
> 1. **SNR assumption and linear approximations**
> The SNR assumption we use enters only when approximating the *direction* of the normalized noisy gradient. In Lemmas A.1–A.2 we start from the exact expression for $\mathbb{E}[\nabla f_\gamma(x) / \|\nabla f_\gamma(x)\|]$ and show that a first-order approximation is tight whenever $\frac{\lvert \nabla f(x) \rvert_2^2}{2 \sigma_{\gamma}^2(d+2)} \ll 1$. Importantly:
> - 1.a The number of trainable parameters $d$ is assumed to be large ($d>10^4$), which is not restrictive;
> - 1.b [1] has extensively experimentally verified that the ratio  $\frac{\lvert \nabla f(x) \rvert_2^2}{2 \sigma_{\gamma}^2} \ll 1$ is at most $O(10^2)$ on a multitude of large-scale deep learning architectures and relevant datasets.
>
> Therefore, our assumption that  $\frac{\lvert \nabla f(x) \rvert_2^2}{2 \sigma_{\gamma}^2(d+2)} \ll 1$ **does not require a high-noise, low-signal regime to hold**, but only requires the number of parameters $d > 10^4$, that is, a **mild assumption** for modern deep learning setups. Importantly, we emphasize that this **assumption is not restrictive in practice**: our theoretical insights are **fully verified on all our benchmarks**, which demonstrates that such an assumption is mild.
>
> 2. **Clarifying Assumption B.2**
> We thank the Reviewer for pointing out the need for additional clarity in Assumption B.2. In Phase 1, each per-example gradient is clipped individually. To model the stochasticity of computing the stochastic gradient on a single datapoint, we model the batch noise as a Student’s t distribution, which is heavy-tailed and more general than what is commonly done in the literature [2-6]. In Phase 2, the clipping operator is not active. Therefore, the fact that the gradients of each datapoint are averaged together within the batch allows us to model the overall noise as Gaussian, which is standard in the literature [2-6]. We clarified this two-phase modeling choice more explicitly in the revised version.
>
> 3. **Experimental validation of Protocol B on StackOverflow**
> We agree that the experimental validation for Protocol B on the StackOverflow dataset is important: We have **added this new experiment** and included it in Figure 4 of the revised paper, together with a detailed description of the experimental setup in Appendix C.5. Our findings are perfectly in line with our theoretical predictions.
>
> ## Questions
>
> 1. **"How do DP-SignSGD and DP-Adam relate to each other?"**
> The theoretical results we derive apply directly to DP-SGD and DP-SignSGD. Although DP-SignSGD is not widely used in practice, it is substantially simpler to analyze than DP-Adam, which is popular in differentially private deep learning [12-14]. Relying on SignSGD as a proxy for Adam is standard in prior work [7-11], and this motivates our focus on DP-SignSGD for the theoretical development. Importantly, setting $\beta_1=\beta_2=0$ reduces DP-Adam to DP-SignSGD. Finally, it is key to observe that our experiments (see Figure 1-4) show that the qualitative behaviors predicted by our analysis for DP-SignSGD also appear in DP-Adam, reinforcing that the resulting insights are both broad and practically relevant.
>
>
> 2. **Experimental validation of Protocol B on StackOverflow**
> As noted above, we have added the missing experiment for Protocol B on the StackOverflow dataset as Figure 4 and provided all relevant experimental details in Appendix C.5 of the revised version of the paper.

---

> > ### Author Response · Authors · 2025-11-19
> > **References**
> >
> > ## References
> >
> > [1] Malladi et al., **On the SDEs and Scaling Rules for Adaptive Gradient Algorithms**, NeurIPS 2022
> >
> > [2] Mandt et al., **A variational analysis of stochastic gradient
> > algorithms**, ICML 2016
> >
> > [3] Wu et al., **On the noisy gradient descent that generalizes as sgd**, ICML 2020
> >
> > [4] Xi et al., **Positive-negative momentum: Manipulating stochastic gradient noise to improve generalization**, ICML 2021
> >
> > [5] Li et al., **Stochastic modified equations and adaptive stochastic gradient algorithms**, ICML 2017
> >
> > [6] Jastrzebski et al., **Three factors influencing minima in sgd**, ICANN 2018
> >
> > [7] Compagnoni et al., **Adaptive Methods through the Lens of SDEs: Theoretical Insights on the Role of Noise**, ICLR 2025
> >
> > [8] Balles et al., **Dissecting Adam: The Sign, Magnitude and Variance of Stochastic Gradients**, PMLR 2018
> >
> > [9] Zou et al., **Understanding the Generalization of Adam in Learning Neural Networks with Proper Regularization**, OPT 2021
> >
> > [10] Peng et al., **Simple Convergence Proof of Adam From a Sign-like Descent Perspective**, 2025
> >
> > [11] Li et al., **On the Optimization and Generalization of Two-layer Transformers with Sign Gradient Descent**, ICLR 2025
> >
> > [12] Zhou et al., **Private Stochastic Non-Convex Optimization: Adaptive Algorithms and Tighter Generalization Bounds**, 2020
> >
> > [13] Li et al., **Large Language Models Can Be Strong Differentially Private Learners**, ICML 2022
> >
> > [14] Gylberth et al., **Differentially private optimization algorithms for deep neural networks**, ICACSIS 2017

---

### Author Response · Authors · 2025-11-19
**General Answer**

Dear Reviewers and AC,

We sincerely appreciate your time, thorough reviews, insightful comments, and interesting questions regarding our paper. Your feedback has greatly contributed to the finalization of our work.

We are pleased that reviewers appreciated our **presentation** (B6gH, A1Sd), as they found our paper to be **well-presented**, "**well-written**" and **"of very high quality"** as it provides "an **interpretable** connection between privacy [and] optimization noise" (B6gH).

Importantly, our **contribution** is deemed good (A1Sd) and **"well-motivated"** (g3be and B6gH), even **"excellent"** by B6gH. B6gH wrote that our contribution provides a "unifying framework for DP and optimization", and that it is **"very impressive"** that our "SDE view is able to capture experimental behaviours". Our results were judged to be **"valuable"**, providing **"strong insights"** which offer "**clear scaling laws** for DP-SGD and DP-SignSGD" (g3be). In terms of **soundness**, A1Sd rated our analysis as **excellent** and B6gH as good, while other reviewers encouraged us to provide further clarifications; **we address these points in detail below**.

From a **novelty** perspective, g3be recognized that this is the "**first** theoretical framework that connects SDE with convergence dynamic[s]" of DP optimizers, a topic which had "not [been] investigated yet" (A1Sd): Ours are **"strong insights"** (g3be) which elucidate "**very interesting** phenomen[a] that [are] supported both by **theory and experiments**" (B6gH).

From a **practical perspective**, g3be praised that we "give **practical guidance**" as we "cover **multiple** tuning protocols under different budget constraints". From an **experimental** perspective, g3be observed that our "theoretical predictions are **thoroughly validated** by experiments across **multiple** datasets", which we further **extended** during the rebuttal to address the requests from some reviewers.

We, of course, take the criticisms very seriously, and we have devoted time to addressing them below. Specifically, the weaknesses and questions concern (i) our modeling assumptions (SNR and noise modeling); (ii) intuition behind the role of the $Sign$ operator; (iii) the relation between DP-SignSGD and DP-Adam; (iv) scope and justification for the SDE framework, including how we treat clipped vs. unclipped updates; (v) the routine used to choose and tune hyperparameters in Protocols A and B; and (vi) the role of the clipping threshold $C$, together with some additional experimental validation.

We stress that these points are about **clarifications** rather than about the core novelty or relevance of our framework. All of them are **properly taken care of** in the updated version through strengthened explanations, extended theoretical discussion (including **new proofs**), and **additional experiments**.

To highlight the relevant changes made to the original submission, we marked them using *purple* in the revised version. Once again, we are thankful to the reviewers for their constructive feedback. We look forward to the upcoming author-reviewer discussion period.

Thank you for your attention.

Best regards,

The Authors

---

### Meta-Review · Area_Chair_bz9G · 2026-01-05

**Summary:**

The paper gives a SDE-based framework for comparing privacy-utility tradeoffs of different optimizers. The theory predicts a better scaling w.r.t. eps for SignSGD (a common proxy for Adam) than SGD, and experimental justification is provided. Reviews were generally positive, with questions about some "leaps of faith" needed in the theory (dropping some terms, SignSGD ~ Adam, etc.) Some of the concerns were addressed as misunderstandings in the rebuttal.

I think the paper is technically solid and the assumptions / simplifications made are relatively standard for "deep learning optimization theory". The overall framework has good potential for providing further insights. I recommend acceptance, with the feedback from reviewers incorporated.

**Reviewer Concerns:**

I think the rebuttal did a good job of clarifying several assumptions / design decisions made in the analysis, e.g., restricting the scope to a few common optimizers. It also clarified the choices made in the experimental setup. The response seemed pretty comprehensive.

**Reviewer Scores:**

The rebuttal seemed to comprehensively address the weaker reviews' questions. I think it's plausible they would both raise their scores.

---

### Decision · Program_Chairs · 2026-01-26

Accept (Poster)